# Anhedonic Traits Do Not Impair Performance in a 3-Arm Bandit Task

**RESEARCH ARTICLES**

ju[ ubiquity press

**ARJUN RAMASWAMY** (ID)

**YUMEYA YAMAMORI** (ID)

**UMESH VIVEKANANDA** (ID)

**VLADIMIR LITVAK** (ID)

**JONATHAN P. ROISER** (ID)

*Author affiliations can be found in the back matter of this article

## ABSTRACT

Anhedonia, a transdiagnostic symptom marked by diminished reward sensitivity, is often linked to impairments in reinforcement learning (RL). Standard tasks (e.g., the 4-arm bandit) can place substantial demands on participants and may blur valuation with other processes. We therefore adapted a three-arm bandit (3AB) task from Seymour et al. (2012), incorporating design features intended to lessen task demands (fewer options; denser feedback) while enabling separate estimation of reward and punishment learning rates and sensitivities. In an online sample pre-screened for anhedonia (N = 206; 111 anhedonic, 95 non-anhedonic), hierarchical Bayesian modelling using a four-parameter specification showed no credible group differences in reward learning rate, punishment learning rate, reward sensitivity, or punishment sensitivity; Bayes factors favoured the null ($BF_{01}$ = 3.36–5.96). Model-agnostic win-stay/lose-shift strategies likewise showed no group differences (Welch's tests, all p > .05). Posterior predictive checks indicated above-chance choice prediction: the model's highest-probability action matched participants' actual choices on 59.6% of trials (chance = 33%). Parameter recovery was excellent for valuation parameters (r = 0.96–0.97) and acceptable for learning rates (r = 0.67–0.85). Simulations generated from fitted parameters preserved individual-difference structure, with high correlations between observed and simulated win-stay (r = 0.89 anhedonic; 0.86 non-anhedonic) and moderate correlations for lose-shift (r = 0.62; 0.67), alongside small systematic mean-level biases (simulated win-stay lower by 3.5–4.9 percentage points; simulated lose-shift higher by 12.8–13.2 points). Model comparison showed that lapse-augmented variants achieved marginally better predictive fit, but group comparisons under both lapse models yielded overlapping posteriors with 95% HDIs including zero for all learning, sensitivity, and lapse parameters, indicating that the null findings were robust to inclusion of lapse terms. Non-anhedonic participants also responded more slowly on average than anhedonic participants, which we treat as exploratory. Together, these results suggest that in this 3AB task, anhedonia is not reliably associated with differences in core RL parameters or simple choice strategies, while providing a detailed characterisation of model performance and limitations in an online setting.

**CORRESPONDING AUTHOR:**
**Arjun Ramaswamy**

UCL Queen Square Institute of Neurology, London, UK; Department of Imaging Neuroscience, UCL, London, UK

arjun.ramaswamy.19@ucl.ac.uk

**KEYWORDS:**
Anhedonia; reinforcement learning; 3-arm bandit; Hierarchical Bayesian modelling; reward and punishment sensitivity; Online behavioural study

**TO CITE THIS ARTICLE:**

Ramaswamy et al. **59**
*Computational Psychiatry*

# 1 INTRODUCTION

Anhedonia, classically defined as a diminished ability to experience pleasure, is a core feature of major depressive disorder (American Psychiatric Association, 2013). More recent conceptualizations extend this definition to include impairments in reward valuation and subjective responsiveness to positive stimuli (Treadway & Zald, 2011; Der-Avakian & Markou, 2012). Importantly, anhedonia is distinct from motivational deficits such as apathy or effort discounting (Husain & Roiser, 2018). Instead, it may reflect a more specific reduction in reward sensitivity—the hedonic impact or subjective value of rewarding outcomes—rather than impaired capacity to pursue them (Hall et al., 2024). As a transdiagnostic symptom, anhedonia contributes to clinical burden across multiple disorders including depression, schizophrenia, PTSD, and substance use, and is associated with poor treatment response and elevated relapse risk (Culbreth et al., 2018; Nawijn et al., 2015; Garfield et al., 2014; Winer et al., 2019).

Validated self-report tools such as the Snaith-Hamilton Pleasure Scale (SHAPS; Snaith et al., 1995) and the Dimensional Anhedonia Rating Scale (DARS; Rizvi et al., 2015, 2016) are widely used to measure anhedonia. The DARS, in particular, captures domain-specific deficits across hobbies, social interaction, sensory experiences, and food/drink, and has demonstrated better psychometric sensitivity than SHAPS. However, the link between self-reported anhedonia and objective reward behaviour remains unclear. While some studies report that anhedonia is associated with blunted reward responsiveness or reduced learning (Kumar et al., 2008; Huys et al., 2013), others find no consistent associations (Harlé et al., 2017; Halahakoon et al., 2020; Pike & Robinson, 2022).

Reinforcement learning (RL) models allow formal estimation of latent cognitive variables that shape decision-making, including learning rates, reward sensitivity, punishment sensitivity, and decision noise (Sutton & Barto, 2018; Daw et al., 2011). Such models are increasingly used in computational psychiatry to parse affective symptoms into mechanistic components (Ahn et al., 2017; Whitton et al., 2015). However, a recent meta-analysis showed that RL differences between individuals with and without depression are modest in size and highly task-dependent (Pike & Robinson, 2022). Notably, reward sensitivity parameters—reflecting the subjective value assigned to rewarding outcomes—may be more closely tied to anhedonia than learning rate or exploration parameters (Kieslich et al., 2022). This is supported by theoretical models that separate "liking" (hedonic valuation) from "wanting" (motivational drive) in the neuroscience of reward (Treadway & Zald, 2011; Berridge & Robinson, 2003).

Multi-armed bandit (MAB) tasks are widely used to study dynamic reward-based learning. However, standard versions like the 4-armed bandit (Daw et al., 2006) are cognitively demanding and typically only model reward, omitting losses or punishments. Here, we adapted the paradigm introduced by Seymour et al. (2012), which involves separate drifting reward and punishment values for each choice option—allowing independent estimation of reward and punishment sensitivity. Our task reduced the number of options from four to three, lowering working memory demands (from 8 expected values to 6) and making the paradigm more suitable for online deployment. We also provided outcome feedback on every trial to keep participants engaged and to increase the number of informative feedback events available for model fitting.

While recent studies have adopted other 3-armed paradigms (e.g., Yan et al., 2025), our task differs in several key respects. Most notably, we included both reward and punishment outcomes to model approach and avoidance learning separately. This choice reflects evidence that depression and anhedonia often involve abnormalities in both reward and punishment processing, and that avoidance of losses can shape reward seeking in clinically relevant ways. Furthermore, we used hierarchical Bayesian modelling (Ahn et al., 2017) to estimate distinct learning rates and sensitivity parameters for reward and punishment, in contrast to Yan et al.'s use of Kalman filtering to examine latent volatility and stochasticity in relation to apathy and anxiety. This modelling framework allows us to test the hypothesis that trait anhedonia reflects reduced sensitivity to reward outcomes, rather than impaired learning or increased randomness.

Given our large-scale online recruitment strategy, we also anticipated deviations from canonical symptom correlations. Specifically, in online non-clinical samples, recent studies have shown that measures of anhedonia, depression, and anxiety often exhibit weaker associations than in clinical

*Computational Psychiatry*

cohorts—potentially due to subclinical symptom levels or response style variability (Ho et al., 2024; Niu et al., 2024). To mitigate concerns about inattentive responding or invalid data, we used item-level attention checks across all measures and excluded participants who failed any check—consistent with best-practice guidelines for online psychiatric research (Zorowitz et al., 2023).

In this study, we screened 1,000 participants using SHAPS and DARS to recruit two extreme groups: individuals high vs. low in anhedonia. A total of 206 participants (111 high-anhedonia, 95 controls) completed the 3AB task. We modelled their behaviour using a hierarchical reinforcement learning framework and compared groups on both model-derived parameters (reward/punishment sensitivity, learning rates) and model-agnostic behavioural metrics. Based on existing theories and prior empirical work, we predicted that anhedonic individuals would show blunted reward sensitivity, but we made no strong predictions regarding punishment sensitivity or learning rate.

## 2 RESULTS

### 2.1 TASK OVERVIEW AND VALIDATION

The 3-arm bandit (3AB) task (Figure 1a) was designed as a simplified and modified version of the traditional 4-arm bandit (4AB) task (Seymour et al., 2012). The 4AB task, while effective in modelling reward and punishment learning, entails a higher cognitive load due to the requirement to maintain reward and punishment probabilities associated with four stimuli over time. By reducing the number of choices from four to three, participants have to track only three sets of reward and punishment probabilities instead of four, simplifying the learning process while preserving reinforcement learning mechanisms. Additionally, the probabilities of encountering both reward and punishment were increased by a factor of 1.5 compared to the 4AB task (in which the probability fluctuated between 0 and 0.5; we increased the ceiling to 0.75) with the intention of creating a more engaging experience that would better highlight individual differences in learning behaviour. This structure is intended to simplify the decision-making process while maintaining sensitivity to learning impairments, particularly those related to anhedonia.

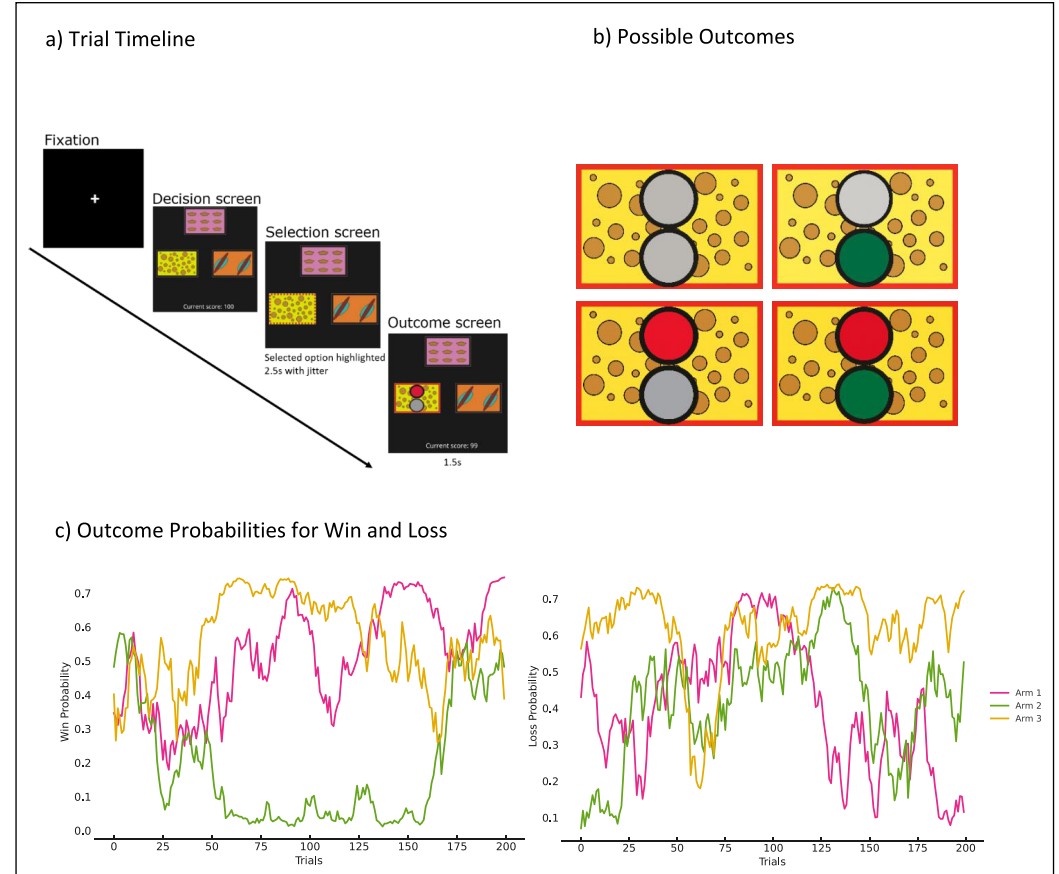

**Figure 1 Structure of the modified 3-Arm Bandit (3AB) Task. (a)** Visual representation of the modified 3-arm bandit task. Subjects choose between three options (arms), each associated with distinct reward and punishment probabilities. **(b)** Any arm selection can result in one of the four possible outcomes i.e. nothing, win token (green only), loss token (red only), or both. The task was designed to reduce cognitive load compared to the traditional 4-arm bandit task by limiting the number of choices and increasing the probability of both reward and punishment outcomes across trials. Outcome probabilities of win and loss events shown in **(c)**.

A total of 111 participants (mean age = 40, SD = 12, 50% female) completed the initial validation study, which involved completing both the 3AB and 4AB tasks in randomized order. We applied the same hierarchical model (banditNarm_4par, i.e. 4 parameter model of reward learning rate [Arew], punishment learning rate [Apun], reward sensitivity [R], and punishment sensitivity [P]) to both datasets and evaluated the consistency of estimated parameters across tasks. As shown in Figure 2, the reward learning rate (r = 0.49, $p < 10^{-7}$), punishment learning rate (r = 0.46, $p < 10^{-6}$), reward sensitivity (r = 0.52, $p < 10^{-8}$), and punishment sensitivity (r = 0.61, $p < 10^{-12}$) each showed moderate-to-strong positive correlations between the two task formats. These findings suggest that the simplified 3AB task preserves core reinforcement learning mechanisms measured by the original 4AB task. As such, it offers a valid alternative for probing learning processes in populations where cognitive load may be a limiting factor.

Ramaswamy et al. **61**
*Computational Psychiatry*

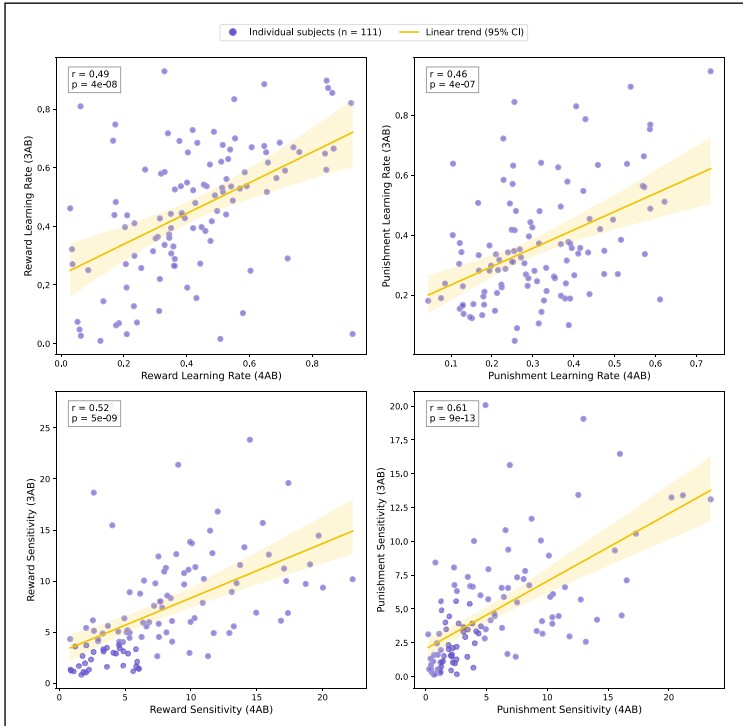

**Figure 2 Validation of the 3AB Task: Correlational Analysis of 4AB and 3AB Model Parameters.** Scatter plots illustrating the correlations between corresponding model parameters from the old 4AB task and the new 3AB task, using data from 111 subjects. Each subplot displays the Pearson correlation coefficient (r) and p-value, assessing the consistency of individual performance across reward learning rate (Arew), punishment learning rate (Apun), reward sensitivity (R), and punishment sensitivity (P). These plots aim to evaluate the validity of the new 3AB task by demonstrating whether similar patterns of behaviour are observed across both tasks.

To further assess the consistency of decision-making behaviour across tasks, we compared model-agnostic strategy use (win-stay and lose-shift percentages) between the 3AB and 4AB tasks. Figure 3 presents group-level bar plots for each strategy. Win-stay behaviour was similar across tasks (**3AB: Mean = 82.0%, SD = 27.5; 4AB: Mean = 81.0%, SD = 26.6**), as was lose-shift behaviour (**3AB: Mean = 68.7%, SD = 22.4; 4AB: Mean = 71.7%, SD = 22.2**). Crucially, subject-level scores were highly correlated across task versions, with r = 0.81, p < .001 for win-stay and r = 0.70, p < .001 for lose-shift. These findings further support the validity of the 3AB task as a consistent and reliable tool for capturing reinforcement learning behaviour.

**Figure 3 Group-level mean win-stay and lose-shift percentages are shown for each task version among participants who completed both tasks (N = 111).** Error bars represent standard error of the mean. Strategy use was highly similar across tasks. Win-stay behaviour averaged 82.0% (SD = 27.5) for the 3AB task and 81.0% (SD = 26.6) for the 4AB task; lose-shift behaviour averaged 68.7% (SD = 22.4) for 3AB and 71.7% (SD = 22.2) for 4AB. Individual-level behaviour was strongly correlated across tasks (win-stay: r = 0.81, p < .001; lose-shift: r = 0.70, p < .001), indicating consistent application of learning strategies across the two task structures. Error bars represent the standard error of the mean.

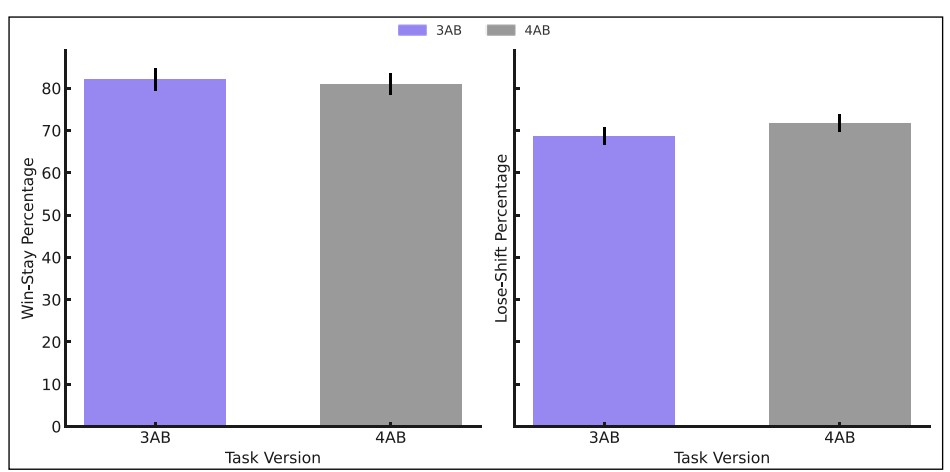

## 2.2 PRE-SCREENING RESULTS

Following the validation of the 3AB task, 1000 participants (mean age = 38, SD = 11, female = 54%) were pre-screened using a combination of the Snaith-Hamilton Pleasure Scale (SHAPS), the Dimensional Anhedonia Rating Scale (DARS), the Generalized Anxiety Disorder 7-item (GAD-7), and the Zung Self-Rating Depression Scale (ZUNG). These mood questionnaires provided a multi-faceted view of participants' emotional and cognitive states, allowing us to distinguish between individuals with high and low hedonic capacity.

Figure 4 presents scatter plots that illustrate the relationships between these different symptom measures in N = 935 individuals (65 participants were excluded on the basis of failing the attention check items in the four questionnaires). Specifically: DARS vs. SHAPS showed a strong negative correlation (r = –0.60), confirming that both scales effectively measure anhedonia, albeit in opposite directions (with higher scores on the SHAPS reflecting higher anhedonia and higher scores on the DARS reflecting lower anhedonia).

DARS vs. GAD showed a moderate negative correlation (r = –0.39), indicating that higher anxiety levels were associated with greater anhedonia. GAD vs. SHAPS showed a moderate positive correlation (r = 0.51), indicating that individuals with higher anxiety also reported greater anhedonia. ZUNG showed expected associations with the other symptom measures, correlating negatively with DARS (r = –0.57) and positively with SHAPS (r = 0.58) and GAD (r = 0.77). Together, these correlations indicate a coherent symptom structure in the pre-screening cohort, with anhedonia measures relating to depression and anxiety in the expected directions.

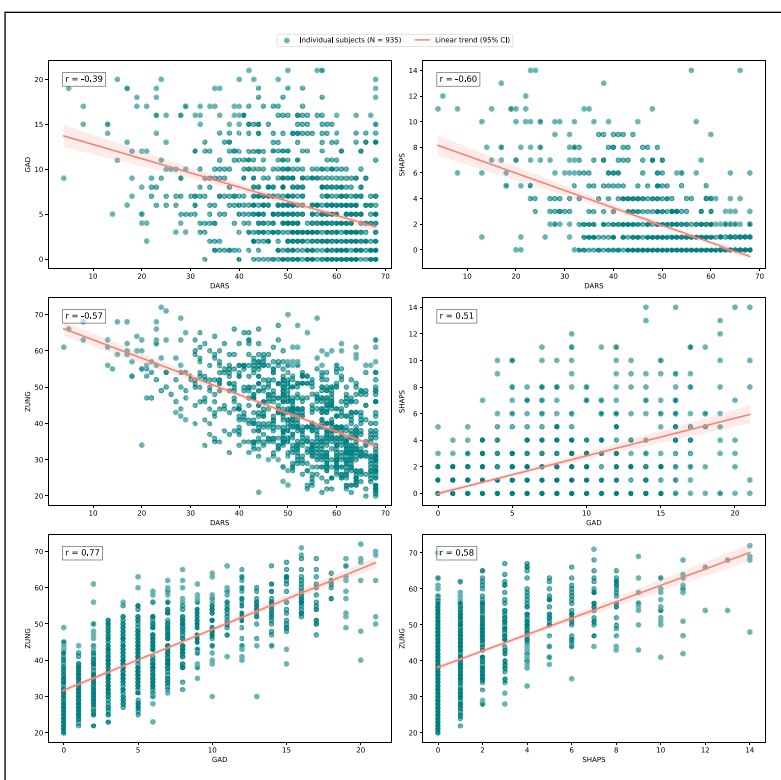

**Figure 4 Scatter plots illustrating the relationships between scores on the Dimensional Anhedonia Rating Scale (DARS), Generalized Anxiety Disorder Assessment 7-item (GAD-7), Snaith-Hamilton Pleasure Scale (SHAPS), and Zung Self- Rating Depression Scale (ZUNG).** Each subplot presents the Pearson correlation coefficient (r) for the respective pairwise comparisons. The data points are represented as teal circular markers, with a salmon-coloured trend line indicating the linear relationship between variables. The sample size (N = 935) is consistent across all plots, reflecting complete cases across all four questionnaire totals. These relationships highlight the varying degrees of association between measures of anhedonia, anxiety, and depression, with strong correlations observed between DARS and SHAPS, and moderate correlations between GAD and SHAPS.

Self-report measures from the final experimental sample (N = 206) i.e. subjects who met the pre-defined criteria for anhedonic (N = 111) and non-anhedonic (N = 95) classification (SHAPS > 2 and DARS < 45 vs. SHAPS = 0 and DARS > 55) are shown in Figure 5 for clarity.

As expected, the two anhedonia scales were strongly negatively correlated (DARS–SHAPS: r = –0.86), supporting construct validity. DARS also showed a moderate negative correlation with anxiety (GAD: r = –0.58). ZUNG showed expected associations with anhedonia and anxiety measures, correlating negatively with DARS (r = –0.75) and positively with SHAPS (r = 0.73) and GAD (r = 0.81). These relationships support the interpretability of subsequent analyses and indicate that symptom measures in the task-performing sample relate in the expected directions.

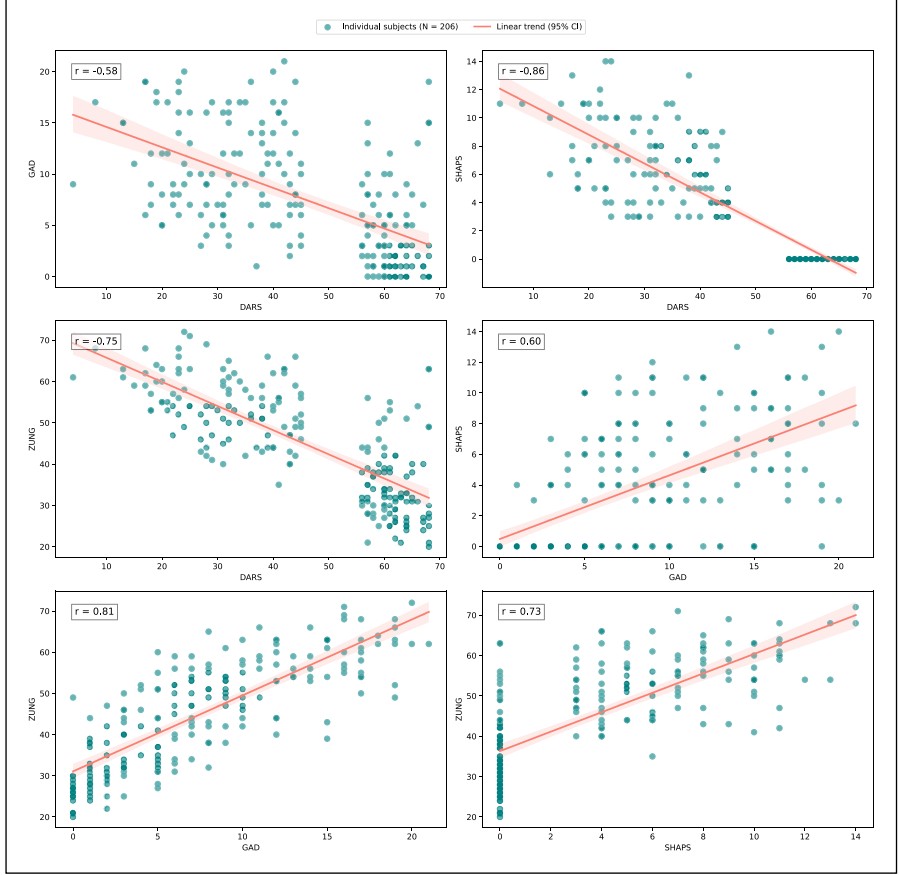

Ramaswamy et al.          **63**
*Computational Psychiatry*

**Figure 5 Pairwise Pearson correlations between questionnaire scores in the final task sample (N = 206).** Scatter plots show individual subject data and Pearson correlation coefficients (r). Panels use pairwise-complete observations. DARS = Dimensional Anhedonia Rating Scale; SHAPS = Snaith-Hamilton Pleasure Scale; GAD = Generalized Anxiety Disorder scale; ZUNG = Zung Depression Scale.

## 2.3 MODEL-BASED ANALYSIS OF LEARNING RATES AND SENSITIVITY

To investigate how anhedonia affects learning from rewards and punishments, we applied hierarchical Bayesian modelling to the data from the 3AB task, estimating individual learning rates and sensitivity to rewards and punishments. Parameter recovery (Figure 6) analysis confirmed that the model successfully captured key cognitive parameters for both anhedonic and non-anhedonic groups, with high correlations between original and simulated values: reward learning rate (Arew) r = 0.79–0.85, punishment learning rate (Apun) r = 0.67–0.74, reward sensitivity (R) r = 0.96–0.97, and punishment sensitivity (P) r = 0.82–0.92. This high level of recovery suggests the model's robustness in reproducing observed data and capturing individual differences.

**Figure 6 Parameter Recovery for Learning Rates and Sensitivity.** The scatter plot displays the parameter recovery results for reward and punishment learning rates, as well as for reward and punishment sensitivity, across anhedonic and non-anhedonic groups. Each scatter plot compares the original parameters with the simulated parameters, with an accompanying trend line and Pearson correlation coefficient (r). Arew: r = 0.79 (anhedonic), r = 0.85 (non-anhedonic), Apun: r = 0.67 (anhedonic), r = 0.74 (non-anhedonic), R: r = 0.96 (anhedonic), r = 0.97 (non-anhedonic), P: r = 0.82 (anhedonic), r = 0.92 (non-anhedonic).

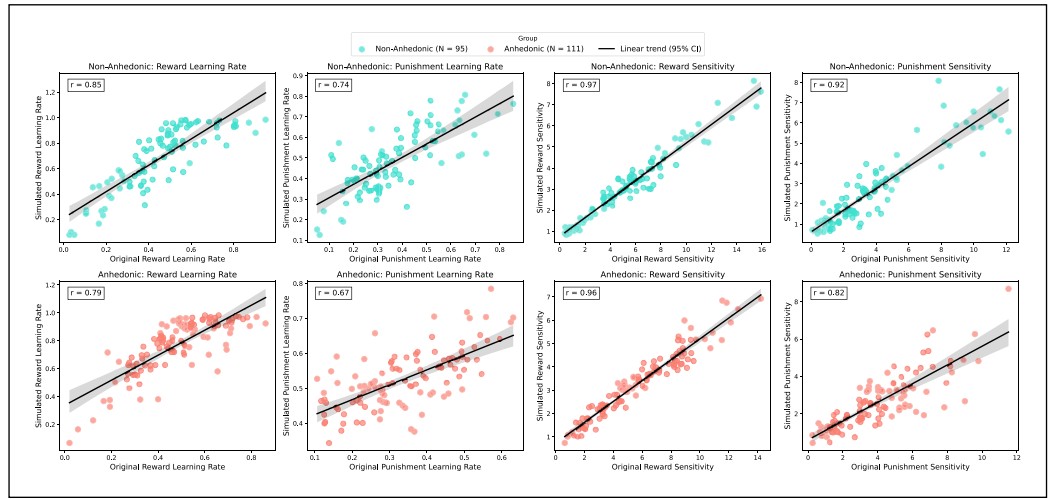

Ramaswamy et al.
*Computational Psychiatry*

## 2.4 COMPARISON OF LEARNING RATES AND SENSITIVITIES ACROSS ANHEDONIC AND NON-ANHEDONIC GROUPS

The anhedonic group exhibited numerically higher learning rates for reward (Arew: M = 0.48, SD = 0.18) but slightly lower learning rates for punishment (Apun: M = 0.34, SD = 0.14) compared to the non-anhedonic group (Arew: M = 0.45, SD = 0.20; Apun: M = 0.36, SD = 0.16). However, these differences were not statistically significant for reward (t = 1.19, p = 0.24) or punishment learning rates (t = –0.72, p = 0.48). Similarly, while the anhedonic group displayed numerically greater sensitivity to both rewards (R: M = 5.89, SD = 3.13) and punishments (P: M = 4.07, SD = 2.48) compared to the non-anhedonic group (R: M =5.53, SD = 3.45; P: M = 3.90, SD = 3.02), these differences were also not statistically significant (reward sensitivity: t = 0.77, p = 0.44; punishment sensitivity: t = 0.45, p = 0.65). Overall, the findings suggest comparable learning rates and sensitivity to rewards and punishments between the two groups, suggesting that reinforcement learning may not be impaired in anhedonia. For each model parameter, Bayesian t-tests indicated moderate-to-strong evidence in favour of the null hypothesis (BF01), suggesting no meaningful differences between the anhedonic and non-anhedonic groups. Specifically, the BF01 values were 3.36 for Reward Learning Rate, 5.14 for Punishment Learning Rate, 4.97 for Reward Sensitivity, and 5.96 for Punishment Sensitivity. These results (Table 1) provide quantitative support for the null hypothesis, reinforcing the interpretation of no significant group differences in these parameters.

| MODEL PARAMETER | GROUP | MEAN | SD | t-STATISTIC | P- VALUE | DEGREES OF FREEDOM (df) | BF01 (EVIDENCE FOR NULL) |
|---|---|---|---|---|---|---|---|
| **Arew** | Anhedonic | 0.48 | 0.18 | 1.19 | 0.24 | 192.0 | 3.36 |
| | Non-Anhedonic | 0.45 | 0.20 | | | | |
| **Apun** | Anhedonic | 0.34 | 0.14 | –0.72 | 0.48 | 186.7 | 5.14 |
| | Non-Anhedonic | 0.36 | 0.16 | | | | |
| **R** | Anhedonic | 5.89 | 3.13 | 0.77 | 0.44 | 191.9 | 4.97 |
| | Non-Anhedonic | 5.53 | 3.45 | | | | |
| **P** | Anhedonic | 4.07 | 2.48 | 0.45 | 0.65 | 181.9 | 5.96 |
| | Non-Anhedonic | 3.90 | 3.02 | | | | |

**Table 1** Reward and Punishment Learning Parameters by Anhedonia Status.

To further assess group-level differences in reinforcement learning parameters, we conducted a Bayesian comparison of posterior distributions for each group-level parameter using 95% Highest Density Intervals (HDIs). Figure 7 visualizes the posterior distributions for each parameter (reward/punishment learning rate and sensitivity) in the Anhedonic and Non-Anhedonic groups. Group differences were defined as Non-Anhedonic minus Anhedonic, allowing us to interpret both the direction and uncertainty of effects. Across all parameters, the HDIs for the difference scores included zero, and posterior probability (pd) values ranged from 0.725 to 0.891, indicating insufficient evidence for reliable group-level differences. These findings align with the Bayes Factor analyses, which also provided moderate to strong support for the null hypothesis. Together, these results suggest that core reinforcement learning processes—including how participants update and weight reward and punishment information—are not meaningfully altered in individuals with anhedonia.

As an additional robustness check, we repeated the group comparison using lapse-augmented models. Specifically, we fit a lapse version of the four-parameter model (banditNarm_lapse; separate reward and punishment learning rates and sensitivities plus a lapse parameter) and a lapse model with a single shared learning rate (banditNarm_singleA_lapse; shared learning rate with separate reward and punishment sensitivities plus a lapse parameter). In both cases, posterior densities overlapped strongly between groups and the 95% HDIs for group differences included zero across learning, sensitivity, and lapse parameters (pd values all < 0.86), indicating that the null group findings were robust to inclusion of lapse terms. Full posterior plots and HDI summaries are provided in Supplementary (Supplementary S2; Figures S4–S5).

Ramaswamy et al.     **65**
*Computational Psychiatry*

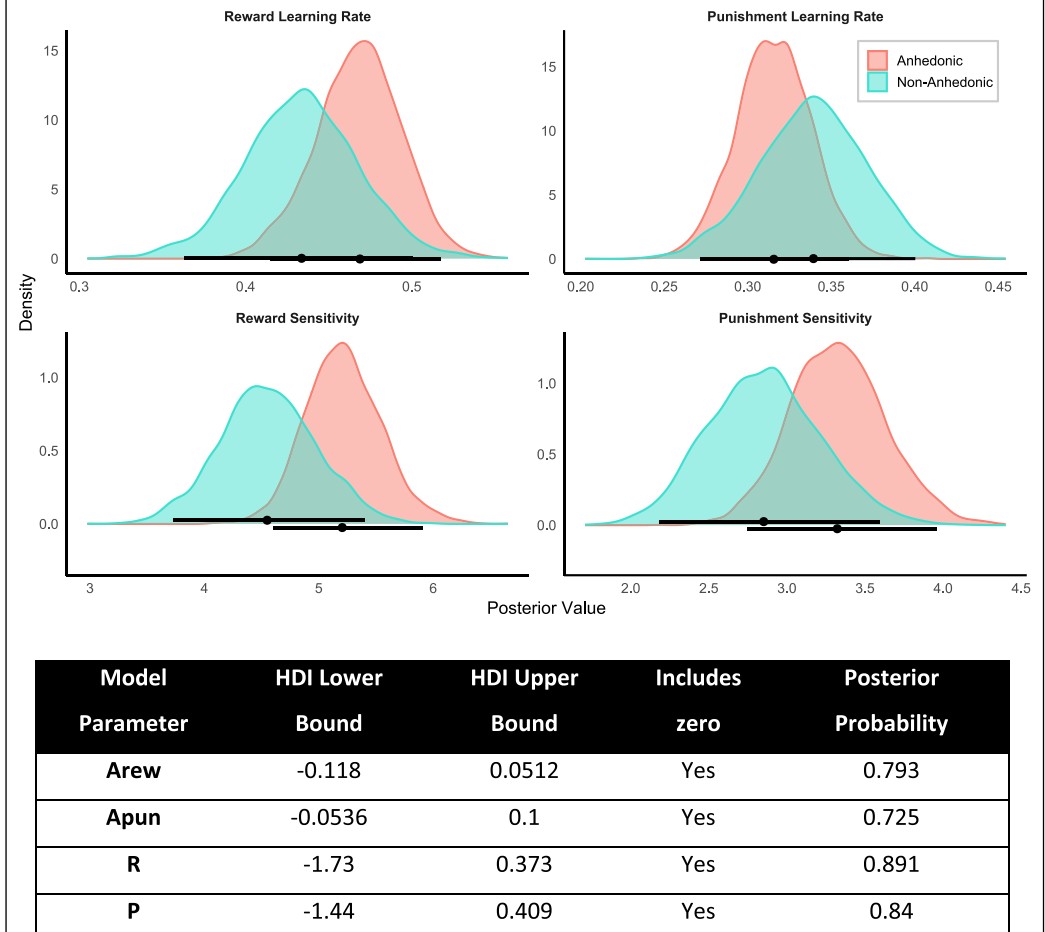

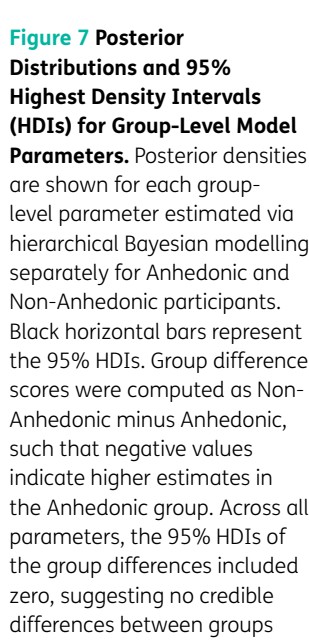

**Figure 7 Posterior Distributions and 95% Highest Density Intervals (HDIs) for Group-Level Model Parameters.** Posterior densities are shown for each group-level parameter estimated via hierarchical Bayesian modelling, separately for Anhedonic and Non-Anhedonic participants. Black horizontal bars represent the 95% HDIs. Group difference scores were computed as Non-Anhedonic minus Anhedonic, such that negative values indicate higher estimates in the Anhedonic group. Across all parameters, the 95% HDIs of the group differences included zero, suggesting no credible differences between groups in learning rate or sensitivity parameters.

| Model Parameter | HDI Lower Bound | HDI Upper Bound | Includes zero | Posterior Probability |
|---|---|---|---|---|
| Arew | -0.118 | 0.0512 | Yes | 0.793 |
| Apun | -0.0536 | 0.1 | Yes | 0.725 |
| R | -1.73 | 0.373 | Yes | 0.891 |
| P | -1.44 | 0.409 | Yes | 0.84 |

## 2.5 WIN-STAY AND LOSE-SHIFT STRATEGY ANALYSIS

In addition to learning rates, we also examined decision-making strategies using the model-agnostic "win-stay" and "lose-shift" metrics. The win-stay strategy refers to repeating a choice after receiving a reward, while lose-shift involves changing the choice after a punishment. These strategies provide insights into how individuals adjust their behaviour based on very recent outcomes.

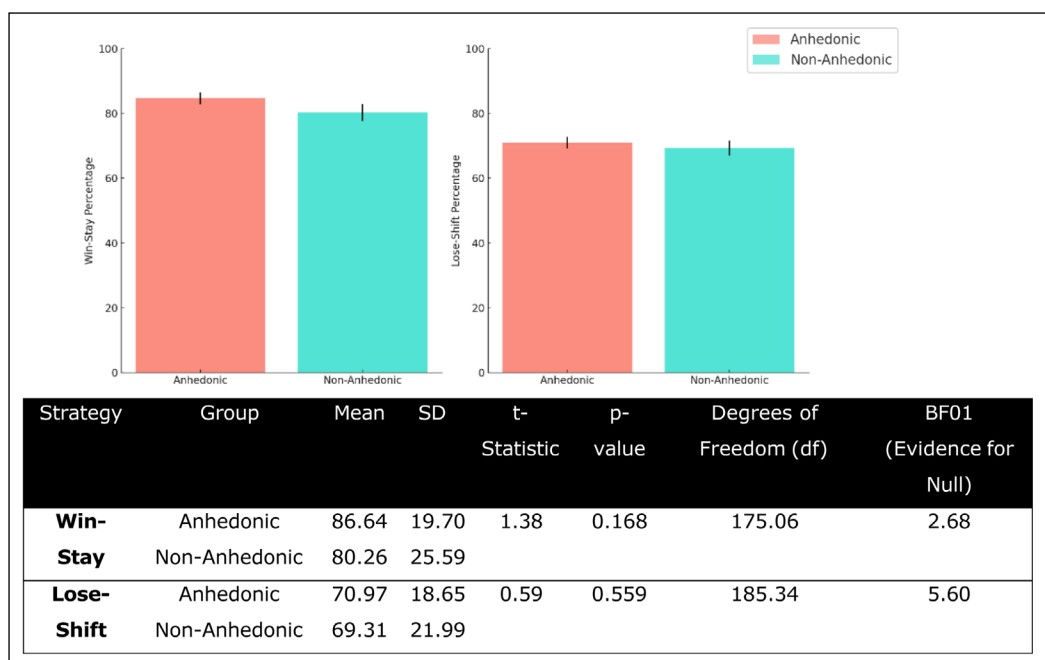

**Figure 8 Win-Stay and Lose-Shift Strategies Between Anhedonic and Non-Anhedonic Groups.** Bar plots represent the average win-stay and lose-shift strategy adoption percentages for the anhedonic and non-anhedonic groups. Error bars represent the standard error of the mean.

| Strategy | Group | Mean | SD | t-Statistic | p-value | Degrees of Freedom (df) | BF01 (Evidence for Null) |
|---|---|---|---|---|---|---|---|
| Win-Stay | Anhedonic | 86.64 | 19.70 | 1.38 | 0.168 | 175.06 | 2.68 |
| | Non-Anhedonic | 80.26 | 25.59 | | | | |
| Lose-Shift | Anhedonic | 70.97 | 18.65 | 0.59 | 0.559 | 185.34 | 5.60 |
| | Non-Anhedonic | 69.31 | 21.99 | | | | |

For the win-stay strategy, the anhedonic group (M = 86.64, SD = 19.70) and the non-anhedonic group (M = 80.26, SD = 25.59) showed no statistically significant difference, t = 1.38, p = 0.168. Similarly, for the lose-shift strategy, the anhedonic group (M = 70.97, SD = 18.65) and the non-anhedonic group (M = 69.31, SD = 21.99) did not differ significantly, t = 0.59, p = 0.559 (Figure 8). For the win-stay strategy, the Bayes Factor (BF01) was 2.68, indicating moderate evidence in support of the null hypothesis. Similarly, for the lose-shift strategy, the BF01 was 5.60 suggesting strong evidence in favour of the null. These Bayesian results strengthen the conclusion that there are no meaningful differences in win-stay or lose-shift strategies between anhedonic and non-anhedonic groups.

Ramaswamy et al. **66**
*Computational Psychiatry*

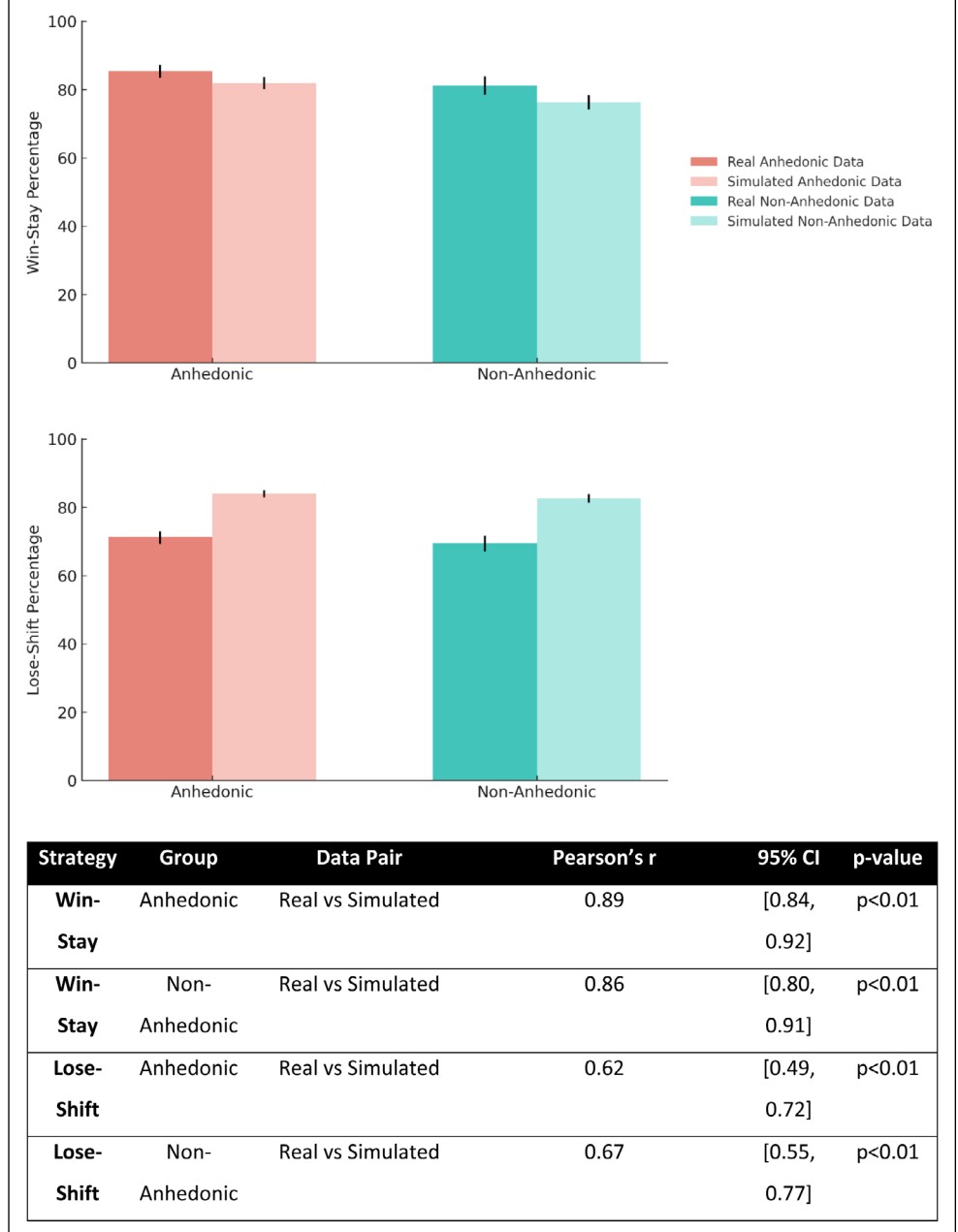

| Strategy | Group | Data Pair | Pearson's r | 95% CI | p-value |
|---|---|---|---|---|---|
| Win-Stay | Anhedonic | Real vs Simulated | 0.89 | [0.84, 0.92] | p<0.01 |
| Win-Stay | Non-Anhedonic | Real vs Simulated | 0.86 | [0.80, 0.91] | p<0.01 |
| Lose-Shift | Anhedonic | Real vs Simulated | 0.62 | [0.49, 0.72] | p<0.01 |
| Lose-Shift | Non-Anhedonic | Real vs Simulated | 0.67 | [0.55, 0.77] | p<0.01 |

**Figure 9 Win-stay and lose-shift strategy rates in real vs simulated data across groups.** Bars show group means for Anhedonic and Non-Anhedonic participants with separate bars for Real (darker) and Simulated (lighter) data; error bars indicate standard error of the mean (SE). Simulations slightly underestimated win-stay (Δ (Sim−Real) ≈ −3.5 to −4.9 percentage points) and over-estimated lose-shift (Δ ≈ +12.8 to +13.2 points); paired within-group comparisons were significant (see Results). Subject-wise Real–Sim correlations were high for win-stay and moderate for lose-shift, indicating preservation of individual-difference structure.

We applied the same model-agnostic win-stay and lose-shift analysis to simulated data generated from each participant's fitted parameters (Figure 9). Within each group, we compared simulated and real strategy rates using paired-sample t-tests. For win-stay, the anhedonic group showed 81.88% (SD = 19.02) in simulations vs 85.39% (SD = 19.88) in real data (Δ (Sim−Real) = −3.51 percentage points, t(110) = −3.97, p < .001, dz = −0.38, 95% CI [−5.26, −1.76]); the non-anhedonic group showed 76.25% (SD = 19.84) vs 81.12% (SD = 25.90) (Δ = −4.87 percentage points, t(94) = −3.55, p = .001, dz = −0.36, 95% CI [−7.59, −2.15]). For lose-shift, the anhedonic group showed 83.99% (SD = 10.66) in simulations vs 71.15% (SD = 18.65) in real data (Δ = +12.84 percentage points, t(110) = 9.25, p < .001, dz = 0.88, 95% CI [+10.09, +15.60]); the non-anhedonic group showed

Ramaswamy et al.                    **67**
*Computational Psychiatry*

82.57% (SD = 11.64) vs 69.42% (SD = 22.03) (Δ = +13.16 percentage points, t(94) = 7.72, p < .001, dz = 0.79, 95% CI [+9.77, +16.54]). Despite these mean-level calibration differences (simulations slightly underestimating win-stay and overestimating lose-shift), individual differences were preserved: Real–Sim correlations were high for win-stay (anhedonic: r = 0.89, 95% CI [0.84, 0.92]; non-anhedonic: r = 0.86, 95% CI [0.80, 0.91]) and moderate for lose-shift (anhedonic: r = 0.62, 95% CI [0.49, 0.72]; non-anhedonic: r = 0.67, 95% CI [0.55, 0.77]). These checks indicate the generator reproduces rank-order structure while exhibiting small, systematic mean-level biases.

## 2.6 PREDICTED ACTION PROBABILITIES AND ACTUAL CHOICES

We compared the model's predicted action probabilities to the subjects' actual choices on a trial-by-trial basis. Figure 10 illustrates the predicted action probabilities for three representative subjects across the 200 trials. The figure provides a visual comparison between model predictions and actual subject behaviour, highlighting trial-by-trial consistency between the two. This alignment between predicted probabilities and observed choices supports the model's ability to capture the underlying decision-making processes during the task.

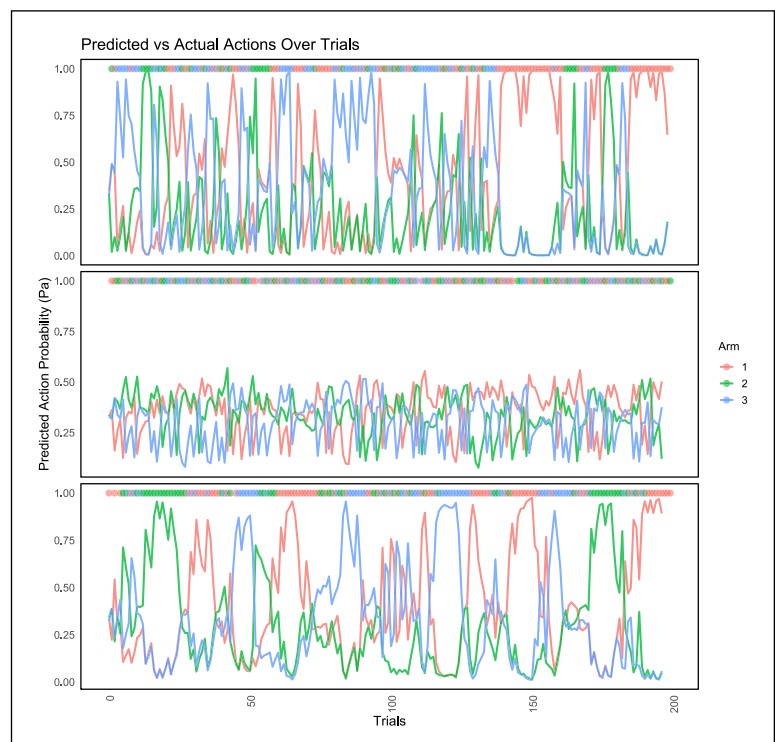

**Figure 10 Predicted Action Probabilities and Actual Choices Across Trials.** Choices and modelled action probabilities for three representative subjects. Solid markers indicate actual choices made by the subjects on each trial (at y = 1), colour-coded by arm. Lines show trial-by-trial predicted probabilities generated by the hierarchical Bayesian model, with the same colour-coding.

To quantitatively assess the model's predictive validity at the group level, we computed the proportion of choices accurately predicted by the model for each participant. For each trial, we identified the option with the highest predicted action probability and compared it to the participant's actual choice. The mean prediction accuracy across the sample was **59.60%** (*SD* = 16.73%; *N* = 206), substantially above chance level (33%). Figure 11 shows the distribution of prediction accuracy across subjects. This analysis provides additional evidence that the model reliably captured participants' choice behaviour at an individual level, beyond the visual inspection of exemplar cases.

We next compared mean reaction times (RTs) between anhedonic and non-anhedonic participants. Using mean RT per participant across all trials, non-anhedonic individuals responded more slowly on average than anhedonic individuals (M_Anhedonic = 465.53 ms, SD = 135.33; M_Non-anhedonic = 539.62 ms, SD = 148.72; t(192.03) = –3.71, p = 2.7 × 10⁻⁴). To test whether this effect was driven by the early high-variance trials visible in the trial-wise RT plot, we repeated the analysis excluding the first 10 trials. The group difference remained very similar (trials 11–200: M_Anhedonic = 453.13 ms, SD = 136.78; M_Non-anhedonic = 524.48 ms, SD = 150.91; t(191.68) = –3.53, p = 5.2 × 10⁻⁴), and comparable results were obtained when excluding the first 15 trials (trials 16–200: t(192.19) = –3.50, p = 5.8 × 10⁻⁴). See **Supplementary S1** for details.

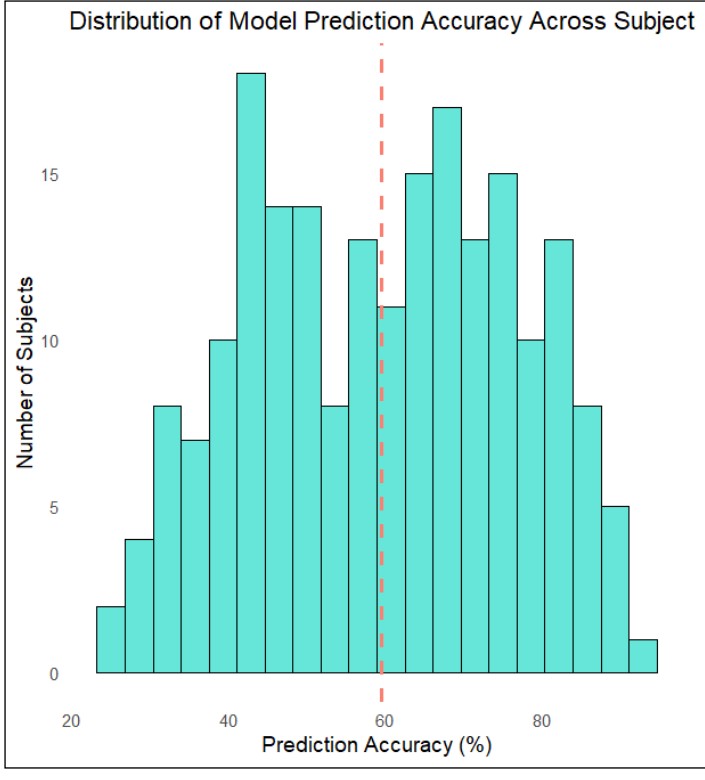

Distribution of Model Prediction Accuracy Across Subject

Ramaswamy et al.          **68**
*Computational Psychiatry*

**Figure 11 Distribution of Model Prediction Accuracy Across Subjects.** Histogram showing the distribution of model-predicted choice accuracy for each subject (N = 206). Accuracy was computed as the proportion of trials where the model's highest predicted action probability (Pa) matched the participant's actual choice. The vertical dashed line indicates the group mean accuracy (59.60%).

## 2.7 CORRELATION BETWEEN MODEL PARAMETERS AND SELF-REPORTED ANHEDONIA MEASURES

Finally, we explored the relationship between self-reported anhedonia, as measured by the Dimensional Anhedonia Rating Scale (DARS) and Snaith-Hamilton Pleasure Scale (SHAPS), and the computational model parameters.

As shown in Figure 12, comparisons between the anhedonic and non- anhedonic groups reveal no significant differences in Reward Learning Rate and Punishment Learning Rate. Given our extreme groups design, where the non-anhedonic group includes participants with minimal or zero scores on SHAPS by design, group comparisons using t-tests (Table 1) are more appropriate than correlation analyses. These tests consistently show no significant differences in Reward and Punishment Learning Rates across anhedonia levels, suggesting that participants' ability to learn from rewards and punishments is not systematically associated with self- reported levels of anhedonia.

**Figure 12 Group Comparisons of Model Parameters and Self-Reported Anhedonia Measures (DARS and SHAPS).** This figure illustrates the relationship between self-reported anhedonia and computational model parameters of reward and punishment learning. Scatter plots display the distribution of scores for the two questionnaire measures— DARS (daily activity and reward engagement) and SHAPS (hedonic capacity)—in relation to four computational model parameters: Reward Learning Rate, Punishment Learning Rate, Reward Sensitivity, and Punishment Sensitivity. Each point is colour-coded by group (anhedonic or non-anhedonic). Although correlation values are presented, these should be interpreted with caution due to the extreme groups design, which limits variability in one group and affects the generalizability of linear relationships. The figure primarily highlights that there is no strong link between subjective anhedonia group status and performance-based measures of reward and punishment processing.

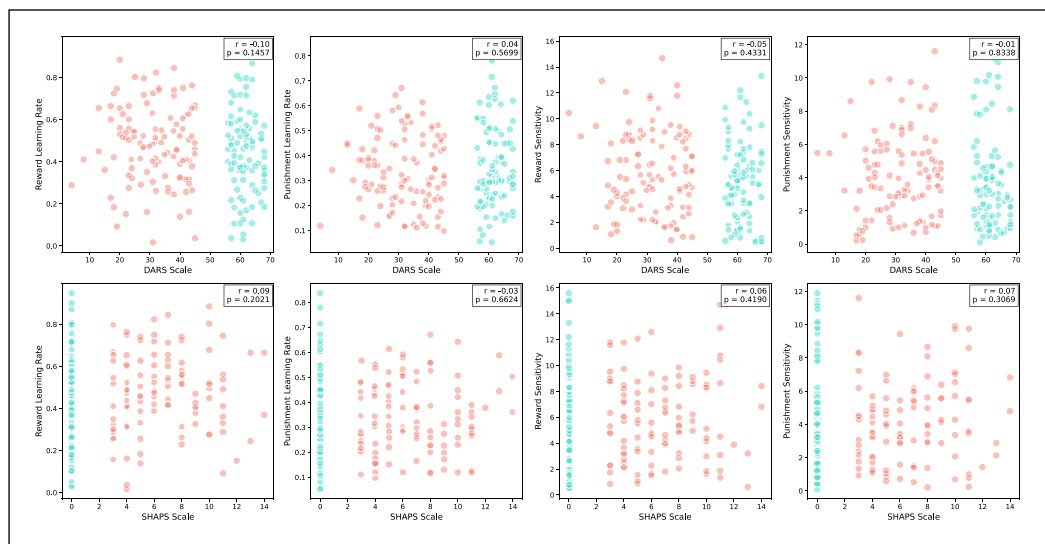

*Computational Psychiatry*

Additionally, to examine the relationship between specific facets of anhedonia and computational parameters, we conducted exploratory correlations between the DARS subscales (Hobbies, Food/Drink, Social Interaction, and Sensory Experiences) and model parameters (Reward Learning Rate, Punishment Learning Rate, Reward Sensitivity, and Punishment Sensitivity). The results are summarized in Table 2. While uncorrected correlation coefficients ranged from –0.12 to 0.09, none of the associations survived correction for multiple comparisons (Bonferroni- corrected $p > 0.05$, Table 2). These findings reinforce the absence of strong or significant relationships between specific facets of anhedonia and computational parameters.

| DARS SUBSCALE | PARAMETER | CORRELATION | P-VALUE |
|---|---|---|---|
| **Hobbies** | Arew | –0.06 | 0.40 |
| | Apun | 0.06 | 0.39 |
| | R | –0.09 | 0.22 |
| | P | 0.01 | 0.92 |
| **Food/Drink** | Arew | –0.09 | 0.19 |
| | Apun | 0.04 | 0.61 |
| | R | –0.07 | 0.29 |
| | P | –0.02 | 0.80 |
| **Social Interaction** | Arew | –0.08 | 0.27 |
| | Apun | 0.09 | 0.19 |
| | R | –0.05 | 0.46 |
| | P | –0.03 | 0.65 |
| **Sensory Experiences** | Arew | –0.12 | 0.09 |
| | Apun | –0.03 | 0.62 |
| | R | 0.00 | 0.98 |
| | P | –0.01 | 0.92 |

**Table 2** Exploratory Correlations Between DARS Subscale Scores and Computational Model Parameters.

Consistent with the findings from the SHAPS-DARS total scores, the computational measures derived from reward and punishment tasks may not strongly capture individual differences in anhedonia, even when examining more granular subscale scores. This suggests that self-reported anhedonia and task-based computational parameters may assess distinct facets of reward and punishment processing or that the influence of anhedonia on these parameters is minimal. Further investigations with larger samples and alternative designs may help clarify these relationships.

# 3 DISCUSSION

## 3.1 SUMMARY OF KEY RESULTS

The current study found no significant differences between anhedonic and non-anhedonic groups relating to any of the computational parameters assessed. Effect sizes for these comparisons were small (ranging from $d = 0.06–0.16$), supporting the interpretation of no meaningful group-level distinctions. Additionally, model-agnostic analyses of win-stay and lose-shift strategies showed no significant group differences. Bayes Factor analyses further provided moderate-to-strong support for the null hypothesis for each parameter in the group comparisons. These results suggest that, in contrast to our hypothesis, cognitive learning processes related to reward and punishment appear largely similar across levels of anhedonia.

We also observed that non-anhedonic participants had modestly slower reaction times than anhedonic participants, an effect that persisted when early trials were excluded; because this does not map directly onto a specific reinforcement-learning hypothesis and may reflect differences in caution, speed–accuracy trade-off, or task engagement rather than anhedonia per se, we treat this RT difference as exploratory and do not base our main conclusions on it.

## 3.2 IMPLICATIONS FOR ANHEDONIA AND COGNITIVE LEARNING

Our findings suggest that individuals with anhedonia are not fundamentally impaired in their ability to learn from rewards and punishments, as they performed comparably to non-anhedonic individuals on reinforcement learning tasks. This result challenges the straightforward hypothesis that anhedonia is directly caused by disruptions in reinforcement learning processes. However, an alternative explanation is that the 3AB task—and by extension, other simple RL paradigms—may not be sufficiently sensitive to detect subtle impairments in reward processing associated with anhedonia. Instead, it indicates that while the core cognitive mechanisms for reinforcement learning remain intact, the deficits associated with anhedonia may stem from how reward- and punishment-related signals are experienced, valued, or integrated into behaviour.

This perspective refines our understanding of anhedonia, suggesting that its origins may lie in impairments in the emotional or motivational aspects of reinforcement learning rather than a breakdown of the cognitive learning mechanisms themselves. These findings underscore the importance of distinguishing between the computational processes that underlie learning and the affective processes that shape the subjective experience of reward and punishment. Future research should investigate how disruptions in these interactions contribute to anhedonia, using multi-modal approaches to bridge the gap between behavioural performance and the underlying emotional and neurobiological mechanisms.

## 3.3 CONTEXTUALIZING FINDINGS WITHIN EXISTING LITERATURE

Reinforcement learning paradigms have been central in research into reward processing abnormalities in psychiatric conditions, such as depression and anhedonia. Unlike static reward tasks, such as the Probabilistic Reward Task (PRT), which emphasize reward bias, in dynamic bandit paradigms there is a need for continuous adaptation to changing contingencies. The present findings suggest that core reinforcement learning mechanisms are preserved in anhedonia and, as such, argue against a broad impairment in reward and punishment learning in anhedonia. Specifically, meta-analyses of reward processing in depression and mood disorders have indicated that deficits are replicable and include both reward and punishment learning deficits (Pike & Robinson, 2022; Halahakoon et al., 2020). In this regard, Halahakoon et al. (2020) have estimated that depression was associated with small to medium impairments in reinforcement learning with standardized mean difference (SMDs) of 0.352, option valuation with SMDs of 0.309, and the greatest impairment in reward bias, with an SMD of 0.644.

Meanwhile, Pike & Robinson (2022) investigated only computational reinforcement learning models and found that for SMD = 0.107, punishment learning was higher, while the reward learning rates were somewhat lower at SMD = –0.021 among individuals with mood and anxiety disorders. However, the results also indicated significant heterogeneity among studies, which may suggest that the size of the effects could be influenced by task design, parameter estimation, and disorder severity.

These meta-analyses encompass reinforcement learning tasks across probabilistic choice paradigms and bandit tasks, among others, but are not confined to measures of reward bias alone, such as the PRT. Our findings extend these observations by showing intact learning mechanisms in anhedonia even in the context of a dynamic decision-making task. This suggests that anhedonia-related impairments may be more pronounced in tasks requiring effort-based decision-making or explicit reward valuation rather than trial-by-trial reinforcement learning efficiency (Treadway et al., 2009).

Beyond meta-analyses, individual studies provide further context. Harlé et al. (2017) found that higher anhedonia was associated with reduced reward-driven choice consistency in a 2-arm bandit task, particularly among participants whose decisions were best explained by a softmax function. This reduction in reward-guided behaviour was indexed by lower inverse temperature values in their model, which reflect choice stochasticity rather than reward valuation per se. In our study, we modelled reward sensitivity (R) as a distinct parameter from decision noise, allowing us to

disentangle valuation processes from choice consistency. However, their study also showed that learning rates remained intact, which agrees with our findings. Likewise, Huys et al. (2013) showed that anhedonia reduces reward sensitivity but does not affect learning rates, further reinforcing the idea that reward valuation, rather than reinforcement learning, may be impaired in anhedonia.

Punishment learning has been suggested as one such separate reinforcement learning mechanism that is influenced by mood disorders. Aylward et al. (2019) reported that in individuals with mood and anxiety symptoms, increased punishment learning rates, indicate that negative feedback may contribute to learning more robustly in these populations. However, our study did not indicate increased punishment sensitivity in anhedonia, further supporting the notion that the reinforcement learning profiles of state anhedonia and mood/anxiety disorders are different. This distinction underlines the importance of task design, as some psychiatric populations may show heightened punishment learning, while anhedonia is perhaps better characterized by impairments in reward valuation.

Halahakoon et al. (2020) and Kieslich et al. (2022) stressed the importance of distinguishing cognitive, emotional, and motivational components of reinforcement learning. Our findings support this perspective: belief updating and reinforcement-based learning mechanisms appear intact in anhedonia, but altered motivation and subjective reward valuation may still influence behaviour. According to Kieslich et al. (2022), impairments in reward processing associated with anhedonia may be more strongly linked to reduced reward valuation and anticipation, and less consistently linked to impairments in reinforcement learning efficiency. This could explain why tasks focused on reinforcement learning, but not reward anticipation or valuation, might not capture expected group differences.

In showing that anhedonic individuals do not exhibit learning deficits on this task, our findings indicate that basic reinforcement learning capabilities are intact. Instead, motivational and affective reward processing dimensions might be more central to an understanding of the impact of anhedonia on decision-making.

## 3.4 CONTRIBUTIONS TO COMPUTATIONAL PSYCHIATRY

This study contributes to computational psychiatry by validating a 3AB task against the traditional 4AB, demonstrating its utility in capturing key reward and punishment learning mechanisms while reducing cognitive load. Notably, this is the first study to use both the Snaith-Hamilton Pleasure Scale (SHAPS) and the Dimensional Anhedonia Rating Scale (DARS) alongside a reinforcement learning task. The inclusion of both scales allowed for a more comprehensive exploration of anhedonia, encompassing not only general hedonic capacity but also domain-specific aspects of reward engagement across daily activities. Exploratory correlations between DARS subscales and computational model parameters provided initial insights into these domain-specific dimensions, although no associations survived correction for multiple comparisons. The task's high parameter recovery, along with consistent results from hierarchical Bayesian modelling, supports its application in research focused on populations where cognitive load might impact performance, such as clinical settings. We also confirmed that mood and anhedonia measures in our task-performing sample showed expected directional relationships (Figure 5), suggesting that the sample's clinical heterogeneity did not introduce inconsistencies in symptom structure that would obscure reward-processing effects.

## 3.5 LIMITATIONS AND FUTURE DIRECTIONS

A sensitivity power analysis revealed that our study was adequately powered (80%) to detect medium-sized effects (d ≥ 0.39) but underpowered to reliably detect smaller effects. This indicates that while the study was well-suited to identify moderate group differences, it may have missed more subtle variations in reinforcement learning processes associated with anhedonia. Consequently, the absence of significant findings should be interpreted with caution, as smaller effects might not have been captured. Future studies should recruit larger sample sizes to improve sensitivity to small effects and refine task designs to explore subtle group differences in reward and punishment processing.

Additionally, the use of an online sample from Prolific, where mental health conditions may be overrepresented, presents both strengths and limitations. Prolific enables access to diverse populations, but the symptom severity and clinical profiles of participants may differ from those in formal clinical settings, where anhedonia is often more pronounced. Although we employed a SHAPS cut-off of 3 to identify clinically significant anhedonia, the self-selected nature of the sample may introduce biases. Moreover, we did not collect information about participants' mental health history or psychiatric medication use, limiting the clinical interpretability of the findings. While the use of validated self-report measures enabled robust group stratification based on anhedonia traits, future studies should incorporate structured clinical assessments, medication history, and neurobiological measures (e.g. neuroimaging or biomarkers) to improve generalizability and provide a more comprehensive understanding of how anhedonia affects reward processing. These approaches could ultimately inform the development of more targeted interventions for mood disorders. This mirrors prior work highlighting potential dissociations between self-report and behavioural measures in psychiatric research (Eisenberg et al., 2019; Enkavi et al., 2019), and may reflect the challenge of capturing trait anhedonia through performance-based metrics alone.

Third, relationships between self-report symptom scales in online samples can differ from those observed in clinical cohorts, particularly under extreme-groups sampling designs. In addition, SHAPS, DARS and Zung probe overlapping but non-identical constructs: SHAPS emphasises consummatory pleasure in everyday sensory and social situations (Snaith et al., 1995; Liu et al., 2012), DARS samples anticipatory, motivational and consummatory aspects across personalised domains (Rizvi et al., 2015; Wellan et al., 2021; Mittmann et al., 2025), and Zung focuses on global depressive symptomatology with only a subset of items directly indexing anhedonia (Zung, 1965; Romera et al., 2008). As a result, our anhedonic and non-anhedonic subgroups defined using SHAPS may not map cleanly onto clinically depressed versus non-depressed individuals. Our null findings regarding reward learning parameters should therefore be interpreted as applying to this particular online extreme-groups sample, rather than as definitive evidence about anhedonia in clinical populations.

Also, we used the SHAPS in its original binary form to enable classification based on established clinical cut-offs, aligning with our aim of testing whether the 3AB task is sensitive to clinically significant anhedonia. While scoring SHAPS on a continuous 1–4 scale can provide greater granularity in general population samples, our primary objective was to identify robust group-level differences that would be applicable to future clinical studies. Importantly, we also explored the relationship between task performance and continuous SHAPS scores across the full sample, and this analysis yielded results consistent with the group comparison approach—further supporting the conclusion that the task may not be sensitive to individual differences in anhedonia, regardless of scoring method.

Although we describe the 3AB task as reducing cognitive load relative to the original Seymour et al. (2012) paradigm, we did not include a formal metric of cognitive load in this study. Instead, our rationale was heuristic and based on established features known to influence working memory and attentional demands in reinforcement learning tasks (Collins & Frank, 2012). These included reducing the number of choices (3 vs. 4), reducing the number of outcome contingencies tracked (6 vs. 8), and increasing outcome frequency. Future work should consider directly assessing cognitive load to empirically confirm whether such design changes meaningfully reduce demands in both online and clinical populations. In the smaller validation sample, participants completed both the original 4-arm bandit and the 3-arm bandit in counterbalanced order. The number of participants per order condition was too small to robustly assess higher-order interactions between task order, group, and reaction times, and we therefore did not attempt to interpret order effects on RT in that subsample.

A further limitation concerns the punishment component of the model. In our parameter-recovery analyses, punishment learning rates were estimated with greater uncertainty than reward learning rates, and posterior predictive simulations captured empirical win–stay behaviour more closely than lose–shift tendencies. This pattern suggests that, in this three-arm implementation, punishment-driven behaviour is noisier and less well constrained by the current model, and that the task may be better suited to studying reward-related processes than to isolating avoidance or

punishment mechanisms in anhedonia. In addition, although lapse-augmented models modestly improved predictive fit, we found that fitting the dual learning rate + lapse model separately in the anhedonic and non-anhedonic groups produced strongly overlapping posteriors and 95% HDIs for group differences that included zero across reward learning rate, punishment learning rate, reward sensitivity, and punishment sensitivity, indicating that the null group findings were robust to inclusion of lapse terms. This robustness extended to both lapse-augmented model families that performed best in model comparison (banditNarm_lapse and banditNarm_singleA_lapse), with 95% HDIs for group differences including zero for learning, sensitivity, and lapse parameters in both cases (see Supplementary S2). Recent work has also highlighted that additional stochastic parameter such as lapse terms can compromise the interpretability of reinforcement-learning parameters unless accompanied by dedicated model- and parameter-recovery analyses (Wilson & Collins, 2019; Eckstein et al., 2021; Aylward et al., 2019). We therefore interpret anhedonia-related group effects on reward and punishment learning and sensitivity primarily within the more parsimonious non-lapse model, and regard the lapse variants as supporting evidence that some random responding is present in this online sample. Future studies, particularly in larger and less noisy in-person samples, could profitably adopt lapse-augmented models as the primary framework, combined with comprehensive simulation-based recovery work to more fully characterise any anhedonia-related differences.

## 4 CONCLUSION

This study contributes to our understanding of anhedonia by examining how reinforcement learning processes might underpin its development. Contrary to traditional models suggesting impaired reward sensitivity, our findings indicate that the core cognitive mechanisms underlying reward and punishment learning remain intact in individuals with anhedonia.

Bayesian analysis provided support for the null hypothesis (BF01 = 3.36–5.96), with no significant group differences in learning rates or reward sensitivity. In contrast, non-anhedonic participants responded more slowly on average than anhedonic participants; because this effect does not map directly onto our reinforcement-learning hypotheses and may reflect differences in caution, speed–accuracy trade-off, or task engagement, we treat it as exploratory and do not base our main conclusions on it (see Supplementary S1 for RT analyses). These results suggest that the impairments associated with anhedonia may not arise from fundamental disruptions in reinforcement learning but could reflect impairments in how reward-related information is valued or integrated.

The disconnect between self-reported anhedonia and computational parameters raises important questions about the relationship between subjective experiences and behavioural performance. This highlights the need for future research to explore how emotional and motivational processes interact with reinforcement learning mechanisms to contribute to anhedonia. Integrating multi-modal approaches, such as neuroimaging and biomarkers, could provide deeper insights into the mechanisms driving anhedonia and inform the development of targeted interventions for mood disorders.

## 5 METHODS

The data and code required to replicate the data analyses are both available online at (https://github.com/arjun-ramaswamy/3AB_Anhedonia_study).

### 5.1 PARTICIPANTS

We recruited participants from Prolific (www.prolific.co). Participants had to be aged 18–60, speak English as their first language, have no language-related disorders/literacy difficulties, have no visual impairments/have no mild cognitive impairment or dementia, and be resident in the UK. Participants were reimbursed at a rate of £6 per hour, and could earn a bonus of up to an additional £3 per hour based on their performance on the task. The study had ethical approval from the University College London Research Ethics Committee (15253/001).

Ramaswamy et al.                    **74**
*Computational Psychiatry*

To recruit 100 participants in the anhedonic range and 100 in the non-anhedonic range, we pre-screened a total of 1,000 participants. We classified participants as anhedonic if they scored >2 on the SHAPS and ≤45 on the DARS. This DARS threshold was determined based on 1 standard deviation (SD) below the mean, as reported in the original DARS validation study (Rizvi et al., 2015). Conversely, participants were classified as non-anhedonic if they scored 0 on the SHAPS and >55 on the DARS, indicating a higher hedonic capacity. This pre-screening aimed to ensure a clear distinction between anhedonic and non-anhedonic groups for the study. While dichotomization can reduce sensitivity to dimensional effects, we selected this extreme groups approach to maximize contrast between participants with clinically significant anhedonia and those with minimal symptoms. This strategy, guided by SHAPS and DARS thresholds, allowed for interpretable comparisons of core reinforcement learning processes across distinct symptom levels. Moreover, the inclusion of both SHAPS and DARS helped address potential psychometric limitations of SHAPS alone.

In a prior validation study, we tested the 3AB task against the 4AB task with 100 participants in each group. From these groups, only 15 participants scored within the anhedonic range based on a SHAPS cut-off score of >2 and a DARS cut-off score of ≤45. Given this low proportion of anhedonic participants in the general population, we pre-screened a larger sample of 1,000 participants to ensure we reached our target of 100 anhedonic participants for this study. A total of 1,000 participants completed the initial pre-screening phase (mean age = 38, SD = 11, 54% female), and 206 participants who met the anhedonic and non-anhedonic cutoffs were selected to complete the final study (mean age = 39, SD = 11, 61% female). Of these, 111 participants were classified as anhedonic and 95 as non-anhedonic (Demographic information breakdown in Table 3).

| METRIC | VALUE | COUNT | PERCENTAGE |
|---|---|---|---|
| Age Range | 18–60 | — | — |
| Mean Age (SD) | 39 (11) | — | — |
| Sex | | | |
| Male | | 78 | 38% |
| Female | | 126 | 61% |
| Other | | 2 | 1% |
| Ethnicity | | | |
| White | | 188 | 91% |
| Asian | | 8 | 4% |
| Black | | 4 | 2% |
| Two or More | | 4 | 2% |
| Other | | 2 | 1% |
| Education | | | |
| High school graduate | | 90 | 44% |
| Bachelor's degree | | 84 | 41% |
| Master's degree | | 13 | 6% |
| Doctorate degree | | 7 | 3% |
| Professional degree | | 5 | 2% |
| Some high school – no diploma | | 5 | 2% |
| Other | | 2 | 1% |

**Table 3** Demographics of Selected Participants (N = 206, Age = 18–60).

## 5.2 MOOD QUESTIONNAIRES

Participants completed a battery of mood-related questionnaires, including the Snaith-Hamilton Pleasure Scale (SHAPS), the Dimensional Anhedonia Rating Scale (DARS), the Generalized Anxiety Disorder Scale (GAD-7), and the Zung Self-Rating Depression Scale (ZUNG), to assess anhedonia and other related symptoms.

Ramaswamy et al.                    75
*Computational Psychiatry*

The SHAPS cut-off score of >2 was used to define anhedonia, as this is a recognized clinical threshold indicating significant deficits in hedonic capacity (Snaith et al., 1995). For the DARS, participants who scored ≤45 were classified as anhedonic. Participants who scored 0 on the SHAPS and >55 on the DARS were classified as non-anhedonic.

In addition to these anhedonia measures, participants completed the GAD-7, which measures anxiety, with cut-off points of 5, 10, and 15 corresponding to mild, moderate, and severe anxiety, respectively (Spitzer et al., 2006). The ZUNG was used to assess depressive symptoms, with a score of ≥50 indicating clinically significant depression (Zung, 1965). ZUNG total scores were computed from quantised item responses using standard reverse-scoring of positively worded items. These additional questionnaires provided context for understanding the broader emotional and psychological profiles of participants.

The SHAPS, GAD-7, and ZUNG questionnaires included an attention check item to identify inattentive responding. Where included, attention-check items were excluded from scoring and used only to identify inattentive responding. These items were chosen to be logically improbable statements, making inattentive responses easily detectable without directly signalling the check. The attention checks were as follows:

1. GAD-7: "Have there been times in your life where you blinked your eyes at least once per day?"

2. SHAPS: "Have there been times of a couple of days or more when you were able to breathe underwater (without an oxygen tank)?"

3. ZUNG: "I have never used a computer."

4. DARS: No attention check was included, as participants provided subjective responses across multiple items, ensuring engagement.

Findings from Zorowitz et al. (2023) underscore the importance of attention checks in symptom surveys, as inattentive responses can artificially inflate correlations between self-reported symptoms and cognitive measures. In our study, any participant failing even one attention check was excluded to ensure robust data quality. This rigorous approach reduces the risk of spurious findings and enhances the reliability of observed relationships between symptom measures and task performance, providing a clearer view of the psychological profiles of anhedonic and non-anhedonic participants.

To complement our earlier descriptive analyses based on the full pre-screened sample (N = 935), we also examined inter-scale correlations within the subset of participants who completed the task (N = 206; 111 anhedonic, 95 non-anhedonic). This targeted analysis provides a clearer characterization of how symptoms of anhedonia, anxiety, and depression relate within the actual experimental sample. We computed Pearson correlations and p-values between all combinations of the DARS, SHAPS, GAD-7, and ZUNG scores. Results are visualized in Figure 5 and reported alongside sample sizes for transparency.

## 5.3 THE 3-ARMED BANDIT REINFORCEMENT LEARNING TASK

Our 3-arm bandit task was adapted from Seymour et al. (2012)'s 4-arm probabilistic bandit paradigm, which included independent reward and punishment feedback for each choice. We made several modifications to optimize the task for online administration and support reliable modelling of approach and avoidance learning processes. First, we reduced the number of options from four to three, with the aim of simplifying decision-making and reducing cognitive load. In the original version, participants had to monitor 8 values (reward and punishment expectations across 4 options), whereas the 3-arm version only requires tracking 6 values (3 options × 2 outcome types), thus reducing working memory demands while maintaining the structure of the learning problem.

Second, we increased the volatility and ceiling of the drifting outcome probabilities, such that each option's probability of producing a reward or punishment independently varied between 0 and 0.75 over time (compared to 0–0.5 in the original version). This increase was designed to ensure that participants received more frequent and informative feedback across trials.

Ramaswamy et al. **76**
*Computational Psychiatry*

Participants completed 200 trials, which took approximately 15–20 minutes. On each trial, they were shown three distinct visual options and instructed to choose one using the W, A, or S keys on their keyboard. If no response was made within 3 seconds, the trial was skipped and a reminder was shown. Chosen options were highlighted briefly before the outcome was revealed.

Each of the three arms was associated with independently drifting reward and punishment probabilities, drawn from pre-generated sequences that were held constant across participants. Outcomes were determined stochastically on each trial according to these probabilities, such that while all participants experienced the same probability structure, the actual feedback they received varied depending on chance. This approach is common in reinforcement learning tasks and ensures equivalent task conditions while preserving trial-level variability in outcome realizations. The outcome for each trial was displayed using two circles: green for win, red for loss, and grey for the absence of outcome. Possible combinations included win-only (green + grey), loss-only (red + grey), both win and loss (green + red), or no outcome (grey + grey). We adopted this uniform visual format to avoid confounding differences in outcome salience across conditions. Participants were instructed to collect as many green tokens as possible, with bonus payment tied to the number of green tokens collected. Reaction times (RTs) were computed as the latency from stimulus onset to keypress; trials with RT < 200 ms, RT > 3000 ms, or missed responses were excluded. Full RT methods and results are provided in Supplementary S1–S2.

## 5.4 TASK VALIDATION WITH ORIGINAL 4-ARM BANDIT TASK

To validate whether the 3AB task captures the same learning mechanisms as the 4AB task, a pilot study was conducted with 111 participants completing both tasks in random order. The use of randomization ensured that there was no learning bias from one task to the other. Hierarchical Bayesian modelling was applied to both tasks to estimate key parameters related to learning rates and sensitivity to rewards and punishments.

## 5.5 DATA CLEANING

As the task was implemented online where we could not ensure the same testing standards as we could in-person, we used 2 exclusion criteria to improve data quality. We excluded those who responded with the same response key on 20 or more consecutive trials (> 10% of all trials). Additionally, we also excluded those who did not respond on 20 or more trials out of the total 200 trials.

## 5.6 COMPUTATIONAL MODELLING USING HIERARCHICAL BAYESIAN APPROACH

### 5.6.1 Model Selection

To evaluate which computational model best accounted for participants' trial-by-trial choices, we used the Leave-One-Out Information Criterion (LOOIC), a robust Bayesian approach to estimate out-of-sample predictive accuracy (Vehtari et al., 2017). Lower LOOIC scores indicate better model fit.

We tested a set of hierarchical reinforcement learning models that varied in their inclusion of valence-specific learning rates (Arew, Apun), reward/punishment sensitivity parameters (R, P), and control terms such as lapse rate (xi), decay rate (d), or inverse temperature (tau). All models were fit using hierarchical priors to estimate both group-level and individual-level parameters, improving parameter stability and generalizability.

These same models were also applied to data from the 4AB task in the validation study, using the same LOOIC-based comparison procedure. For interpretability, we report relative LOOIC values for the 3AB task (Figure 13), computed by subtracting the minimum LOOIC across models so that the model with a relative LOOIC of zero is the best-performing model. In the 3AB dataset, the lapse-augmented model with separate reward and punishment learning rates and sensitivities (banditNarm_lapse) achieved the lowest LOOIC. The single-learning-rate lapse model with split sensitivities (banditNarm_singleA_lapse) performed almost identically, and the four-parameter model with separate learning rates and sensitivities (banditNarm_4par) was the next best-

performing non-lapse candidate. Given the practical equivalence among these top models and our focus on valence-specific learning as the primary theoretical question, we retained banditNarm_4par as our primary hypothesis-testing model, while treating the lapse variants as robustness checks and reporting the full set of model comparison results.

Lapse-augmented models (banditNarm_lapse, banditNarm_singleA_lapse) achieved slightly better predictive performance than the four-parameter non-lapse model, consistent with a modest degree of stimulus-independent random responding in this online sample. This pattern aligns with previous bandit work in mood and anxiety disorders, where lapse terms have been interpreted as capturing 'trembling hand' or unexplained choices in otherwise well-specified RL models (Aylward et al., 2019; Mkrtchian et al., 2023). However, in line with broader modelling guidelines highlighting identifiability trade-offs when adding extra noise parameters (Wilson & Collins, 2019; Eckstein et al., 2021), we treat these lapse models as complementary checks on model fit rather than as our primary framework for hypothesis testing. To confirm that our main inference was not dependent on excluding lapse terms, we fit the dual learning rate + lapse model separately in the anhedonic and non-anhedonic groups and compared posterior group differences. For reward learning rate, punishment learning rate, reward sensitivity, and punishment sensitivity, the 95% HDIs for group differences all included zero (pd ≈ 0.64–0.74), and posterior densities overlapped strongly between groups. Estimated lapse rates were low on average (anhedonic mean ξ = 0.018; non-anhedonic mean ξ = 0.033), corresponding to approximately 2–3% lapses per trial on average.

Full posterior group-difference summaries for both lapse-augmented model families (banditNarm_lapse and banditNarm_singleA_lapse) are provided in the Supplementary Material (Supplementary S2; Figures S4–S5).

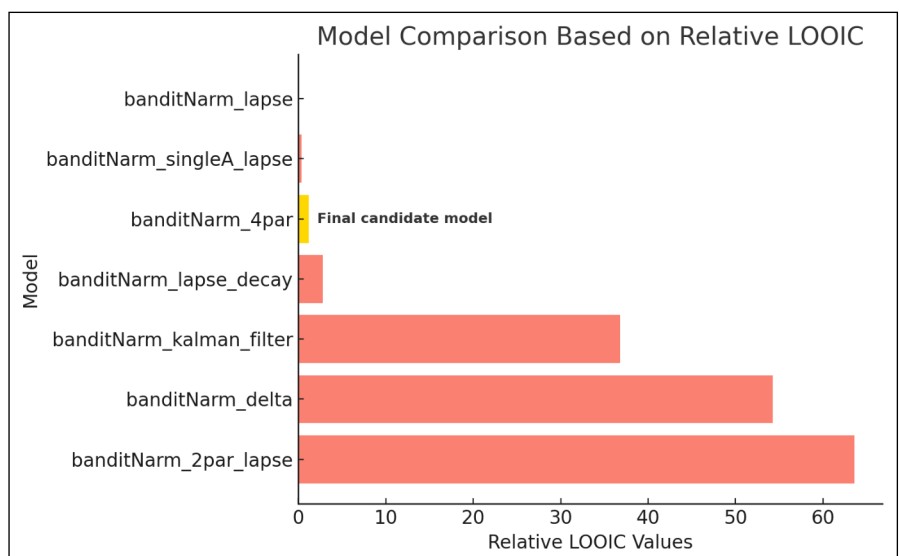

**Figure 13 Model Comparison Results Based on Relative LOOIC Values.** Bars show each model's difference in LOOIC relative to the best-fitting model (relative LOOIC = 0; lower is better). In this dataset, banditNarm_lapse achieved the lowest LOOIC, with banditNarm_singleA_lapse and banditNarm_4par performing very similarly. We retained banditNarm_4par as the primary model for hypothesis testing because it directly targets valence-specific learning and sensitivity parameters central to our anhedonia-related hypotheses, while lapse models were treated as robustness checks.

## 5.6.2 Model Specification

The final candidate model (banditNarm_4par) was adapted from Seymour et al. (2012) and previous work modelling aversive and appetitive learning. It tracks value updates for each option using separate Q-values for reward and punishment, updated independently via valence-specific learning rates.

- The model included the following participant-level parameters:
- Arew: learning rate for reward outcomes (0–1)
- Apun: learning rate for punishment outcomes (0–1)
- R: sensitivity to reward magnitude (0–30)
- P: sensitivity to punishment magnitude (0–30)

In each trial, Q-values for rewards and punishments (Qr, Qp) are updated as follows:

Ramaswamy et al.
**78**
*Computational Psychiatry*

$$Qr_{t+1}(a) = Qr_t(a) + A_{rew} \times \delta_{rt}(a)$$

$$Qp_{t+1}(a) = Qp_t(a) + A_{pun} \times \delta_{pt}(a)$$

Reward and punishment sensitivities enter the model by scaling the observed outcomes inside the prediction errors. For participant *i*, arm *a* and trial *t*, the reward and punishment prediction errors are:

$$\delta_{i,t}^{(r)}(a) = R_i \cdot r_{i,t}(a) - Q_{i,t}^{(r)}(a)$$

$$\delta_{i,t}^{(p)}(a) = P_i \cdot l_{i,t}(a) - Q_{i,t}^{(p)}(a)$$

where $r_{i,t}(a)$ and $l_{i,t}(a)$ denote the reward and punishment outcomes on that trial. The corresponding Q values are then updated as:

$$Q_{i,t+1}^{(r)}(a) = Q_{i,t}^{(r)}(a) + A_i^{(r)} \cdot \delta_{i,t}^{(r)}(a)$$

$$Q_{i,t+1}^{(p)}(a) = Q_{i,t}^{(p)}(a) + A_i^{(p)} \cdot \delta_{i,t}^{(p)}(a)$$

Fictive updates are applied to the unchosen options via $\delta_{fic}^r(a) = -Q_{i,t}^{(r)}(a)$ and $\delta_{fic}^p(a) = -Q_{i,t}^{(p)}(a)$, which produces gradual decay of unchosen values. Choices are generated from a softmax over the sum of reward and punishment values:

$$Q_{i,t}^{(sum)}(a) = Q_{i,t}^{(r)}(a) + Q_{i,t}^{(p)}(a)$$

$$P(c_{i,t} = a) = \frac{\exp\left(Q_{i,t}^{sum}(a)\right)}{\sum_b \exp\left(Q_{i,t}^{sum}(b)\right)}$$

We chose to scale outcomes via R and P rather than include a separate inverse temperature parameter, to keep valuation and decision noise distinct and to avoid strong collinearity between these terms. Full Stan code for this model and all alternatives are provided in Supplementary.

Although some models tested included an additional inverse temperature parameter or a lapse rate, we opted to scale the Q-values directly via R and P rather than include a separate temperature term. This avoids overparameterization and ensures clear interpretation of reward and punishment valuation, which were the focus of our hypotheses. Model comparison indicated that several candidate models achieved very similar predictive performance; we therefore retained banditNarm_4par as our primary hypothesis-testing model for interpretability and theoretical focus.

We used weakly informative priors for all group-level parameters to support stable estimation without imposing strong assumptions. Specifically, the group-level means (mu_pr) were drawn from a standard normal distribution ($N(0,1)$), and the standard deviations (sigma) from a half-normal prior $N^+(0,0.2)$, consistent with prior literature on hierarchical reinforcement learning models (e.g., Ahn et al., 2017; Wiecki et al., 2013). Subject-level parameters were then transformed via the probit ($\Phi$) function to ensure appropriate bounds for each parameter (e.g., [0,1] for learning rates; [0,30] for sensitivities). These choices are consistent with standard practice in hierarchical Bayesian modelling for cognitive tasks and were not intended to encode strong prior beliefs.

### 5.6.3 Model Variants Tested

In addition to the four-parameter model (banditNarm_4par; separate reward and punishment learning rates Arew, Apun and sensitivities R, P), we fit several alternative model families based on the N arm bandit implementations in hBayesDM. These included:

i. lapse models (banditNarm_lapse, banditNarm_lapse_decay, banditNarm_2par_lapse), which use the same value updates but introduce a lapse parameter ξ to mix softmax choice probabilities with a uniform random policy (with banditNarm_lapse_decay additionally including a decay term);

Ramaswamy et al.          **79**
*Computational Psychiatry*

**ii.** single learning rate models (banditNarm_delta, banditNarm_singleA_lapse), in which reward and punishment share a common learning rate A while either using a classical inverse temperature parameter in the softmax (banditNarm_delta) or retaining separate sensitivities R and P with a lapse parameter ξ (banditNarm_singleA_lapse); and

**iii.** a Kalman filter model (banditNarm_kalman_filter), which tracks option values and their uncertainty with parameters governing mean reversion (λ, θ), diffusion variance (sD) and initial uncertainty (s0), together with an inverse temperature β in the softmax decision rule.

A detailed summary of all models and parameters is given in Table 4, and full Stan code is provided in the Supplementary. A total of seven models were tested (Figure 13), spanning Rescorla–Wagner variants and models with additional lapse, decay, or Kalman filter components (Table 4).

| MODEL | PARAMETERS INCLUDED | KEY FEATURES |
|---|---|---|
| **banditNarm_2par_lapse** | Arew, Apun, ξ | Reduced lapse model with separate reward and punishment learning rates (Arew, Apun) but no sensitivity parameters; includes lapse ξ capturing stimulus-independent random responding |
| **banditNarm_4par** | Arew, Apun, R, P | Valence-specific learning rates and separate reward/punishment sensitivities (R, P); no lapse or temperature term; focus model for testing anhedonia-related hypotheses. |
| **banditNarm_delta** | A (shared), tau | Simple Rescorla–Wagner model with a single shared learning rate A across reward and punishment and an inverse temperature tau; no separate sensitivity parameters. |
| **banditNarm_kalman_filter** | λ, θ, s0, sD, β | Kalman filter over option values with mean reversion (λ, θ), uncertainty tracking (s0, sD) and inverse temperature β; no explicit learning-rate parameter. |
| **banditNarm_lapse** | Arew, Apun, R, P, ξ | Same valence-specific learning rates and sensitivities as banditNarm_4par, plus a lapse parameter ξ that mixes softmax choice with uniform random responding. |
| **banditNarm_lapse_decay** | Arew, Apun, R, P, ξ, decay | As banditNarm_lapse but with an additional decay term on Q-values; tests combined effects of sensitivity, ξ, and slow forgetting of option values over time. |
| **banditNarm_singleA_lapse** | A (shared), R, P, ξ | Single shared learning rate A with separate reward and punishment sensitivities (R, P) and lapse ξ; directly probes whether a shared vs valence-specific learning rate better explains behaviour. |

**Table 4 Model Variants Tested.** Comparison of reinforcement learning models tested on the 3-arm bandit task. Each model varies in included parameters and computational assumptions. Arew = reward learning rate; Apun = punishment learning rate; A = shared learning rate; R = reward sensitivity; P = punishment sensitivity; ξ = lapse parameter (choice noise); τ = inverse temperature (softmax); β = inverse temperature/precision parameter in the choice rule; λ, θ, s0, sD = Kalman filter parameters governing mean reversion and uncertainty dynamics (see Supplementary for full definitions); decay = Q-value decay rate.

In the lapse models (banditNarm_lapse, banditNarm_singleA_lapse), a subject-specific lapse parameter ξ mixes the softmax policy over option values with a uniform random policy, capturing stimulus-independent lapses that are expected in online samples. Although such terms can improve predictive performance by absorbing random or unmodelled choices, recent tutorials emphasise that adding extra stochastic parameters can make it harder to uniquely interpret the remaining learning and sensitivity parameters, and that these richer models require careful simulation-based model and parameter-recovery analyses to assess identifiability (Wilson & Collins, 2019; Eckstein et al., 2021). In the present study we therefore used lapse-augmented models primarily for model comparison and posterior predictive checks, and based our group-level inferences about reward and punishment learning and sensitivity parameters on the corresponding non-lapse models, where recovery was adequate.

## 5.6.4 Model Fitting and Estimation

We implemented the candidate model in a hierarchical Bayesian framework using MCMC sampling in Stan (2,000 iterations, 1,000 warmups, 4 chains), which estimated both individual and group-level parameters. We initially fit the hierarchical model jointly across all participants to estimate

Ramaswamy et al. **80**
*Computational Psychiatry*

shared group-level parameters. As this analysis revealed no significant group differences, we then fit the model separately for the anhedonic and non-anhedonic groups to allow for distinct group-level priors and posterior distributions. Both approaches yielded comparable results, supporting the robustness of our findings across model-fitting strategies. The group-level parameters for learning rates and sensitivities were modelled as normally distributed, with learning rates and sensitivities constrained by an inverse probit transformation to lie within [0,1] and [0,30], respectively. Model fit was assessed via log-likelihood of observed choices, with posterior predictive checks to confirm that the model reproduced observed choice behaviour. Convergence diagnostics, such as the Gelman-Rubin statistic, verified adequate parameter convergence.

### 5.6.5 Model Prediction Accuracy and Comparison to Observed Choices

To assess the predictive validity of the hierarchical Bayesian model, we compared model-generated action probabilities to participants' actual choices on each trial. For each participant and trial, we computed the predicted probability of selecting each of the three arms, based on the posterior samples of individual parameter estimates. The predicted choice was defined as the arm with the highest predicted probability.

We calculated per-subject prediction accuracy as the proportion of trials (out of 200) in which the model's top-predicted action matched the participant's actual choice. This provided an intuitive index of how well the model reproduced observed behaviour. Accuracy scores were then summarized across subjects to yield a group-level distribution, which we visualized in a histogram (see Figure 11). The average accuracy across the sample was 59.60% (SD = 16.73%), substantially above the chance level of 33%.

In addition, we plotted trial-by-trial predicted probabilities alongside actual choices for three representative participants (see Figure 10). This visual comparison further illustrates the model's ability to capture individual decision dynamics throughout the task.

### 5.6.6 Simulated Data

To validate model parameters, we simulated 200 trials per subject in a 3-armed bandit structure. Choices were generated based on each participant's estimated parameters (Arew, Apun, R, P), using the softmax function to determine probabilities of selecting each option. Outcomes were sampled based on predefined reward and punishment probabilities, and Q-values were updated accordingly. For unchosen options, Q-values were updated using counterfactual updates, where no reward or punishment (i.e., an outcome of zero) was assumed. This approach reflects implicit learning effects, capturing how participants might adjust expectations for unselected options even in the absence of observed outcomes. By incorporating these updates in the simulation, we ensured consistency with the assumptions of our hierarchical Bayesian model.

### 5.6.7 Parameter Recovery

We performed parameter recovery to validate model accuracy by generating simulated data based on original model parameters and fitting the model again to this data. Comparing recovered parameters with original parameters allowed us to evaluate the accuracy of estimates for reward learning rate (Arew), punishment learning rate (Apun), reward sensitivity (R), and punishment sensitivity (P). The hierarchical Bayesian model fit was repeated using MCMC sampling in the hBayesDM package, with the posterior distributions providing reliable estimates of individual differences. Convergence diagnostics and posterior predictive checks further confirmed model validity.

## 5.7 MODEL-AGNOSTIC ANALYSIS

### 5.7.1 Win-Stay and Lose-Shift Strategy Calculation

To assess participants' decision-making, we examined two common behavioural patterns: win-stay and lose-shift. A win-stay strategy occurs when participants repeat a choice following a reward, while a lose-shift strategy occurs when they switch choices after a punishment. For each

Ramaswamy et al.                    **81**
*Computational Psychiatry*

participant, we extracted trial-by-trial choices and outcomes. Win-stay instances were defined as trials where a participant received a reward on the previous trial and repeated the same choice. Lose-shift instances were trials where a participant received a punishment and chose a different option on the following trial. The percentage of win-stay and lose-shift strategies was calculated by dividing the relevant instances by the total opportunities for each strategy, then multiplying by 100. We report the mean and standard error of these percentages across participants, presented in a bar plot with error bars indicating the standard error. This analysis offers a model- agnostic perspective on the prevalence of win-stay and lose-shift strategies in decision-making.

## 5.8 BAYES FACTOR TEST

To further assess the evidence for no difference between the anhedonic and non-anhedonic groups, a Bayesian independent samples t-test was conducted for each model parameter (Arew, Apun, R and P). Bayes factors (BF10 and BF01) were calculated using the BayesFactor package in R, with BF01 representing the evidence in favour of the null hypothesis relative to the alternative hypothesis.

## 5.9 BAYESIAN ESTIMATION OF GROUP DIFFERENCES USING HIGHEST DENSITY INTERVALS (HDIS)

In addition to frequentist and Bayes Factor comparisons, we estimated the difference in group-level posterior means for each computational parameter (Reward Learning Rate, Punishment Learning Rate, Reward Sensitivity, and Punishment Sensitivity) using 95% Highest Density Intervals (HDIs). For each parameter, we extracted the group-level posterior samples (mu parameters) for the Anhedonic and Non-Anhedonic groups from the hierarchical Bayesian model. These samples were used to compute the posterior distribution of the difference (Non-Anhedonic – Anhedonic). HDIs were computed using the HPDinterval function from the coda package in R, which identifies the narrowest interval containing 95% of the posterior mass. We also calculated the probability of direction (pd), defined as the proportion of posterior samples falling consistently above or below zero. Differences for which the 95% HDI excluded zero were considered credibly different across groups. Posterior densities and intervals were visualized using the tidybayes and ggplot2 packages, with separate panels for each parameter. These analyses provide a Bayesian alternative to frequentist t-tests and complement the Bayes Factor approach.

## 5.10 USE OF GENERATIVE AI

Generative AI (ChatGPT, OpenAI) was used solely for manuscript refinement, including copyediting and language polishing. All scientific content, study design, data analysis, and interpretation were developed by the authors, with AI assistance limited to improving clarity and readability without altering the narrative or conclusions.

## ADDITIONAL FILE

The additional file for this article can be found as follows:

- **Supplementary Material.** Supplementary material containing additional reaction-time (RT) analyses and robustness checks for the reinforcement-learning models, including subject-level and trial-level RT comparisons between anhedonic and non-anhedonic groups, and posterior group-difference HDI summaries and density plots for lapse-augmented model variants. DOI: https://doi.org/10.5334/cpsy.135.s1

## FUNDING INFORMATION

This research was supported by the Medical Research Foundation (MRF) grant, which facilitated the study. The funding body had no influence on the study design, data collection, analysis, or interpretation of the results, nor on the decision to submit the manuscript for publication.

## COMPETING INTERESTS

The authors have no competing interests to declare.

## AUTHOR AFFILIATIONS

**Arjun Ramaswamy** orcid.org/0000-0002-7173-1500
UCL Queen Square Institute of Neurology, London, UK; Department of Imaging Neuroscience, UCL, London, UK

**Yumeya Yamamori** orcid.org/0000-0001-5508-7965
UCL Institute of Cognitive Neuroscience, London, UK

**Umesh Vivekananda** orcid.org/0000-0001-6116-2335
UCL Institute of Cognitive Neuroscience, London, UK; Department of Clinical and Experimental Epilepsy, UCL Queen Square Institute of Neurology, London, UK

**Vladimir Litvak** orcid.org/0000-0001-8535-7452
UCL Queen Square Institute of Neurology, London, UK; Department of Imaging Neuroscience, UCL, London, UK

**Jonathan P. Roiser** orcid.org/0000-0001-8269-1228
UCL Institute of Cognitive Neuroscience, London, UK

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
