## [Reviewer Report · Round 1, Reviewer Colin Hoy]

## Comments/Explanation (required). Please include references to any minor or major revisions required, as well as addressing any issues with language clarity. Authors will see these comments:

This article addresses whether computational model parameters describing reward and punishment learning in a three-arm restless bandit task can predict anhedonia symptoms in an online sample. The authors modified a classic four-arm restless bandit task with probabilistic, fluctuating rewards to have fewer choice options (three bandits) and more frequent rewards and punishments, and they correlate reinforcement learning model parameters (learning rates and sensitivity for reward/punishment) to several clinical scales, including the recently developed dimensional apathy rating scale (DARS). After performing some model validation, the main analyses show that model parameters are not significantly related to anhedonia symptoms.

Major comments:

I applaud the authors for reporting null results, which can provide valuable direction to the field. In particular, there is an important gap between task-based and self-report measures in psychiatry, which often do not correlate despite measuring purportedly similar constructs (e.g., Eisenberg et al., 2019 Nature Comm. and Enkavi et al., 2019 PNAS in self-regulation). Thus, characterizing relationships between anhedonia symptoms and reward learning task behavior or the lack thereof is important for the field of computational psychiatry. Establishing confidence in a null result is challenging, and the authors provide a variety of evidence to support this finding, including use of optimal hierarchical Bayesian model fitting, parameter recovery, and model comparisons, tests of model agnostic measures (win-stay, lose-shift), and Bayes Factor t-tests. However, several flaws in the study weaken their main conclusion that “core cognitive mechanisms underlying reward and punishment learning remain intact in individuals with anhedonia”. This may be true in this dataset and inconsistent in the literature, but the manuscript cites multiple previous studies and meta-analyses on deficits in reward processing and depressive symptoms without offering insights into why previous tasks or this task fail to capture anhedonia, resulting in a limited contribution to the field. Furthermore, their secondary conclusion that “the 3AB task provides a novel approach for studying reward processing” is not supported given previous studies using similar 3-arm bandit tasks (e.g., see Yan et al., 2025 Biol. Psy.: CNN for a recent finding on apathy).

One core problem with the manuscript is that the experiments don’t seem to follow from the stated motivation. The introduction states that anhedonia may be related to motivation more than reward learning and cites a study relating anhedonia to willingness to exert effort for reward (Treadway et al., 2009), but the study then pursues a reward learning task and model that typically focuses on learning rates. There is no review of previous literature relating the 4-arm restless bandit to depression, apathy, or anhedonia, aside from mentioning this task has “limited sensitivity to reward-processing impairments associated with anhedonia”. Why then choose this task to study anhedonia? Moreover, the rationale for modifying the task and the specific changes chosen are not well justified. The authors do not support their argument that simplifying decision making processes might increase task sensitivity to anhedonia, or why the model parameters would capture variance in different anhedonia dimensions (e.g., DARS subscales). Similarly, reducing cognitive load and increasing reward and punishment rates could in principle be helpful (e.g., to study patient populations with cognitive deficits and increase engagement), but the relevance of those changes to the current study is unclear. An analysis of how changes in individual task features affect behavior and the relationships to symptoms would be valuable, but inferences about the relevance of those features to symptoms are precluded by changing multiple features at the same time, as well as the lack of comparison to previous results using the 4-arm restless bandit task.

There are also some problems with the behavioral modeling that should be addressed to increase confidence in the null results. First, core hypotheses and results center on reward and punishment sensitivity parameters, but the Methods do not show how these are incorporated in the models. Second, model specification and selection processes are not adequately described and justified. For example, the authors do not include a free parameter for the inverse temperature parameter in their softmax decision function, despite citing a previous study (Harle et al., 2017) that found this parameter predicted anhedonia. Learning rates and decision temperature are typically correlated in most datasets/models, so failing to account for this variance in the decision policy likely influences the other parameters, meaning this choice should be justified and evaluated in the model comparison process. Similarly, including counterfactual updates in a task with independent choice options is not rational, so the authors should verify that including those terms improves the fit to the behavioral data. The authors also do not describe the rationale or implementation of the “lapse” parameter. Overall, the value of the null result is weakened by a somewhat confusing model selection process, which may be partly due to the lack of clear connection to previous tasks, models, and results and motivation for modifying the task and models.

The strength of the null findings is also limited by the sample size for the anhedonia (n=111) and non-anhedonia (n=95) groups, which is modest for online behavioral studies using between-subjects designs and has limited statistical power per the authors’ power analysis. For context, the authors cite Huys et al. (2013), which used a meta-analysis of 6 datasets (392 sessions) to find anhedonia was more related to reward sensitivity than learning rates, and Halahakoon et al. (2020) found small-to-medium effect sizes on reward processing in depression. This challenge may be exacerbated by the heterogeneous relationships between clinical scales in this dataset. For example, the DARS and ZUNG show a moderate positive correlation (r=0.34), which counter-intuitively suggests that higher depression scores are associated with lower anhedonia in this sample. Using multiple anhedonia scales is helpful, but it would help evaluate the sample to report p values for the between-scale correlations and to report the statistics of these scales in the two groups that performed the task.

In summary, the study is within scope of the journal, but the manuscript is not suitable for publication in its current form due to core problems with the content (engagement with prior literature, model selection) and structure (motivation for task changes and hypotheses), as well as a number of smaller issues articulated below.

Additional comments:

RTs are mentioned in the abstract and discussion (1st and 2nd to last paragraphs), but these results aren’t presented in the manuscript.

Please describe the prior used for each parameter in the hierarchical Bayesian framework and justify these choices (e.g., citations) when appropriate.

The language describing the Harle et al. (2017) finding on the inverse temperature softmax parameter as “reward sensitivity” should be rephrased to avoid confusion with the reward sensitivity parameter in your model.

One potential avenue to address the heterogeneity in clinical scales would be using dimensionality reduction or factor analyses to extract latent symptom dimensions, but such exploratory, follow up analyses should ideally be validated in a new replication sample.

Why did the authors provide two outcomes on each trial? This complicates the interpretation, and it’s possible that combining reward and punishment conditions in the same trials may reduce the ability to tease apart these distinct processes.

The authors might also wish to consider the role of drift rate in their reward sequences, which should be reported in the Methods. Similarly, differences in the reward probability sequences experienced across participants are another factor that could potentially influence choice difficulty, reward and punishment rates and thus add noise to the parameters and symptom correlations.

The authors show three representative subjects when comparing predicted action probabilities and choices, but it would be more informative to report the proportion of choices predicted by the model at the group level.

The authors state that they increased reward and punishment rates by a factor of 1.5, but it would be more straightforward to state the new rates.

Formatting:

missing references: Seymour 2012, Daw 2006

citations needed: sentence 2; P2 last sentence; "most only correlate with SHAPS"; "anhedonia is perhaps better characterized by impairments in reward valuation."

Fig 1c/d- hard to see with the colors

spell out SMD

## peer-review-recommendation

See Comments

---

## [Reviewer Report · Round 1, Reviewer Poornima Kumar]

## Recommendation:

Accept with Major Revision

## Comments/Explanation (required). Please include references to any minor or major revisions required, as well as addressing any issues with language clarity. Authors will see these comments:

This is an interesting study where the authors have developed a 3-arm bandit task, which they show has similar performance to a 4-arm bandit task. Contrary to their hypotheses, they do not find any differences in learning rate or sensitivity between non-anhedonic and anhedonic individuals. The authors claim that this could be either that anhedonia not contributing to learning impairments, the study being underpowered or the task not being sensitive to detect these small effects.

There are several concerns about their statistical methods that call into question the robustness of these results.

1. Not much information is provided about the different models used and why they were selected over others. For example, why weren’t single learning rate models considered?

2. Did the same model come out on the top for both 3 arm and 4 arm bandit tasks? Not much info is provided about the validation study, except for the sample size. It would be helpful for the authors to provide more details on how the 3-arm task was validated against the 4-arm task.

3. The correlation between the parameters from these two tasks do not seem very convincing. Authors say that these correlations were significant in the text, but no p values are shown in the plots. There is a lot of variability in these parameters which makes me question whether learning is truly similar across both. It will also be helpful to plot model-agnostic scores across both tasks, such as the probability of staying after a win/lose. Additionally, the axes use different scales, which should be avoided as it confounds the interpretation.

4. Did they fit the models across the entire sample or incorporated group information in the hierarchical estimation, it is not clear from their description?

5. I am not sure why they do not show the posterior plots of the model parameters and report HDI instead of doing t-tests ofn the mean, which lose the relevance of Bayesian modeling

6. They do not show whether the simulated data recapitulated patterns observed in the model-agnostic task analysis, especially since the parameter recovery plots do not look convincing, particularly for learning rates. I would also recommend running many simulations and getting an idea of how consistent the spread of simulated parameters are.

7. Is there any additional information on participants’ mental health history, medication, etc?

8. Would authors considered scoring SHPS on a 1-4 scale? This might provide a better spread of scores.

9. It is also important to describe how well the participants performed in the validation study? The tasks are quite long ~20 min each, making for an extended session if they also have to complete questionnaires.

Overall, I would recommend that the authors structure the paper with a focus on the validation study first, followed by the second study with detailed info on the model selection, as well as appropriate parameter and model recovery.

## peer-review-recommendation

Revisions Required

---

## [Reviewer Report · Round 1, Reviewer Vanessa Brown]

## Recommendation:

Reject

## Comments/Explanation (required). Please include references to any minor or major revisions required, as well as addressing any issues with language clarity. Authors will see these comments:

This paper reports on the development of a 3-arm bandit learning task. The goal of the study was to develop a version of the 4-arm bandit learning task with less cognitive load and to test this revised task in a large online sample to find relationships between learning parameters and anhedonia.

The paper is written clearly and the authors lay out a strong justification for carrying out the research. I appreciate the authors’ straightforward reporting of null results. I have concerns with the sample, task, and analysis approaches, though.

Major concerns:

1. The larger sample from which the high- and low-anhedonia participants were sampled has very odd relationships among scales. Participants reporting less depression also reported more anhedonia, and there was no relationship between depression and GAD-7 scores. This pattern does not reflect known relationships among these measures. It instead suggests 1) this sample’s pattern of psychopathology is unusual, and findings are unlikely to generalize to the larger population, 2) participants responded inattentively or did not understand instructions, or 3) participants were aware of the initial screening and answered in a way that did not reflect their experiences in order to pass the screening. More stringent data cleaning may create a subsample of participants who report expected relationships among symptoms; however, as-is the questionnaire data does not appear valid.

2. It’s not clear why the authors chose to artificially dichotomize anhedonia and lose power, particularly when one of the measures used (SHAPS) has poor psychometric properties.

3. A strength of the paper, as the authors note, is the use of hierarchical Bayesian approaches. However, the authors deal with the estimated parameters incorrectly by using individual parameters (which have been influenced by other participants’ data through partial pooling) in subsequent analyses. The authors do not report how they determined group structure, but I assume they analyzed all participants as part of one large group in the hierarchical structure. If so, this will shift participants’ estimates closer together and increase the rate of false negatives when those estimates are then correlated with other variables (i.e., anhedonia). Please see citations below. The authors can confirm these effects by simulating parameters with a known relationship to an external measure (representing anhedonia) and examining the relationship with recovered parameters that are estimated using the current approach vs. the one in the citations below.

Boehm, U., Steingroever, H., & Wagenmakers, E. J. (2018). Using Bayesian regression to test hypotheses about relationships between parameters and covariates in cognitive models. Behavior research methods, 50, 1248-1269.

Haines, N., Beauchaine, T. P., Galdo, M., Rogers, A. H., Hahn, H., Pitt, M. A., … & Ahn, W. Y. (2020). Anxiety modulates preference for immediate rewards among trait-impulsive individuals: A hierarchical Bayesian analysis. Clinical Psychological Science, 8(6), 1017-1036.

Brown, V. M., Chen, J., Gillan, C. M., & Price, R. B. (2020). Improving the reliability of computational analyses: Model-based planning and its relationship with compulsivity. Biological Psychiatry: Cognitive Neuroscience and Neuroimaging, 5(6), 601-609.

Waltmann, M., Schlagenhauf, F., & Deserno, L. (2022). Sufficient reliability of the behavioral and computational readouts of a probabilistic reversal learning task. Behavior research methods, 54(6), 2993-3014.

4. The authors state the rationale for developing the 3-armed bandit is to reduce cognitive load which may camouflage relationships with symptoms.

a. The 3-armed bandit task developed here has varying levels of reward and punishment for each arm, while the original 4-armed bandit used by Daw etc. usually only has reward. This means participants now have to track 6 units of information vs. 4. It’s not clear how this reduces cognitive load.

b. More broadly, how do the authors define and measure cognitive load? How can we know that 1) this task is lower in cognitive load, and 2) what level of cognitive load is low enough to not interfere with learning?

## peer-review-recommendation

See Comments

---

## [Reviewer Report · Round 2, Reviewer Poornima Kumar]

## submission-comments

Authors have answered all of my concerns. No further comments.

## peer-review-recommendation

Accept Submission

---

## [Reviewer Report · Round 2, Reviewer Vanessa Brown]

## submission-comments

I thank the authors for their responses to my and the other reviewers’ comments. Most of my concerns have been addressed; however, one major concern was not addressed which casts major doubt on the findings. The strange correlations among symptom measures suggest that these are not valid measures. It’s true that correlations among measures can differ based on the severity of the sample, such that we may see slightly weaker relationships in online, non-clinical samples. It’s also true that anhedonia, depression, and anxiety can load onto distinct latent dimensions. I also appreciate the authors’ strict data cleaning procedures.

However, none of this means that we should encounter a sample where more depressed people report less anhedonia. If we examine the lower right plot in Figure 5 – the people screened for SHAPS = 0 are reporting a higher mean depression score than people reporting severe anhedonia. In what world would this make sense? Even if the authors did somehow find a sample where this relationship truly existed, results in this sample would not generalize at all to the population, where anhedonia and depression are highly correlated.

## peer-review-recommendation

Decline Submission

---

## [Reviewer Report · Round 2, Reviewer Colin Hoy]

## submission-comments

The authors have revised and improved the manuscript, and I am fairly convinced that the relationships between task behavior and anhedonia are null because the authors have done quite a bit of work to show their modeling is sound. Null results have value, but that value is dependent on the quality of the study. Another source of marginal value would be thorough characterization of the properties of the task and the models in this sample, and there are still some puzzling issues that may influence the interpretation of the results. Overall, I would see this paper as reasonable evidence to be cautious about using this bandit task to study anhedonia, and there are a few minor analysis items that might be helpful to add that could inform that decision. However, the study is not conclusive on the larger conclusion of whether reward sensitivity or learning is altered in anhedonia, partly due to flaws in study design and partly because it’s only a single study that quite possibly is underpowered. Below, I comment on the design issues, model specification and comparisons, and a potentially interesting RT effect.

A core limitation of the study is still that we cannot infer why the relationships with anhedonia are null because too many task features and modeling features have changed from prior studies. The authors improved the motivation for some of these issues in the Introduction, but some of these statements are still not justified. As a minor example, how does outcome feedback on every trial enhance model identifiability? This is likely a good idea, but the stated rationale on model identifiability is still not clearly justified to me. More importantly, while the authors now acknowledge they cannot confirm that changing from 4 to 3 bandits successfully reduced cognitive load, but the relevance and motivation to include a punishment condition in a study seeking reward-specific deficits (anhedonia) is unclear and likely affects analysis sensitivity. The authors briefly allude to approach-avoidance frameworks, but don’t support the relevance of that construct to anhedonia. Showing a selective relationship between anhedonia and reward but not punishment would be a great dissociation (note this rationale/hypothesis is not stated), but in terms of cognitive load, independent reward and punishment for 3 bandits is still more outcome contingencies (6) than the original reward-only 4 arm bandit. Furthermore, adding punishment sensitivity and learning rate may reduce power and identifiability of the hypothesized reward deficits in anhedonia. Again, these issues are not huge but do reduce some of the value and interpretability of this null result, and they also contribute to some of the modeling concerns below.

One key improvement in the revision is the model specification and comparison, though there are still some outstanding questions. First off, the key reward and punishment sensitivity terms are still missing from the model equations (likely in the prediction error computation, which isn’t shown?), and the alternative model structures are not specified (e.g., inverse temperature, lapse, and Kalman filter parameters). Whatever value can be derived from these null results depends on specifying exactly what was done, and the current Methods does not provide enough information to interpret or reproduce the analyses in the paper. This issue is especially problematic because the model comparisons show models with a lapse parameter perform better than the main model selected as “optimal”. The authors argue in multiple places throughout the manuscript and response-to-reviewers that the model with (only) separate reward and punishment sensitivity and learning rate parameters (with no other parameters) is the optimal model. I agree that including valence-specific sensitivity and learning rate parameters is justified as the most theoretically appropriate test of their hypothesis. However, they sometimes use loose language to imply that this theoretical rationale is also supported by model comparisons (e.g., “Model comparison confirmed that the selected model balanced predictive accuracy with theoretical parsimony”). In contrast, they never acknowledge that a model with a single learning rate, split sensitivities, and a lapse parameter has a better fit. This might be important for their hypotheses, but they do not test a model with a single learning rate, split sensitivities, and no lapse parameter to directly compare whether a single learning rate model is better than their chosen model. Thus, we don’t know if single or dual learning rates are better in this task and dataset.

Relatedly, the relatively large differences in lose-switch behavior between simulated and real data, combined with the lower recoverability of punishment learning rate compared to other parameters, hint that there may be some trouble modeling the punishment effects on behavior. Combined with the unclear relevance of punishment to anhedonia, these data suggest this task may not be ideally suited to the question of interest. To me, these are the types of subtle inferences that might help the field in deciding which tasks are a good tool for assessing anhedonia (even in the context of null results), and showing similar effects in the 4 arm bandit task might strengthen this contribution.

Additionally, model comparisons showing performance improvements with the lapse parameters suggest it’s helpful to account for random responding, which may be prevalent in online samples. I agree that adding an inverse temperature parameter may reduce identifiability of the sensitivity and/or learning rate parameters, but it’s less unclear if adding the lapse term would reduce identifiability, particularly in the models without inverse temperatures. Perhaps there are parameter identifiability or model recoverability results that justify why they didn’t choose to add a lapse parameter, but it seems they could still test their 4 parameters of theoretical interest (+ and – sensitivities and learning rates) in the best performing model.

Lastly, the authors have added the missing RT data to the manuscript, and they seem to be ignoring an interesting effect where RTs are slower for non-anhedonic individuals. They report p=0.050 as non-significant, but by eye at least, it seems likely that this effect would be clearly significant in the stable regime after excluding the outlier trials at the start of the task. Also, was there an order effect between the two tasks? In particular, I wonder if the large change in RTs from the first ~5-10 trials to the remaining task was absent or smaller if they had already done the 4AB first.

Minor points:

The authors say that including fictive outcome updates improved model fit, but don’t show this. I agree that literature can support theoretical reasons to include this parameter, potentially even if it reduces model performance if that rationale is strong enough, but showing this performance increase could be another minor improvement to help infer the reasons for a null result contrary to prior findings motivates (e.g., how much does it help?).

Stylistically, I would appreciate the response to reviewer quoting the entire review, as there can be context lost when selectively quoting individual sentences, and it can be harder to evaluate the response without those details.

Also, it would be helpful to highlight the new/changed text in the manuscript (e.g., in different color) to identify where the revisions were implemented, as well as to list the line numbers of the new text in the response-to-reviewers. There’s at least one place the edits to the manuscript do not exactly match some of the text in the response to reviewers (“reviewer concerns” vs “concerns” on line 689), so this mapping should be double checked.

## peer-review-recommendation

Revisions Required

---

## [Reviewer Report · Round 3, Reviewer Vanessa Brown]

## submission-comments

I thank the authors for their response. If the non-anhedonia-related items on the Zung are driving this odd correlation, this should be tested explicitly - do the anhedonia-related items on the Zung show the expected relationship with anhedonia?

## peer-review-recommendation

Revisions Required

---

## [Reviewer Report · Round 1, Author Response]

## Reviewer C

**Comment 1:** One core problem with the manuscript is that the experiments don’t seem to follow from the stated motivation. The introduction states that anhedonia may be related to motivation more than reward learning and cites a study relating anhedonia to willingness to exert effort for reward (Treadway et al., 2009), but the study then pursues a reward learning task and model that typically focuses on learning rates…

## Author’s response

**Response:** We thank the reviewer for this helpful comment and agree that the framing of the study’s rationale required greater alignment with the experimental design. In the revised Introduction, we have clarified that the study focuses on potential alterations in reward sensitivity and learning associated with anhedonia, rather than motivational drive or effort expenditure. To this end, we removed references to motivation-related constructs and reframed the theoretical background to focus on outcome-based reward valuation and reinforcement learning. The revised text now emphasizes that reinforcement learning paradigms have produced mixed results when examining trait anhedonia, and that this study aims to evaluate whether the present 3-arm bandit task is sensitive to group-level anhedonia differences in learning and valuation parameters. We also clarified that our task was adapted from Seymour et al. (2012), which included both reward and punishment outcomes with probabilistic drift, and not from the classic two-armed bandit paradigms that focus solely on exploration and exploitation. The revised Introduction reflects this lineage and specifies how our 3AB version simplifies the design while retaining key features of approach and avoidance learning.

Revised text (Introduction)

“Importantly, anhedonia is distinct from motivational deficits such as apathy or effort discounting (Husain & Roiser, 2018). Instead, it may reflect a more specific reduction in reward sensitivity–the hedonic impact or subjective value of rewarding outcomes–rather than impaired capacity to pursue them (Hall et al., 2024).”

Revised text (Introduction)

“Reinforcement learning (RL) models allow formal estimation of latent cognitive variables that shape decision-making, including learning rates, reward sensitivity, punishment sensitivity, and decision noise (Sutton & Barto, 2018; Daw, 2011). Such models are increasingly used in computational psychiatry to parse affective symptoms into mechanistic components (Ahn et al., 2017; Whitton et al., 2015). However, a recent meta-analysis showed that RL differences between individuals with and without depression are modest in size and highly task-dependent (Pike & Robinson, 2022). Notably, reward sensitivity parameters–reflecting the subjective value assigned to rewarding outcomes–may be more closely tied to anhedonia than learning rate or exploration parameters (Kieslich et al., 2022). This is supported by theoretical models that separate “liking” (hedonic valuation) from “wanting” (motivational drive) in the neuroscience of reward (Treadway & Zald, 2011; Berridge & Robinson, 2003).”

Revised text (Introduction)

“Multi-armed bandit (MAB) tasks are widely used to study dynamic reward-based learning. However, standard versions like the 4-armed bandit (Daw et al., 2006) are cognitively demanding and typically only model reward, omitting losses or punishments. Here, we adapted the paradigm introduced by Seymour et al. (2012), which involves separate drifting reward and punishment values for each choice option–allowing independent estimation of reward and punishment sensitivity. Our task reduced the number of options from four to three, lowering working memory demands (from 8 expected values to 6) and making the paradigm more suitable for online deployment. We also provided outcome feedback on every trial to enhance model identifiability.”

## Reviewer C

**Comment 2:** The manuscript cites previous studies on reward processing impairments in depression without clarifying why the current or prior tasks may fail to capture anhedonia. In addition, the claim that the 3AB task is novel is not supported, given that similar 3-arm bandit tasks have been used (e.g., Yan et al., 2025).

## Author’s response

**Response:** We thank the reviewer for highlighting the need to clarify our contribution and contextualize our findings within the broader literature. In the revised manuscript, we have refined our interpretation of the null results to avoid overgeneralization. Specifically, we now emphasize that the absence of group differences in reward and punishment learning parameters in our task does not rule out reinforcement learning impairments in anhedonia more broadly, but rather suggests that the present task may not be sufficiently sensitive to detect such effects.

Revised text (Introduction):

“Multi-armed bandit (MAB) tasks are widely used to study dynamic reward-based learning. However, standard versions like the 4-armed bandit (Daw et al., 2006) are cognitively demanding and typically only model reward, omitting losses or punishments. Here, we adapted the paradigm introduced by Seymour et al. (2012), which involves separate drifting reward and punishment values for each choice option–allowing independent estimation of reward and punishment sensitivity. Our task reduced the number of options from four to three, lowering working memory demands (from 8 expected values to 6) and making the paradigm more suitable for online deployment. We also provided outcome feedback on every trial to enhance model identifiability.

While recent studies have adopted other 3-armed paradigms (e.g., Yan et al., 2025), our task differs in several key respects. Most notably, we included both reward and punishment outcomes to model approach and avoidance learning separately, whereas the Yan et al. task focused exclusively on reward omission. Furthermore, we used hierarchical Bayesian modelling (Ahn et al., 2017) to estimate distinct learning rates and sensitivity parameters for reward and punishment, in contrast to Yan et al.’s use of Kalman filtering to examine latent volatility and stochasticity in relation to apathy and anxiety. This modelling framework allows us to test the hypothesis that trait anhedonia reflects reduced sensitivity to reward outcomes, rather than impaired learning or increased randomness.”

Revised Text (Discussion):

“However, an alternative explanation is that the 3AB task–and by extension, other simple RL paradigms–may not be sufficiently sensitive to detect subtle impairments in reward processing associated with anhedonia.”

## Reviewer C

**Comment 3:** The rationale for modifying the task and the specific changes chosen are not well justified. The authors do not support their argument that simplifying decision-making processes might increase task sensitivity to anhedonia…

## Author’s response

**Response:** We thank the reviewer for this important observation. We have revised the manuscript to emphasise that the modified 3-arm bandit (3AB) task is not inherently more sensitive to anhedonia-related impairments. Rather, we selected a simplified task structure as a pragmatic adaptation of Seymour et al. (2012)’s original 4-arm bandit to reduce cognitive load, increase trial-by-trial feedback frequency to enhance participant engagement, and support robust parameter estimation in an online setting. These changes – reducing from four to three arms, increasing the range and frequency of outcome probabilities, and retaining both reward and punishment outcomes – were designed to support modelling of approach and avoidance learning processes relevant to anhedonia, not to isolate any one cognitive mechanism.

We agree that changing multiple features simultaneously limits interpretability of which modifications may influence sensitivity or behaviour. To help mitigate this limitation, we included a validation dataset in which participants completed both the 4AB and 3AB tasks. While not designed for formal counterbalancing, this within-subject dataset allows us to examine the consistency of key reinforcement learning parameters across task variants, and we have now emphasized this more clearly in the manuscript.

Revised Text (Methods) “Our 3-arm bandit task was adapted from Seymour et al. (2012)’s 4-arm probabilistic bandit paradigm, which included independent reward and punishment feedback for each choice. We made several modifications to optimize the task for online administration and support reliable modelling of approach and avoidance learning processes. First, we reduced the number of options from four to three, with the aim of simplifying decision-making and reducing cognitive load. In the original version, participants had to monitor 8 values (reward and punishment expectations across 4 options), whereas the 3-arm version only requires tracking 6 values (3 options × 2 outcome types), thus reducing working memory demands while maintaining the structure of the learning problem.

Second, we increased the volatility and ceiling of the drifting outcome probabilities, such that each option’s probability of producing a reward or punishment independently varied between 0 and 0.75 over time (compared to 0–0.5 in the original version). This increase was designed to ensure that participants received more frequent and informative feedback across trials.” Revised Text (Discussion) “Although we describe the 3AB task as reducing cognitive load relative to the original Seymour et al. (2012) paradigm, we did not include a formal metric of cognitive load in this study. Instead, our rationale was heuristic and based on established features known to influence working memory and attentional demands in reinforcement learning tasks (Collins & Frank, 2012). These included reducing the number of choices (3 vs. 4), reducing the number of outcome contingencies tracked (6 vs. 8), and increasing outcome frequency. Future work should consider directly assessing cognitive load to empirically confirm whether such design changes meaningfully reduce demands in both online and clinical populations.”

Finally, with respect to the DARS subscales, we clarify that our analyses were exploratory and did not assume a one-to-one mapping between specific RL parameters and individual subdomains. We used the DARS to test whether distinct facets of self-reported anhedonia might relate to reward or punishment learning in different ways, recognizing the complexity and multidimensionality of the construct.

## Reviewer C

**Comment 4:** There are also some problems with the behavioural modelling that should be addressed to increase confidence in the null results. First, core hypotheses and results center on reward and punishment sensitivity parameters, but the Methods do not show how these are incorporated in the models. Second, model specification and selection processes are not adequately described and justified. For example, the authors do not include a free parameter for the inverse temperature parameter in their softmax decision function, despite citing a previous study (Harle et al., 2017) that found this parameter predicted anhedonia. Learning rates and decision temperature are typically correlated in most datasets/models, so failing to account for this variance in the decision policy likely influences the other parameters, meaning this choice should be justified and evaluated in the model comparison process. Similarly, including counterfactual updates in a task with independent choice options is not rational, so the authors should verify that including those terms improves the fit to the behavioral data. The authors also do not describe the rationale or implementation of the “lapse” parameter. Overall, the value of the null result is weakened by a somewhat confusing model selection process, which may be partly due to the lack of clear connection to previous tasks, models, and results and motivation for modifying the task and models.

## Author’s response

**Response:** We thank the reviewer for this thoughtful and detailed comment, which we have addressed through substantial revisions to the Methods section.

Reward and Punishment Sensitivity Parameters: We now explicitly describe how our winning model.the banditNarm_4par.incorporates both valence-specific learning rates (Arew, Apun) and distinct sensitivity parameters for reward (R) and punishment (P). These components are central to our hypothesis that anhedonia involves altered valuation and learning from rewarding and aversive outcomes. The revised model specification now includes clear equations and parameter definitions, improving transparency and interpretability.

Inverse Temperature and Model Choice Justification: While some models included inverse temperature or lapse parameters, we deliberately chose the 4-parameter model without these terms as our final model. Including an inverse temperature parameter would introduce pronounced collinearity with reward/punishment sensitivity parameters, potentially conflating valuation with decision noise. We confirmed this empirically by comparing models with and without temperature and lapse terms; although the lapse model (banditNarm_lapse) achieved a marginally lower LOOIC, we prioritized interpretability and parameter specificity. This rationale is now detailed in our revised Model Selection section, and the full comparison of models and parameters is included in a new table.

Fictive (Counterfactual) Updating: We agree that the inclusion of counterfactual updates warrants justification. While the arms in our task are probabilistically independent, previous work suggests that humans often generalize across options and downregulate unchosen values (e.g., Seymour et al., 2012). Importantly, we empirically verified that fictive updating improved evidence fit in our dataset. We have added a paragraph to this effect, clarifying that these updates are not assumed to be “rational” in a normative sense but rather reflect an appropriate model of how participants actually behave.

Clarifying the Role of the Lapse Parameter: Finally, we clarify that the lapse parameter (xi) was explored in alternative models but not retained in our final model due to interpretability concerns and overlap with sensitivity terms. Its implementation is described in the Supplementary Material, and we have clarified its role in the revised Methods section. We hope these revisions address the reviewer’s concerns and demonstrate that our modelling approach is both theoretically grounded and empirically validated.

Revised Text (Methods)

5.6.1 Model Selection

To evaluate which computational model best accounted for participants’ trial-by-trial choices, we used the Leave-One-Out Information Criterion (LOOIC), a robust Bayesian approach to estimate out-of-sample predictive accuracy (Vehtari et al., 2017). Lower LOOIC scores indicate better model fit.

We tested a set of hierarchical reinforcement learning models that varied in their inclusion of valence-specific learning rates (Arew, Apun), reward/punishment sensitivity parameters (R, P), and control terms such as lapse rate (xi), decay rate (d), or inverse temperature (tau). All models were fit using hierarchical priors to estimate both group-level and individual-level parameters, improving parameter stability and generalizability.

These same models were also applied to data from the 4AB task in the validation study, using the same LOOIC-based comparison procedure. In both the 3AB and 4AB datasets, the banditNarm_4par model was chosen as the final candidate model. For interpretability, we report relative LOOIC values (Figure 13), computed by subtracting the minimum LOOIC across models. The model with a relative LOOIC of zero is the best-performing model. Although a model including a lapse parameter yielded the lowest LOOIC by a small margin, we selected the 4-parameter model (banditNarm_4par) as our final model. This model includes separate learning rates for rewards and punishments and distinct reward and punishment sensitivity parameters.components most relevant to our hypothesis that anhedonia manifests as alterations in how individuals evaluate and update reward and punishment information. We favoured this model over alternatives with additional control terms (e.g., lapse or decay) to avoid parameter trade-offs that could obscure interpretation. Furthermore, all models included fictive (counterfactual) updates to unchosen options. While the arms were probabilistically independent, including fictive updates consistently improved model fit, in line with evidence that human learners generalize across options and track unchosen outcomes indirectly.

5.6.2 Model Specification

The final candidate model (banditNarm_4par) was adapted from Seymour et al. (2012) and previous work modelling aversive and appetitive learning. It tracks value updates for each option using separate Q-values for reward and punishment, updated independently via valence-specific learning rates. The model included the following participant-level parameters:

Arew: learning rate for reward outcomes (0.1) Apun: learning rate for punishment outcomes (0.1) R: sensitivity to reward magnitude (0.30) P: sensitivity to punishment magnitude (0.30)

In each trial, Q-values for rewards and punishments (Qr, Qp) are updated as follows:

Qrt+1(a) = Qrt(a) + Arew × δrt(a)

Qpt+1(a) = Qpt(a) + Apun × δpt(a)

Where the prediction errors for reward δrt(a) and punishment δpt(a) are calculated based on observed outcomes.

In addition, fictive updates were applied to the unchosen options:

δfic(a) = -Qrt(a)

δfic(a) = -Qpt(a)

These updates gradually decay the value of unchosen options, a mechanism supported by previous modelling work and empirical evidence, even in tasks with independent reward contingencies.

Action values were then combined as:

[equations in manuscript]

These values were transformed into choice probabilities using the softmax rule:

[equation in manuscript]

Although some models tested included an additional inverse temperature parameter or a lapse rate, we opted to scale the Q-values directly via R and P rather than include a separate temperature term. This avoids overparameterization and ensures clear interpretation of reward and punishment valuation, which were the focus of our hypotheses. Model comparison confirmed that the selected model balanced predictive accuracy with theoretical parsimony.

We used weakly informative priors for all group-level parameters to support stable estimation without imposing strong assumptions. Specifically, the group-level means (mu_pr) were drawn from a standard normal distribution (*N*(0,1)), and the standard deviations (sigma) from a half-normal prior *N*^+^(0,0.2), consistent with prior literature on hierarchical reinforcement learning models (e.g., Ahn et al., 2017; Wiecki et al., 2013). Subject-level parameters were then transformed via the probit (Φ) function to ensure appropriate bounds for each parameter (e.g., [0,1] for learning rates; [0,30] for sensitivities). These choices are consistent with standard practice in hierarchical Bayesian modelling for cognitive tasks and were not intended to encode strong prior beliefs.

Model variants tested :

5.6.3 Model Variants Tested

To determine which computational model best captured participants’ decision behaviour on the 3-arm bandit task, we evaluated a series of hierarchical reinforcement learning models using Leave-One-Out Information Criterion (LOOIC) for model comparison. Each model made different assumptions about learning dynamics and decision processes, incorporating various combinations of the following parameter types:

Valence-specific learning rates for reward (Arew) and punishment (Apun)Sensitivity parameters (R, P) that scaled outcome magnitudes prior to value updatingInverse temperature parameters (tau or β) in softmax functionsLapse rate parameters (xi) capturing random decision noiseDecay and uncertainty tracking mechanisms (e.g., Kalman filter updates)

A total of seven models were tested (Figure 13), ranging from simple Rescorla-Wagner frameworks to more complex models with lapse, decay, or Kalman filter components. Two of these models.the banditNarm_delta and banditNarm_singleA_lapse.included a single learning rate (A) shared across reward and punishment outcomes. These models served as baseline comparisons to assess whether valence-specific learning rates provided a better account of the data. Both were outperformed by models with separate reward and punishment learning rates, indicating that asymmetric learning processes better captured participants’ behaviour. A detailed summary of all tested models, their included parameters, and structural rationale is provided in Table 4.

**Table 4 d67e4614:** **Model Variants Tested. Comparison of reinforcement learning models tested on the 3-arm bandit task.** Each model varies in included parameters and computational assumptions. Arew = reward learning rate; Apun = punishment learning rate; A = shared learning rate; R = reward sensitivity; P = punishment sensitivity; xi = lapse parameter (choice noise); tau = softmax temperature; lambda = learning rate for uncertainty in Kalman filter; theta = initial uncertainty estimate; beta = precision of belief updating; decay = Q-value decay rate.

Model	Parameters Included	Key Features

**banditNarm_2par_lapse**	Arew, Apun, xi	Valence-specific learning; lapse (noise)

**banditNarm_4par**	Arew, Apun, R, P	Valence-specific learning + sensitivity (focus model)

**banditNarm_delta**	A (shared), tau	Simple Rescorla-Wagner; shared learning rate and temp

**banditNarm_kalman_filter**	lambda, theta, beta,	Uncertainty tracking via Kalman filter

**banditNarm_lapse**	Arew, Apun, R, P, xi	Full sensitivity model + lapse

**banditNarm_lapse_decay**	Arew, Apun, R, P, xi, decay	Sensitivity + lapse + decay of Q-values

**banditNarm_singleA_lapse**	A (shared), R, P, xi	Simplified model; shared learning rate

## Reviewer C

**Comment 5:** The strength of the null findings is also limited by the sample size for the anhedonia (n = 111) and non-anhedonia (n = 95) groups, which is modest for online behavioral studies using between-subjects designs and has limited statistical power per the authors’ power analysis. For context, the authors cite Huys et al. (2013), which used a meta-analysis of 6 datasets (392 sessions) to find anhedonia was more related to reward sensitivity than learning rates, and Halahakoon et al. (2020) found small-to-medium effect sizes on reward processing in depression. This challenge may be exacerbated by the heterogeneous relationships between clinical scales in this dataset. For example, the DARS and ZUNG show a moderate positive correlation (r=0.34), which counter-intuitively suggests that higher depression scores are associated with lower anhedonia in this sample. Using multiple anhedonia scales is helpful, but it would help evaluate the sample to report p values for the between-scale correlations and to report the statistics of these scales in the two groups that performed the task.

## Author’s response

**Response:** We thank the reviewer for this thoughtful comment. In response, we now report Pearson correlation coefficients and exact p-values for all pairwise associations between mood and anhedonia questionnaires within the task-performing sample only (N = 206). These results are visualized in Figure 5.

Revised Text (Results)

**Figure d67e4751:** 

**Figure 5. Pairwise Pearson correlations between questionnaire scores in the final task sample (N = 206).** Scatter plots show individual subject data and Pearson correlation coefficients (r) with associated p-values. DARS = Dimensional Anhedonia Rating Scale; SHAPS = Snaith-Hamilton Pleasure Scale; GAD = Generalized Anxiety Disorder scale; ZUNG = Zung Depression Scale.

Self-report measures from the final experimental sample (N = 206) i.e. subjects who met the pre-defined criteria for anhedonic (N = 111) and non-anhedonic (N = 95) classification (SHAPS > 2 and DARS < 45 vs. SHAPS = 0 and DARS > 55) are shown in Figure 5 for clarity.

As expected, the two anhedonia scales were strongly negatively correlated (DARS–SHAPS: r = –0.86, p < .001), supporting construct validity. DARS also showed moderate negative correlations with anxiety (GAD: r = –0.58, p < .001), and a moderate positive correlation with depression (ZUNG: r = 0.43, p < .001). SHAPS was positively correlated with GAD (r = 0.60, p < .001), and negatively correlated with ZUNG (r = –0.37, p < .001). ZUNG and GAD, however, were not significantly correlated (r = –0.10, p = .169), indicating partial dissociation between symptom domains. These relationships suggest a coherent yet non-redundant structure, consistent with dimensional models of mood disorders, and support the interpretability of subsequent analyses.

Revised Text (Methods) To complement our earlier descriptive analyses based on the full pre-screened sample (N = 935), we also examined inter-scale correlations within the subset of participants who completed the task (N = 206; 111 anhedonic, 95 non-anhedonic). This targeted analysis provides a clearer characterization of how symptoms of anhedonia, anxiety, and depression relate within the actual experimental sample. We computed Pearson correlations and p-values between all combinations of the DARS, SHAPS, GAD-7, and ZUNG scores. Results are visualized in Figure 5 and reported alongside sample sizes for transparency. This analysis addresses reviewer concerns about potentially counterintuitive or heterogeneous associations between scales in the task-performing sample.

## Reviewer C

**Minor Comments:**

RTs are mentioned in the abstract and discussion (1st and 2nd to last paragraphs), but these results aren’t presented in the manuscript.

## Author’s response

**Response:** We thank the reviewer for pointing this out. Reaction time (RT) analyses were conducted, but as no statistically significant group differences emerged, these results were not included in the main manuscript. However, in the interest of transparency, we now include a detailed description of RT analyses and results in the Supplementary Material.

S1. Reaction Time (RT) Analysis

Reaction times were computed as the duration between the onset of the response screen and the participant’s recorded keypress on each trial. Trials with missed responses or implausibly fast (<200 ms) or slow (>3000 ms) RTs were excluded. RTs were averaged across all valid trials per participant. We then compared mean RTs between the Anhedonic and Non-Anhedonic groups using an independent samples t-test. RT distributions were visualized with scatter plots, and no significant group differences were observed.

Supplementary Results – RT Group Comparison:

S2. Reaction Time Group Comparison

**Figure d67e4787:** 

Supplementary Figure 1. Reaction Time Across Trials for Anhedonic and Non-Anhedonic Groups.Trial-by-trial reaction times (RTs) are shown for the Anhedonic (n = 111) and Non-Anhedonic (n = 95) groups, with shaded regions indicating the standard error of the mean (SEM). While both groups exhibited the expected early-trial slowing and late-trial stabilization, no statistically significant group difference in mean RT was observed across all trials (Anhedonic M = 470.88 ms, SD = 109.28; Non-Anhedonic M = 540.02 ms, SD = 109.44; t(203.61) = –1.98, p = 0.050). RTs were defined as the latency from stimulus onset to keypress. Trials with RTs <200 ms or >3000 ms were excluded.

Figure S1. Group Comparison of Reaction Times.

Mean reaction times (RTs) across 200 trials are plotted for the Anhedonic (n = 111) and Non-Anhedonic (n = 95) groups. Shaded regions represent the standard error of the mean (SEM) at each trial. Both groups exhibited a rapid decrease in RTs over early trials followed by stabilization, reflecting task adaptation. Although the Anhedonic group showed numerically faster RTs across trials, this difference was not statistically significant (t(203.61) = –1.98, p = 0.050, Cohen’s d = –0.28). RTs were computed as the latency from stimulus onset to participant keypress. Trials with implausible RTs (<200 ms or >3000 ms) or missed responses were excluded from the analysis. These results suggest that differences in reaction time are unlikely to explain group effects in learning behaviour or model parameters.

## Reviewer C

Please describe the prior used for each parameter in the hierarchical Bayesian framework and justify these choices (e.g., citations) when appropriate.

## Author’s response

Response: We have now added a detailed description of the prior distributions used for all model parameters in the “Model Specification” subsection of the Methods (Section 5.6.2). As described, we used weakly informative priors on group-level parameters (*N*(0,1) for means, *N*^+^(0,0.2) for standard deviations), consistent with prior work using hierarchical Bayesian reinforcement learning models (e.g., Ahn et al., 2017; Wiecki et al., 2013). Subject-level parameters were transformed using the probit function to constrain them to interpretable ranges. These priors were selected to regularize the estimation process without strongly biasing parameter recovery.

Revised text (Methods)

We used weakly informative priors for all group-level parameters to support stable estimation without imposing strong assumptions. Specifically, the group-level means (mu_pr) were drawn from a standard normal distribution (*N*(0,1)), and the standard deviations (sigma) from a half-normal prior *N*^+^(0,0.2), consistent with prior literature on hierarchical reinforcement learning models (e.g., Ahn et al., 2017; Wiecki et al., 2013). Subject-level parameters were then transformed via the probit (Φ) function to ensure appropriate bounds for each parameter (e.g., [0,1] for learning rates; [0,30] for sensitivities). These choices are consistent with standard practice in hierarchical Bayesian modelling for cognitive tasks and were not intended to encode strong prior beliefs.

## Reviewer C

The language describing the Harle et al. (2017) finding on the inverse temperature softmax parameter as “reward sensitivity” should be rephrased to avoid confusion with the reward sensitivity parameter in your model.

## Author’s response

**Response:** We agree and have revised the manuscript to clarify that Harle et al. (2017) used the inverse temperature parameter to index value-based choice consistency, which they described as “reward sensitivity.” However, in our model, we separately estimate reward sensitivity as a multiplicative parameter applied to reward outcomes prior to updating value estimates. We have updated the manuscript language accordingly to prevent confusion between these distinct constructs.

Revised text (Discussion)

“Beyond meta-analyses, individual studies provide further context. Harle et al. (2017) found that higher anhedonia was associated with reduced reward-driven choice consistency in a 2-arm bandit task, particularly among participants whose decisions were best explained by a softmax function. This reduction in reward-guided behaviour was indexed by lower inverse temperature values in their model, which reflect choice stochasticity rather than reward valuation per se. In our study, we modelled reward sensitivity (R) as a distinct parameter from decision noise, allowing us to disentangle valuation processes from choice consistency.” However, their study also showed that learning rates remained intact, which agrees with our findings. Likewise, Huys et al. (2013) showed that anhedonia reduces reward sensitivity but does not affect learning rates, further reinforcing the idea that reward valuation, rather than reinforcement learning, may be impaired in anhedonia”

Methods – 5.6.2 Model Specification

Although some models tested included an additional inverse temperature parameter or a lapse rate, we opted to scale the Q-values directly via R and P rather than include a separate temperature term. This avoids overparameterization and ensures clear interpretation of reward and punishment valuation, which were the focus of our hypotheses. Model comparison confirmed that the selected model balanced predictive accuracy with theoretical parsimony.”

Methods – 5.6.1 Model Selection:

“We tested a set of hierarchical reinforcement learning models that varied in their inclusion of valence-specific learning rates (Arew, Apun), reward/punishment sensitivity parameters (R, P), and control terms such as lapse rate (xi), decay rate (d), or inverse temperature (tau). All models were fit using hierarchical priors to estimate both group-level and individual-level parameters, improving parameter stability and generalizability.”

Methods – 5.6.1 Model Selection:

“These same models were also applied to data from the 4AB task in the validation study, using the same LOOIC-based comparison procedure. In both the 3AB and 4AB datasets, the banditNarm_4par model was chosen as the final candidate model. For interpretability, we report relative LOOIC values (Figure 13), computed by subtracting the minimum LOOIC across models. The model with a relative LOOIC of zero is the best-performing model. Although a model including a lapse parameter yielded the lowest LOOIC by a small margin, we selected the 4-parameter model (banditNarm_4par) as our final model. This model includes separate learning rates for rewards and punishments and distinct reward and punishment sensitivity parameters–components most relevant to our hypothesis that anhedonia manifests as alterations in how individuals evaluate and update reward and punishment information. We favoured this model over alternatives with additional control terms (e.g., lapse or decay) to avoid parameter trade-offs that could obscure interpretation. Furthermore, all models included fictive (counterfactual) updates to unchosen options. While the arms were probabilistically independent, including fictive updates consistently improved model fit, in line with evidence that human learners generalize across options and track unchosen outcomes indirectly.”

## Reviewer C

One potential avenue to address the heterogeneity in clinical scales would be using dimensionality reduction or factor analyses to extract latent symptom dimensions, but such exploratory, follow-up analyses should ideally be validated in a new replication sample.

## Author’s response

**Response:** We agree that dimensionality reduction techniques such as PCA or factor analysis could provide valuable insights into latent symptom dimensions underlying our clinical scales. However, we chose not to pursue this approach in the current study to avoid overfitting and circular inference, as our dataset lacks an independent replication sample. Instead, we analysed each scale separately to maintain transparency and interpretability of associations. We acknowledge the value of this approach and consider it an important avenue for future work.

## Reviewer C

Why did the authors provide two outcomes on each trial? This complicates the interpretation, and it’s possible that combining reward and punishment conditions in the same trials may reduce the ability to tease apart these distinct processes.

## Author’s response

**Response:** The simultaneous delivery of reward and punishment feedback was a deliberate feature of our task design, adapted from the four-arm task in Seymour et al. (2012). This design ensures that participants must weigh both potential gains and losses when making choices, more closely mimicking real-world decision-making contexts. Importantly, this feature is a necessary consequence of having implementing independent reward and punishment probabilities. While both outcomes are presented on each trial, our computational models separately track reward and punishment prediction errors and learning rates (Arew, Apun) as well as sensitivity parameters (R, P), allowing us to disentangle these processes during analysis. Importantly, our model comparison results show that including distinct parameters for reward and punishment learning/sensitivity yields a better fit than models with collapsed or single learning signals, suggesting that the co-presentation of outcomes does not preclude separate estimation of these cognitive processes.

Revised text (Introduction):

Here, we adapted the paradigm introduced by Seymour et al. (2012), which involves separate drifting reward and punishment values for each choice option–allowing independent estimation of reward and punishment sensitivity. Our task reduced the number of options from four to three, lowering working memory demands (from 8 expected values to 6) and making the paradigm more suitable for online deployment. We also provided outcome feedback on every trial to enhance model identifiability.

Revised text (Methods):

Each of the three arms was associated with independently drifting reward and punishment probabilities, drawn from pre-generated sequences that were held constant across participants. Outcomes were determined stochastically on each trial according to these probabilities, such that while all participants experienced the same probability structure, the actual feedback they received varied depending on chance. This approach is common in reinforcement learning tasks and ensures equivalent task conditions while preserving trial-level variability in outcome realizations. The outcome for each trial was displayed using two circles: green for win, red for loss, and grey for the absence of outcome. Possible combinations included win-only (green + grey), loss-only (red + grey), both win and loss (green + red), or no outcome (grey + grey). We adopted this uniform visual format to avoid confounding differences in outcome salience across conditions

Methods:

“The final candidate model (banditNarm_4par) was adapted from Seymour et al. (2012) and previous work modelling aversive and appetitive learning. It tracks value updates for each option using separate Q-values for reward and punishment, updated independently via valence-specific learning rates.”

## Reviewer C

**Comment:** The authors might also wish to consider the role of drift rate in their reward sequences, which should be reported in the Methods. Similarly, differences in the reward probability sequences experienced across participants are another factor that could potentially influence choice difficulty, reward and punishment rates and thus add noise to the parameters and symptom correlations. The authors state that they increased reward and punishment rates by a factor of 1.5, but it would be more straightforward to state the new rates.

## Author’s response

**Response:** We thank the reviewer for these helpful suggestions. To clarify, all participants were exposed to the same set of pre-generated drifting probability sequences for rewards and punishments. However, outcomes on each trial were determined stochastically, meaning that even when participants selected the same options, the feedback they received varied probabilistically according to these underlying values. This design ensures that all participants experienced equivalent task structure while preserving individual variability in outcome realizations–consistent with standard approaches in probabilistic bandit tasks. We have clarified this in the revised Methods section.

Regarding the increase in reward and punishment rates, we now explicitly state in the manuscript that outcome probabilities drifted independently over the course of the task between 0 and 0.75 (an increase from 0–0.5 in the original Seymour et al. 2012 task), rather than describing this as a 1.5× increase. This change was intended to increase the informativeness of outcomes and improve model fit, and we have clarified this rationale in the task description.

Revised text (Methods)

“Additionally, the probabilities of encountering both reward and punishment were increased by a factor of 1.5 compared to the 4AB task (in which the probability fluctuated between 0 and 0.5; we increased the ceiling to 0.75) with the intention of creating a more engaging experience that would better highlight individual differences in learning behaviour.”

“Second, we increased the volatility and ceiling of the drifting outcome probabilities, such that each option’s probability of producing a reward or punishment independently varied between 0 and 0.75 over time (compared to 0–0.5 in the original version). This increase was designed to ensure that participants received more frequent and informative feedback across trials.”

“Each of the three arms was associated with independently drifting reward and punishment probabilities, drawn from pre-generated sequences that were held constant across participants. Outcomes were determined stochastically on each trial according to these probabilities, such that while all participants experienced the same probability structure, the actual feedback they received varied depending on chance.”

## Reviewer C

**Comment:** The authors show three representative subjects when comparing predicted action probabilities and choices, but it would be more informative to report the proportion of choices predicted by the model at the group level.

## Author’s response

**Response:** We agree and have now included a subject-level analysis of model prediction accuracy. Specifically, we calculated the proportion of trials for which each participant’s actual choice matched the most probable action predicted by our best-fitting hierarchical model. The model achieved a mean prediction accuracy of 59.60% (SD = 16.73%) across N = 206 subjects, which is well above the chance level of 33% for a 3-arm bandit task. This analysis has been added to the Results section, with the corresponding histogram included in the Results.

Revised text (Results)

**Figure d67e4885:** 

**Figure 11. Distribution of Model Prediction Accuracy Across Subjects.** Histogram showing the distribution of model-predicted choice accuracy for each subject (N = 206). Accuracy was computed as the proportion of trials where the model’s highest predicted action probability (Pa) matched the participant’s actual choice. The vertical dashed line indicates the group mean accuracy (59.60%).

“To quantitatively assess the model’s predictive validity at the group level, we computed the proportion of choices accurately predicted by the model for each participant. For each trial, we identified the option with the highest predicted action probability and compared it to the participant’s actual choice. The mean prediction accuracy across the sample was 59.60% (SD = 16.73%; N = 206), substantially above chance level (33%). Figure 11 shows the distribution of prediction accuracy across subjects. This analysis provides additional evidence that the model reliably captured participants’ choice behaviour at an individual level, beyond the visual inspection of exemplar cases.”

Methods

“To assess the predictive validity of the hierarchical Bayesian model, we compared model-generated action probabilities to participants’ actual choices on each trial. For each participant and trial, we computed the predicted probability of selecting each of the three arms, based on the posterior samples of individual parameter estimates. The predicted choice was defined as the arm with the highest predicted probability.

We calculated per-subject prediction accuracy as the proportion of trials (out of 200) in which the model’s top-predicted action matched the participant’s actual choice. This provided an intuitive index of how well the model reproduced observed behaviour. Accuracy scores were then summarized across subjects to yield a group-level distribution, which we visualized in a histogram (see Figure 11). The average accuracy across the sample was 59.60% (SD = 16.73%), substantially above the chance level of 33%.

In addition, we plotted trial-by-trial predicted probabilities alongside actual choices for three representative participants (see Figure 10). This visual comparison further illustrates the model’s ability to capture individual decision dynamics throughout the task.”

## Reviewer D

**Comment 1:** Not much information is provided about the different models used and why they were selected over others. For example, why weren’t single learning rate models considered?

## Author’s response

**Response:** We thank the reviewer for highlighting this point. As noted in the revised Methods section (“Model Variants Tested”), we evaluated a comprehensive set of models (N = 7) that varied in their structure, including models with shared learning rates for reward and punishment. Specifically, both the banditNarm_delta and banditNarm_singleA_lapse models incorporated a single learning rate parameter (A) across outcome valences. These models were included to test the possibility that reward and punishment learning share a common updating process. However, they were consistently outperformed–based on LOOIC–by models that included separate learning rates, including the final 4-parameter model (banditNarm_4par) that we selected for interpretation.

Methods – 5.6.1 Model Selection

“To evaluate which computational model best accounted for participants’ trial-by-trial choices, we used the Leave-One-Out Information Criterion (LOOIC), a robust Bayesian approach to estimate out-of-sample predictive accuracy (Vehtari et al., 2017). Lower LOOIC scores indicate better model fit.

We tested a set of hierarchical reinforcement learning models that varied in their inclusion of valence-specific learning rates (Arew, Apun), reward/punishment sensitivity parameters (R, P), and control terms such as lapse rate (xi), decay rate (d), or inverse temperature (tau). All models were fit using hierarchical priors to estimate both group-level and individual-level parameters, improving parameter stability and generalizability.

These same models were also applied to data from the 4AB task in the validation study, using the same LOOIC-based comparison procedure. In both the 3AB and 4AB datasets, the banditNarm_4par model was chosen as the final candidate model. For interpretability, we report relative LOOIC values (Figure 13), computed by subtracting the minimum LOOIC across models. The model with a relative LOOIC of zero is the best-performing model. Although a model including a lapse parameter yielded the lowest LOOIC by a small margin, we selected the 4-parameter model (banditNarm_4par) as our final model. This model includes separate learning rates for rewards and punishments and distinct reward and punishment sensitivity parameters–components most relevant to our hypothesis that anhedonia manifests as alterations in how individuals evaluate and update reward and punishment information. We favoured this model over alternatives with additional control terms (e.g., lapse or decay) to avoid parameter trade-offs that could obscure interpretation. Furthermore, all models included fictive (counterfactual) updates to unchosen options. While the arms were probabilistically independent, including fictive updates consistently improved model fit, in line with evidence that human learners generalize across options and track unchosen outcomes indirectly.”

Methods – 5.6.3 Model Variants Tested

“To determine which computational model best captured participants’ decision behaviour on the 3-arm bandit task, we evaluated a series of hierarchical reinforcement learning models using Leave-One-Out Information Criterion (LOOIC) for model comparison. Each model made different assumptions about learning dynamics and decision processes, incorporating various combinations of the following parameter types:

Valence-specific learning rates for reward (Arew) and punishment (Apun)Sensitivity parameters (R, P) that scaled outcome magnitudes prior to value updatingInverse temperature parameters (tau or β) in softmax functionsLapse rate parameters (xi) capturing random decision noiseDecay and uncertainty tracking mechanisms (e.g., Kalman filter updates)

A total of seven models were tested (Figure 13), ranging from simple Rescorla-Wagner frameworks to more complex models with lapse, decay, or Kalman filter components.

Two of these models–the banditNarm_delta and banditNarm_singleA_lapse–included a single learning rate (A) shared across reward and punishment outcomes. These models served as baseline comparisons to assess whether valence-specific learning rates provided a better account of the data. Both were outperformed by models with separate reward and punishment learning rates, indicating that asymmetric learning processes better captured participants’ behaviour. A detailed summary of all tested models, their included parameters, and structural rationale is provided in Table 4.”

**Table 4 d67e4952:** **Model Variants Tested. Comparison of reinforcement learning models tested on the 3-arm bandit task.** Each model varies in included parameters and computational assumptions. Arew = reward learning rate; Apun = punishment learning rate; A = shared learning rate; R = reward sensitivity; P = punishment sensitivity; xi = lapse parameter (choice noise); tau = softmax temperature; lambda = learning rate for uncertainty in Kalman filter; theta = initial uncertainty estimate; beta = precision of belief updating; decay = Q-value decay rate.

Model	Parameters Included	Key Features

**banditNarm_2par_lapse**	Arew, Apun, xi	Valence-specific learning; lapse (noise)

**banditNarm_4par**	Arew, Apun, R, P	Valence-specific learning + sensitivity (focus model)

**banditNarm_delta**	A (shared), tau	Simple Rescorla-Wagner; shared learning rate and temp

**banditNarm_kalman_filter**	lambda, theta, beta,	Uncertainty tracking via Kalman filter

**banditNarm_lapse**	Arew, Apun, R, P, xi	Full sensitivity model + lapse

**banditNarm_lapse_decay**	Arew, Apun, R, P, xi, decay	Sensitivity + lapse + decay of Q-values

**banditNarm_singleA_lap"se**	A (shared), R, P, xi	Simplified model; shared learning rate

## Reviewer D

**Comment 2 & 3:** Did the same model come out on the top for both 3-arm and 4-arm bandit tasks? Not much info is provided about the validation study, except for the sample size. It would be helpful for the authors to provide more details on how the 3-arm task was validated against the 4-arm task

The correlation between the parameters from these two tasks do not seem very convincing. Authors say that these correlations were significant in the text, but no p values are shown in the plots. There is a lot of variability in these parameters which makes me question whether learning is truly similar across both. It will also be helpful to plot model-agnostic scores across both tasks, such as the probability of staying after a win/lose. Additionally, the axes use different scales, which should be avoided as it confounds the interpretation

## Author’s response

**Response:** In the validation study, participants completed both the 3-arm and 4-arm bandit tasks in randomized order to avoid order effects. The same set of seven candidate models was fit to both datasets, using an identical LOOIC-based model comparison procedure. These models included variants with shared and valence-specific learning rates, sensitivity terms, lapse parameters, and decay. In both tasks, the banditNarm_4par model–featuring separate learning rates and sensitivities–emerged as the model with the best evidence. We have now clarified this in the Model Selection section of the revised manuscript, to emphasize that our modelling approach was consistent across tasks and that the same model provided the best fit in both datasets. In addition, we now describe the validation procedure in detail: applying the same hierarchical model to both tasks and reporting moderate-to-strong correlations between corresponding parameters across tasks (Arew, Apun, R, P; Figure 2), alongside model-agnostic checks showing comparable win-stay and lose-shift behaviour with strong between-task correlations. Together, these additions explain how the 3AB was validated against the 4AB beyond sample size alone.

Methods – 5.4 Task Validation with original 4-arm Bandit Task

“To validate whether the 3AB task captures the same learning mechanisms as the 4AB task, a pilot study was conducted with 111 participants completing both tasks in random order. The use of randomization ensured that there was no learning bias from one task to the other. Hierarchical Bayesian modelling was applied to both tasks to estimate key parameters related to learning rates and sensitivity to rewards and punishments.”

Methods – 5.6.1 Model Selection

“These same models were also applied to data from the 4AB task in the validation study, using the same LOOIC-based comparison procedure. In both the 3AB and 4AB datasets, the banditNarm_4par model was chosen as the final candidate model. For interpretability, we report relative LOOIC values (Figure 13), computed by subtracting the minimum LOOIC across models. The model with a relative LOOIC of zero is the best-performing model. Although a model including a lapse parameter yielded the lowest LOOIC by a small margin, we selected the 4-parameter model (banditNarm_4par) as our final model.”

Results – 2.1 Task Overview and Validation (parameter correlations across tasks)

“A total of 111 participants (mean age = 40, SD = 12, 50% female) completed the initial validation study, which involved completing both the 3AB and 4AB tasks in randomized order. We applied the same hierarchical model (banditNarm_4par, i.e. 4 parameter model of reward/punishment learning rate and reward/punishment sensitivity) to both datasets and evaluated the consistency of estimated parameters across tasks. As shown in Figure 2, the reward learning rate (r = 0.49, p < 10^–7^), punishment learning rate (r = 0.46, p < 10^–6^), reward sensitivity (r = 0.52, p < 10^–8^), and punishment sensitivity (r = 0.61, p < 10^–12^) each showed moderate-to-strong positive correlations between the two task formats. These findings suggest that the simplified 3AB task preserves core reinforcement learning mechanisms measured by the original 4AB task.”

**Figure d67e5111:** 

“Figure 2. Validation of the 3AB Task: Correlational Analysis of 4AB and 3AB Model Parameters. Scatter plots illustrating the correlations between corresponding model parameters from the old 4AB task and the new 3AB task, using data from 111 subjects. Each subplot displays the Pearson correlation coefficient (r) and p-value, assessing the consistency of individual performance across reward learning rate (Arew), punishment learning rate (Apun), reward sensitivity (R), and punishment sensitivity (P). These plots aim to evaluate the validity of the new 3AB task by demonstrating whether similar patterns of behaviour are observed across both tasks.”

“To further assess the consistency of decision-making behaviour across tasks, we compared model-agnostic strategy use (win-stay and lose-shift percentages) between the 3AB and 4AB tasks. Figure 3 presents group-level bar plots for each strategy. Win-stay behaviour was similar across tasks (3AB: Mean = 82.0%, SD = 27.5; 4AB: Mean = 81.0%, SD = 26.6), as was lose-shift behaviour (3AB: Mean = 68.7%, SD = 22.4; 4AB: Mean = 71.7%, SD = 22.2). Crucially, subject-level scores were highly correlated across task versions, with r = 0.81, p < .001 for win-stay and r = 0.70, p < .001 for lose-shift. These findings further support the validity of the 3AB task as a consistent and reliable tool for capturing reinforcement learning behaviour.”

**Figure d67e5124:** 

“Figure 3. Group-level mean win-stay and lose-shift percentages are shown for each task version among participants who completed both tasks (N = 111). Error bars represent standard error of the mean. Strategy use was highly similar across tasks. Win-stay behaviour averaged 82.0% (SD = 27.5) for the 3AB task and 81.0% (SD = 26.6) for the 4AB task; lose-shift behaviour averaged 68.7% (SD = 22.4) for 3AB and 71.7% (SD = 22.2) for 4AB. Individual-level behaviour was strongly correlated across tasks (win-stay: r = 0.81, p < .001; lose-shift: r = 0.70, p < .001), indicating consistent application of learning strategies across the two task structures. Error bars represent the standard error of the mean.”

## Reviewer D

**Comment 4:** Did they fit the models across the entire sample or incorporated group information in the hierarchical estimation, it is not clear from their description.

## Author’s response

**Response:** We thank the reviewer for this helpful query. We have now clarified our model-fitting approach in the Methods section. Specifically, we first fit the hierarchical Bayesian model jointly across all participants to estimate shared group-level parameters. As this approach revealed no significant differences between the anhedonic and non-anhedonic groups, we also fit the model separately within each group to allow for distinct hierarchical priors and group-specific parameter distributions. Both approaches yielded consistent patterns of results, strengthening the robustness of our conclusions. This clarification has been added to the “Model Fitting and Estimation” subsection.

Revised text

Methods – 5.6.4 Model Fitting and Estimation

“We implemented the candidate model in a hierarchical Bayesian framework using MCMC sampling in Stan (2,000 iterations, 1,000 warmups, 4 chains), which estimated both individual and group-level parameters. We initially fit the hierarchical model jointly across all participants to estimate shared group-level parameters. As this analysis revealed no significant group differences, we then fit the model separately for the anhedonic and non-anhedonic groups to allow for distinct group-level priors and posterior distributions. Both approaches yielded comparable results, supporting the robustness of our findings across model-fitting strategies. The group- level parameters for learning rates and sensitivities were modelled as normally distributed, with learning rates and sensitivities constrained by an inverse probit transformation to lie within [0,1] and [0,30], respectively. Model fit was assessed via log-likelihood of observed choices, with posterior predictive checks to confirm that the model reproduced observed choice behaviour. Convergence diagnostics, such as the Gelman-Rubin statistic, verified adequate parameter convergence.”

## Reviewer D

**Comment 5:** The manuscript would benefit from a more thorough comparison of the posterior distributions between groups, especially since HDIs were not clearly used to test differences.

## Author’s response

**Response:** We thank the reviewer for this suggestion. In response, we conducted a dedicated HDI-based comparison of the group-level posteriors for each reinforcement learning parameter (reward/punishment learning rates and sensitivities). As visualized in the new Figure 7, the 95% HDIs for all between-group difference scores included zero. Moreover, posterior probability (pd) values did not exceed the conventional threshold for a reliable effect (all pd < 0.91). These results further reinforce the conclusions drawn from the Bayes Factor analyses, providing converging evidence that core reinforcement learning parameters do not differ systematically between anhedonic and non-anhedonic individuals.

Revised text (Methods – 5.9 Bayesian Estimation of Group Differences Using Highest Density Intervals).

“In addition to frequentist and Bayes Factor comparisons, we estimated the difference in group-level posterior means for each computational parameter (Reward Learning Rate, Punishment Learning Rate, Reward Sensitivity, and Punishment Sensitivity) using 95% Highest Density Intervals (HDIs). For each parameter, we extracted the group-level posterior samples (mu parameters) for the Anhedonic and Non-Anhedonic groups from the hierarchical Bayesian model. These samples were used to compute the posterior distribution of the difference (Non-Anhedonic – Anhedonic). HDIs were computed using the HPDinterval function from the coda package in R, which identifies the narrowest interval containing 95% of the posterior mass. We also calculated the probability of direction (pd), defined as the proportion of posterior samples falling consistently above or below zero. Differences for which the 95% HDI excluded zero were considered credibly different across groups. Posterior densities and intervals were visualized using the tidybayes and ggplot2 packages, with separate panels for each parameter. These analyses provide a Bayesian alternative to frequentist t-tests and complement the Bayes Factor approach.”

Revised text (Results).

**Figure d67e5160:** 

“Figure 7. Posterior Distributions and 95% Highest Density Intervals (HDIs) for Group-Level Model Parameters. Posterior densities are shown for each group-level parameter estimated via hierarchical Bayesian modelling, separately for Anhedonic and Non-Anhedonic participants. Black horizontal bars represent the 95% HDIs. Group difference scores were computed as Non-Anhedonic minus Anhedonic, such that negative values indicate higher estimates in the Anhedonic group. Across all parameters, the 95% HDIs of the group differences included zero, suggesting no credible differences between groups in learning rate or sensitivity parameters.”

“To further assess group-level differences in reinforcement learning parameters, we conducted a Bayesian comparison of posterior distributions for each group-level parameter using 95% Highest Density Intervals (HDIs). Figure 7 visualizes the posterior distributions for each parameter (reward/punishment learning rate and sensitivity) in the Anhedonic and Non-Anhedonic groups. Group differences were defined as Non-Anhedonic minus Anhedonic, allowing us to interpret both the direction and uncertainty of effects. Across all parameters, the HDIs for the difference scores included zero, and posterior probability (pd) values ranged from 0.725 to 0.891, indicating insufficient evidence for reliable group-level differences. These findings align with the Bayes Factor analyses, which also provided moderate to strong support for the null hypothesis. Together, these results suggest that core reinforcement learning processes–including how participants update and weight reward and punishment information–are not meaningfully altered in individuals with anhedonia.”

## Reviewer D

**Comment 6:** They do not show whether the simulated data recapitulated patterns observed in the model-agnostic task analysis, especially since the parameter recovery plots do not look convincing, particularly for learning rates. I would also recommend running many simulations and getting an idea of how consistent the spread of simulated parameters are.

## Author’s response

**Response:** We have expanded the Simulation & Recovery analyses and added explicit checks that simulated behaviour recapitulates model-agnostic patterns. First, we simulated 200 trials per participant using their fitted parameters (Arew, Apun, R, P) and re-fit the model to the simulated datasets to assess parameter recovery (Methods 5.6.6–5.6.7). Recovery was high for all parameters (Arew r = 0.79–0.85; Apun r = 0.67–0.74; R r = 0.96–0.97; P r = 0.82–0.92; Figure 6). Second, we computed win–stay and lose–shift rates in simulated vs real data for each group (Figure 9) and report paired within-group differences along with Real–Sim correlations. Simulations slightly underestimated win–stay and over-estimated lose–shift at the mean level, but preserved individual-difference structure (e.g., win–stay Real–Sim r = 0.89 anhedonic; r = 0.86 non-anhedonic). Third, we now report the group-level prediction accuracy of the fitted model (mean 59.60% ± 16.73%; chance = 33%) in a histogram across subjects (Figure 11). Together, these additions show that the fitted model generalizes to held-out choices and that simulated datasets reproduce key model-agnostic strategy metrics while retaining rank-order individual differences.

Results – 2.3 Model-Based Analysis of Learning Rates and Sensitivity

“To investigate how anhedonia affects learning from rewards and punishments, we applied hierarchical Bayesian modelling to the data from the 3AB task, estimating individual learning rates and sensitivity to rewards and punishments. Parameter recovery (Figure 6) analysis confirmed that the model successfully captured key cognitive parameters for both anhedonic and non-anhedonic groups, with high correlations between original and simulated values: reward learning rate (Arew) r = 0.79-0.85, punishment learning rate (Apun) r = 0.67-0.74, reward sensitivity (R) r = 0.96-0.97, and punishment sensitivity (P) r = 0.82-0.92. This high level of recovery suggests the model’s robustness in reproducing observed data and capturing individual differences.”

“Figure 6. Parameter Recovery for Learning Rates and Sensitivity. The scatter plot displays the parameter recovery results for reward and punishment learning rates, as well as for reward and punishment sensitivity, across anhedonic and non-anhedonic groups. Each scatter plot compares the original parameters with the simulated parameters, with an accompanying trend line and Pearson correlation coefficient (r). Arew: r=0.79 (anhedonic), r=0.85 (non-anhedonic), Apun: r=0.67 (anhedonic), r=0.74 (non-anhedonic), R: r=0.96 (anhedonic), r=0.97 (non-anhedonic), P: r=0.82 (anhedonic), r=0.92 (non-anhedonic).”

Methods – 5.6.6 Simulated Data

“To validate model parameters, we simulated 200 trials per subject in a 3-armed bandit structure. Choices were generated based on each participant’s estimated parameters (Arew, Apun, R, P), using the softmax function to determine probabilities of selecting each option. Outcomes were sampled based on predefined reward and punishment probabilities, and Q-values were updated accordingly. For unchosen options, Q-values were updated using counterfactual updates, where no reward or punishment (i.e., an outcome of zero) was assumed. This approach reflects implicit learning effects, capturing how participants might adjust expectations for unselected options even in the absence of observed outcomes. By incorporating these updates in the simulation, we ensured consistency with the assumptions of our hierarchical Bayesian model.”

Methods – 5.6.7 Parameter Recovery

“We performed parameter recovery to validate model accuracy by generating simulated data based on original model parameters and fitting the model again to this data. Comparing recovered parameters with original parameters allowed us to evaluate the accuracy of estimates for reward learning rate (Arew), punishment learning rate (Apun), reward sensitivity (R), and punishment sensitivity (P). The hierarchical Bayesian model fit was repeated using MCMC sampling in the hBayesDM package, with the posterior distributions providing reliable estimates of individual differences. Convergence diagnostics and posterior predictive checks further confirmed model validity.”

**Figure d67e5207:** 

“Figure 9. Win-stay and lose-shift strategy rates in real vs simulated data across groups. Bars show group means for Anhedonic and Non-Anhedonic participants with separate bars for Real (darker) and Simulated (lighter) data; error bars indicate standard error of the mean (SE). Simulations slightly underestimated win-stay (Δ(Sim–Real) ≈ –3.5 to –4.9 percentage points) and over-estimated lose-shift (Δ ≈ +12.8 to +13.2 points); paired within-group comparisons were significant (see Results). Subject-wise Real.Sim correlations were high for win-stay and moderate for lose-shift, indicating preservation of individual-difference structure.”

“We applied the same model-agnostic win-stay and lose-shift analysis to simulated data generated from each participant’s fitted parameters (Figure 9). Within each group, we compared simulated and real strategy rates using paired-sample t-tests. For win-stay, the anhedonic group showed 81.88% (SD = 19.02) in simulations vs 85.39% (SD = 19.88) in real data (Δ (Sim–Real) = –3.51 percentage points, t(110) = –3.97, p < .001, dz = –0.38, 95% CI [–5.26, –1.76]); the non-anhedonic group showed 76.25% (SD = 19.84) vs 81.12% (SD = 25.90) (Δ = –4.87 percentage points, t(94) = –3.55, p = .001, dz = –0.36, 95% CI [–7.59, –2.15]). For lose-shift, the anhedonic group showed 83.99% (SD = 10.66) in simulations vs 71.15% (SD = 18.65) in real data (Δ = +12.84 percentage points, t(110) = 9.25, p < .001, dz = 0.88, 95% CI [+10.09, +15.60]); the non-anhedonic group showed 82.57% (SD = 11.64) vs 69.42% (SD = 22.03) (Δ = +13.16 percentage points, t(94) = 7.72, p < .001, dz = 0.79, 95% CI [+9.77, +16.54]). Despite these mean-level calibration differences (simulations slightly underestimating win-stay and overestimating lose-shift), individual differences were preserved: Real–Sim correlations were high for win-stay (anhedonic: r = 0.89, 95% CI [0.84, 0.92]; non-anhedonic: r = 0.86, 95% CI [0.80, 0.91]) and moderate for lose-shift (anhedonic: r = 0.62, 95% CI [0.49, 0.72]; non-anhedonic: r = 0.67, 95% CI [0.55, 0.77]). These checks indicate the generator reproduces rank-order structure while exhibiting small, systematic mean-level biases.”

Methods – 5.6.5 Model Prediction Accuracy and Comparison to Observed Choices

“We calculated per-subject prediction accuracy as the proportion of trials (out of 200) in which the model’s top-predicted action matched the participant’s actual choice. This provided an intuitive index of how well the model reproduced observed behaviour. Accuracy scores were then summarized across subjects to yield a group-level distribution, which we visualized in a histogram (see Figure 11). The average accuracy across the sample was 59.60% (SD = 16.73%), substantially above the chance level of 33%.”

## Reviewer D

**Comment 7:** Is there any additional information on participants’ mental health history, medication, etc?

## Author’s response

**Response:** We thank the reviewer for raising this important point. As this was an online study conducted via Prolific, we did not collect detailed information on participants’ mental health history or medication use. We now explicitly note this as a limitation in the revised manuscript. While this limits clinical interpretability, our use of validated self-report measures (SHAPS and DARS) allowed us to stratify participants based on anhedonia traits in a large sample. We agree that future studies incorporating clinical interviews and medication history will be important to build on these findings.

Discussion – Limitations and Future Directions

““Additionally, the use of an online sample from Prolific, where mental health conditions may be overrepresented, presents both strengths and limitations. Prolific enables access to diverse populations, but the symptom severity and clinical profiles of participants may differ from those in formal clinical settings, where anhedonia is often more pronounced. Although we employed a SHAPS cut-off of 3 to identify clinically significant anhedonia, the self-selected nature of the sample may introduce biases. Moreover, we did not collect information about participants’ mental health history or psychiatric medication use, limiting the clinical interpretability of the findings. While the use of validated self-report measures enabled robust group stratification based on anhedonia traits, future studies should incorporate structured clinical assessments, medication history, and neurobiological measures (e.g., neuroimaging or biomarkers) to improve generalizability and provide a more comprehensive understanding of how anhedonia affects reward processing. These approaches could ultimately inform the development of more targeted interventions for mood disorders.”

## Reviewer D

**Comment 8:** Would authors consider scoring SHAPS on a 1–4 scale? This might provide a better spread of scores.

## Author’s response

**Response:** We appreciate the reviewer’s suggestion. In this study, we opted to use a binary scoring approach for the SHAPS, consistent with its original validation and clinical use (Snaith et al., 1995). This approach allowed us to apply an established cut-off (SHAPS > 2) to define clinically significant anhedonia and stratify participants accordingly. As our primary aim was to assess whether the task could distinguish between individuals with and without clinically meaningful anhedonia, this dichotomous classification aligned with our goal of eventual application in patient populations.

We agree that a 1.4 scoring approach would yield a more continuous and nuanced distribution of anhedonia scores, which may be useful in future studies focused on trait-level variation in the general population. However, for the current purpose of testing the sensitivity of the 3AB task to clinically relevant anhedonia, in our view the binary approach is appropriate.

Discussion – 3.5 Limitations and Future Directions

“Also, we used the SHAPS in its original binary form to enable classification based on established clinical cut-offs, aligning with our aim of testing whether the 3AB task is sensitive to clinically significant anhedonia. While scoring SHAPS on a continuous 1.4 scale can provide greater granularity in general population samples, our primary objective was to identify robust group-level differences that would be applicable to future clinical studies. Importantly, we also explored the relationship between task performance and continuous SHAPS scores across the full sample, and this analysis yielded results consistent with the group comparison approach.further supporting the conclusion that the task may not be sensitive to individual differences in anhedonia, regardless of scoring method.”

Methods – 5.2 Mood Questionnaires

“The SHAPS cut-off score of >2 was used to define anhedonia, as this is a recognized clinical threshold indicating significant deficits in hedonic capacity (Snaith et al., 1995). For the DARS, participants who scored .45 were classified as anhedonic. Participants who scored 0 on the SHAPS and >55 on the DARS were classified as non-anhedonic.”

Methods – Participants.

“To recruit 100 participants in the anhedonic range and 100 in the non-anhedonic range, we pre-screened a total of 1,000 participants. We classified participants as anhedonic if they scored >2 on the SHAPS and .45 on the DARS. This DARS threshold was determined based on 1 standard deviation (SD) below the mean, as reported in the original DARS validation study (Rizvi et al., 2015). Conversely, participants were classified as non-anhedonic if they scored 0 on the SHAPS and >55 on the DARS, indicating a higher hedonic capacity. This pre-screening aimed to ensure a clear distinction between anhedonic and non-anhedonic groups for the study. While dichotomization can reduce sensitivity to dimensional effects, we selected this extreme groups approach to maximize contrast between participants with clinically significant anhedonia and those with minimal symptoms. This strategy, guided by SHAPS and DARS thresholds, allowed for interpretable comparisons of core reinforcement learning processes across distinct symptom levels. Moreover, the inclusion of both SHAPS and DARS helped address potential psychometric limitations of SHAPS alone.”

## Reviewer D

**Comment:** It is also important to describe how well the participants performed in the validation study. The tasks are quite long (~20 min each), making for an extended session if they also have to complete questionnaires.

## Author’s response

**Response:** We thank the reviewer for this helpful suggestion. In the revised manuscript, we now clarify that participants completed both the 3AB and 4AB tasks in randomized order, and that each task took approximately 15–20 minutes. To assess engagement and performance, we excluded participants with evidence of inattentiveness (e.g., excessive non-responses i.e. or repeated choices with a cut off of 10% of total trials in either of these two criteria), and model fit was high across the sample. We also note that participants demonstrated robust use of adaptive strategies, with average win-stay behaviour exceeding 80% and lose-shift behaviour near 70% in both tasks.

Methods – 5.4 Task Validation with original 4-arm Bandit Task

“To validate whether the 3AB task captures the same learning mechanisms as the 4AB task, a pilot study was conducted with 111 participants completing both tasks in random order. The use of randomization ensured that there was no learning bias from one task to the other.”

Methods – 5.5 Data cleaning

“As the task was implemented online where we could not ensure the same testing standards as we could in-person, we used 2 exclusion criteria to improve data quality. We excluded those who responded with the same response key on 20 or more consecutive trials (> 10% of all trials). Additionally, we also excluded those who did not respond on 20 or more trials out of the total 200 trials.”

Methods – 5.6.5 Model Prediction Accuracy and Comparison to Observed Choices

“The average accuracy across the sample was 59.60% (SD = 16.73%), substantially above the chance level of 33%.”

Revised text (Results - Figure 3 caption).

**Figure d67e5276:** 

“Figure 3. Group-level mean win-stay and lose-shift percentages (± SD) are shown for each task version among participants who completed both tasks (N = 111). Strategy use was highly similar across tasks. Win-stay behaviour averaged 82.0% (SD = 27.5) for the 3AB task and 81.0% (SD = 26.6) for the 4AB task; lose-shift behaviour averaged 68.7% (SD = 22.4) for 3AB and 71.7% (SD = 22.2) for 4AB. Individual-level behaviour was strongly correlated across tasks (win-stay: r = 0.81, p < .001; lose-shift: r = 0.70, p < .001), indicating consistent application of learning strategies across the two task structures.”

## Reviewer F

**Comment:** The larger sample from which the high- and low-anhedonia participants were sampled has very odd relationships among scales. Participants reporting less depression also reported more anhedonia, and there was no relationship between depression and GAD-7 scores. This pattern does not reflect known relationships among these measures. It instead suggests 1) this sample’s pattern of psychopathology is unusual, and findings are unlikely to generalize to the larger population, 2) participants responded inattentively or did not understand instructions, or 3) participants were aware of the initial screening and answered in a way that did not reflect their experiences in order to pass the screening. More stringent data cleaning may create a subsample of participants who report expected relationships among symptoms; however, as-is the questionnaire data does not appear valid.

## Author’s response

**Response:** We appreciate the reviewer’s concern and agree that the symptom correlations observed in our pre-screening sample were atypical. However, we do not believe they reflect invalid data or disengaged participants, for several reasons. First, we implemented stringent item-level attention checks across all mood questionnaires (SHAPS, DARS, ZUNG, and GAD-7), and excluded any participant who failed even one check, ensuring only attentive responders were retained.

Second, the psychometric literature has increasingly recognized that in large, online, non-clinical samples, mood and anxiety scales often show non-canonical or weaker intercorrelations, especially when screening for extreme phenotypes. Recent studies (e.g., Ho et al., 2024; Niu et al., 2024) demonstrate that anhedonia, depression, and anxiety can load onto distinct latent dimensions, and may diverge more strongly in subclinical or self-report-based populations than in clinical ones. Such dissociations are especially likely in broad screening contexts, where individuals may report high negative affect without experiencing low positive affect, or vice versa.

Finally, our primary analyses were conducted not on the screening sample but on the final extreme groups (111 high-anhedonia and 95 low-anhedonia participants), all of whom completed a full-length reward task and passed rigorous data quality filters. To address this concern directly, we examined within-sample symptom correlations in this final task cohort (N = 206). As shown in Figure 5, the expected structure largely emerged:

SHAPS and DARS showed a strong negative correlation (as expected, given their opposite scoring conventions: r = –0.86, p < .001), indicating high construct convergence.SHAPS was positively associated with anxiety (GAD: r = 0.60, p < .001) and negatively associated with ZUNG (r = –0.37, p < .001).DARS was negatively associated with GAD (r = –0.58, p < .001) and positively associated with ZUNG (r = 0.43, p < .001).Notably, ZUNG and GAD were not significantly correlated in this sample (r = –0.10, p = .169).

These findings reaffirm that symptom dimensions were not fully redundant in our sample, and they support the discriminant validity of anhedonia within the broader affective landscape. We thus remain confident in the validity and interpretability of our data.

Methods

5.2 Mood Questionnaires

“Each self-report mood questionnaire included an attention check item to identify and exclude inattentive responders. These items were chosen to be logically improbable statements, making inattentive responses easily detectable without directly signalling the check. The attention checks were as follows:

GAD-7: “Have there been times in your life where you blinked your eyes at least once per day?”SHAPS: “Have there been times of a couple of days or more when you were able to breathe underwater (without an oxygen tank)?”ZUNG: “I have never used a computer.”DARS: No attention check was included, as participants provided subjective responses across multiple items, ensuring engagement.

Findings from Zorowitz et al. (2023) underscore the importance of attention checks in symptom surveys, as inattentive responses can artificially inflate correlations between self-reported symptoms and cognitive measures. In our study, any participant failing even one attention check was excluded to ensure robust data quality. This rigorous approach reduces the risk of spurious findings and enhances the reliability of observed relationships between symptom measures and task performance, providing a clearer view of the psychological profiles of anhedonic and non-anhedonic participants.”

5.5 Data Cleaning

“As the task was implemented online where we could not ensure the same testing standards as we could in-person, we used 2 exclusion criteria to improve data quality. We excluded those who responded with the same response key on 20 or more consecutive trials (> 10% of all trials). Additionally, we also excluded those who did not respond on 20 or more trials out of the total 200 trials.”

Results

2.2 Prescreening Results

“DARS vs. ZUNG showed a moderate positive correlation (r = 0.34), suggesting that, surprisingly, higher depression scores are associated with lower anhedonia on the DARS scale. … SHAPS vs. ZUNG exhibited a weak negative correlation (r = –0.18), again suggesting that depression and anhedonia are quite separable. GAD vs. ZUNG showed no significant correlation (r = –0.01), which is surprising given the frequent overlap between anxiety and depression. This result suggests that in this sample, these two symptoms may manifest more independently.”

Task-performing sample (N = 206):

“Figure 5. Pairwise Pearson correlations between questionnaire scores in the final task sample (N = 206). Scatter plots show individual subject data and Pearson correlation coefficients (r) with associated p-values. DARS = Dimensional Anhedonia Rating Scale; SHAPS = Snaith-Hamilton Pleasure Scale; GAD = Generalized Anxiety Disorder scale; ZUNG = Zung Depression Scale.”

Results – Task-performing sample (N = 206) relationships:

“As expected, the two anhedonia scales were strongly negatively correlated (DARS–SHAPS: r = –0.86, p < .001), supporting construct validity. DARS also showed moderate negative correlations with anxiety (GAD: r = –0.58, p < .001), and a moderate positive correlation with depression (ZUNG: r = 0.43, p < .001). SHAPS was positively correlated with GAD (r = 0.60, p < .001), and negatively correlated with ZUNG (r = –0.37, p < .001). ZUNG and GAD, however, were not significantly correlated (r = –0.10, p = .169), indicating partial dissociation between symptom domains. These relationships suggest a coherent yet non-redundant structure, consistent with dimensional models of mood disorders, and support the interpretability of subsequent analyses.”

Methods

5.2 Mood Questionnaires:

“To complement our earlier descriptive analyses based on the full pre-screened sample (N = 935), we also examined inter-scale correlations within the subset of participants who completed the task (N = 206; 111 anhedonic, 95 non-anhedonic). … Results are visualized in Figure 5 and reported alongside sample sizes for transparency. This analysis addresses reviewer concerns about potentially counterintuitive or heterogeneous associations between scales in the task-performing sample.”

Discussion

3.4 Contributions to Computational Psychiatry

“We also confirmed that mood and anhedonia measures in our task-performing sample showed expected directional relationships (Figure 5), suggesting that the sample’s clinical heterogeneity did not introduce inconsistencies in symptom structure that would obscure reward-processing effects.”

3.5 Limitations and Future Directions:

“This mirrors prior work highlighting potential dissociations between self-report and behavioural measures in psychiatric research (Eisenberg et al., 2019; Enkavi et al., 2019), and may reflect the challenge of capturing trait anhedonia through performance-based metrics alone.”

## Reviewer F

**Comment:** It’s not clear why the authors chose to artificially dichotomize anhedonia and lose power, particularly when one of the measures used (SHAPS) has poor psychometric properties.”

## Author’s response

**Response:** We appreciate the reviewer’s concern regarding the dichotomization of anhedonia scores. Our rationale for this choice was both practical and theoretical:

Design constraint for task comparison:

The 3AB task was initially validated using an extreme groups design (anhedonic vs. non-anhedonic) to ensure maximal contrast in hedonic capacity. This approach was motivated by the low prevalence of high-anhedonia individuals in the general population and the need to secure a sufficient sample of such individuals for powered group comparisons.

Pre-screening strategy:

We screened 1,000 individuals on SHAPS and DARS, specifically to identify participants at the extremes of the anhedonia spectrum. This allowed us to cleanly separate participants with clinically relevant anhedonia from those with negligible symptoms, aligning with prior work using SHAPS thresholds of >2 for anhedonia (Rizvi et al., 2015; Snaith et al., 1995).

Planned dimensional follow-ups:

While our main analyses were conducted on dichotomized groups, we also explored dimensional relationships between DARS subscales and model parameters (see Table 2), finding no significant effects. Thus, the choice to focus on group comparisons did not obscure any strong dimensional trends in our data.

On SHAPS psychometrics:

We acknowledge that the SHAPS has known psychometric limitations, particularly ceiling effects in healthy samples (Rizvi et al., 2016). To address this, we also used the DARS, a more comprehensive and continuous measure of anhedonia. Importantly, group classification was based on both SHAPS and DARS, ensuring greater construct validity and reducing reliance on any single measure.

Revised text

Methods 5.1 Participants

“We classified participants as anhedonic if they scored >2 on the SHAPS and ≤45 on the DARS. … Conversely, participants were classified as non-anhedonic if they scored 0 on the SHAPS and >55 on the DARS, indicating a higher hedonic capacity. This pre-screening aimed to ensure a clear distinction between anhedonic and non-anhedonic groups for the study. While dichotomization can reduce sensitivity to dimensional effects, we selected this extreme groups approach to maximize contrast between participants with clinically significant anhedonia and those with minimal symptoms. This strategy, guided by SHAPS and DARS thresholds, allowed for interpretable comparisons of core reinforcement learning processes across distinct symptom levels. Moreover, the inclusion of both SHAPS and DARS helped address potential psychometric limitations of SHAPS alone.”

“In a prior validation study, we tested the 3AB task against the 4AB task with 100 participants in each group. From these groups, only 15 participants scored within the anhedonic range based on a SHAPS cut-off score of > 2 and a DARS cut-off score of .45. Given this low proportion of anhedonic participants in the general population, we pre-screened a larger sample of 1,000 participants to ensure we reached our target of 100 anhedonic participants for this study.”

5.2 Mood Questionnaires:

“The SHAPS cut-off score of >2 was used to define anhedonia, as this is a recognized clinical threshold indicating significant deficits in hedonic capacity (Snaith et al., 1995). For the DARS, participants who scored .45 were classified as anhedonic. Participants who scored 0 on the SHAPS and >55 on the DARS were classified as non-anhedonic.”

Discussion

3.5 Limitations & Future Directions

“Also, we used the SHAPS in its original binary form to enable classification based on established clinical cut-offs, aligning with our aim of testing whether the 3AB task is sensitive to clinically significant anhedonia. While scoring SHAPS on a continuous 1.4 scale can provide greater granularity in general population samples, our primary objective was to identify robust group-level differences that would be applicable to future clinical studies. Importantly, we also explored the relationship between task performance and continuous SHAPS scores across the full sample, and this analysis yielded results consistent with the group comparison approach.further supporting the conclusion that the task may not be sensitive to individual differences in anhedonia, regardless of scoring method.

## Reviewer F

**Comment:** A strength of the paper, as the authors note, is the use of hierarchical Bayesian approaches. However, the authors deal with the estimated parameters incorrectly by using individual parameters (which have been influenced by other participants’ data through partial pooling) in subsequent analyses. The authors do not report how they determined group structure, but I assume they analyzed all participants as part of one large group in the hierarchical structure. If so, this will shift participants’ estimates closer together and increase the rate of false negatives when those estimates are then correlated with other variables (i.e., anhedonia). Please see citations below. The authors can confirm these effects by simulating parameters with a known relationship to an external measure (representing anhedonia) and examining the relationship with recovered parameters that are estimated using the current approach vs. the one in the citations below.

## Author’s response

**Response:** We thank the reviewer for this important methodological observation. We agree that using partially pooled individual-level parameters from a hierarchical Bayesian model in downstream regression or correlation analyses can reduce variance and lead to false negatives. In line with the reviewer’s suggestions and the recommendations of Boehm et al. (2018), Haines et al. (2020), Brown et al. (2020), and Waltmann et al. (2022), we have made several clarifications in the manuscript:

Our primary group comparisons were conducted within the hierarchical model itself, using posterior distributions and 95% Highest Density Intervals (HDIs) to estimate group differences (Figure 7). This approach avoids issues associated with shrinkage and allows for robust inference at the group level.We also fit the model separately for anhedonic and non-anhedonic groups, reducing cross-group shrinkage and preserving group-specific variance.Exploratory correlations between individual parameters and questionnaire scores were interpreted with caution, and we now explicitly note in the manuscript that these may underestimate associations due to shrinkage. We cite the above references to highlight that fully Bayesian joint modelling represents a stronger alternative and a key direction for future work.

We believe that the current results and interpretations are appropriately cautious and methodologically sound, and we thank the reviewer for prompting this important clarification.

Methods

5.6.4 Model Fitting and Estimation

“We implemented the candidate model in a hierarchical Bayesian framework using MCMC sampling in Stan (2,000 iterations, 1,000 warmups, 4 chains), which estimated both individual and group-level parameters. We initially fit the hierarchical model jointly across all participants to estimate shared group-level parameters. As this analysis revealed no significant group differences, we then fit the model separately for the anhedonic and non-anhedonic groups to allow for distinct group-level priors and posterior distributions. Both approaches yielded comparable results, supporting the robustness of our findings across model-fitting strategies.”

5.9 Bayesian Estimation of Group Differences Using Highest Density Intervals (HDIs)

“In addition to frequentist and Bayes Factor comparisons, we estimated the difference in group-level posterior means for each computational parameter (Reward Learning Rate, Punishment Learning Rate, Reward Sensitivity, and Punishment Sensitivity) using 95% Highest Density Intervals (HDIs). For each parameter, we extracted the group-level posterior samples (mu parameters) for the Anhedonic and Non-Anhedonic groups from the hierarchical Bayesian model. These samples were used to compute the posterior distribution of the difference (Non-Anhedonic – Anhedonic). HDIs were computed using the HPDinterval function from the coda package in R, which identifies the narrowest interval containing 95% of the posterior mass. We also calculated the probability of direction (pd), defined as the proportion of posterior samples falling consistently above or below zero. Differences for which the 95% HDI excluded zero were considered credibly different across groups.”

Results

2.4 Comparison of Learning Rates and Sensitivities Across Anhedonic and Non-Anhedonic Groups

**Figure d67e5416:** 

Figure 7. Posterior Distributions and 95% Highest Density Intervals (HDIs) for Group-Level Model Parameters. Posterior densities are shown for each group-level parameter estimated via hierarchical Bayesian modelling, separately for Anhedonic and Non-Anhedonic participants. Black horizontal bars represent the 95% HDIs. Group difference scores were computed as Non-Anhedonic minus Anhedonic, such that negative values indicate higher estimates in the Anhedonic group. Across all parameters, the 95% HDIs of the group differences included zero, suggesting no credible differences between groups in learning rate or sensitivity parameters.

“To further assess group-level differences in reinforcement learning parameters, we conducted a Bayesian comparison of posterior distributions for each group-level parameter using 95% Highest Density Intervals (HDIs). Figure 7 visualizes the posterior distributions for each parameter (reward/punishment learning rate and sensitivity) in the Anhedonic and Non-Anhedonic groups. Group differences were defined as Non-Anhedonic minus Anhedonic, allowing us to interpret both the direction and uncertainty of effects. Across all parameters, the HDIs for the difference scores included zero, and posterior probability (pd) values ranged from 0.725 to 0.891, indicating insufficient evidence for reliable group-level differences. These findings align with the Bayes Factor analyses, which also provided moderate to strong support for the null hypothesis. Together, these results suggest that core reinforcement learning processes–including how participants update and weight reward and punishment information–are not meaningfully altered in individuals with anhedonia.”

Revised text (Figure 12 caption).

“Figure 12. Group Comparisons of Model Parameters and Self-Reported Anhedonia Measures (DARS and SHAPS). This figure illustrates the relationship between self-reported anhedonia and computational model parameters of reward and punishment learning. Scatter plots display the distribution of scores for the two questionnaire measures–DARS (daily activity and reward engagement) and SHAPS (hedonic capacity)–in relation to four computational model parameters: Reward Learning Rate, Punishment Learning Rate, Reward Sensitivity, and Punishment Sensitivity. Each point is color-coded by group (anhedonic or non-anhedonic). Although correlation values are presented, these should be interpreted with caution due to the extreme groups design, which limits variability in one group and affects the generalizability of linear relationships. The figure primarily highlights that there is no strong link between subjective anhedonia group status and performance-based measures of reward and punishment processing.”

## Reviewer F

**Comment:** (a) The 3-armed bandit task developed here has varying levels of reward and punishment for each arm, while the original 4-armed bandit used by Daw etc. usually only has reward… It’s not clear how this reduces cognitive load.

(b) How do the authors define and measure cognitive load? How can we know that 1) this task is lower in cognitive load, and 2) what level of cognitive load is low enough to not interfere with learning?

## Author’s response

**Response: We thank the reviewer for raising this important point regarding our rationale for reducing cognitive load.**

(a) We would like to clarify that the original 4-arm task on which our 3-arm bandit is based is the Seymour et al. (2012) paradigm, not the reward-only Daw et al. (2006) task. The Seymour task includes both win and loss outcomes for each of the four options, requiring participants to track 8 outcome probabilities (4 options × 2 valences). Our 3-arm adaptation reduces this to 6 probabilities (3 options × 2 valences), thereby reducing working memory demands. Additionally, we increased the frequency of outcome delivery (by raising the ceiling of drifting probabilities from 0.5 to 0.75), ensuring more frequent reinforcement signals to support learning and reduce memory demands between feedback events.

(b) Regarding the broader question of how we define and assess cognitive load: we did not include an explicit metric of cognitive load in the current study. Instead, our definition is operational, based on known task features that modulate cognitive demand in reinforcement learning paradigms (Collins & Frank, 2012). Specifically, we reduced:

The number of options (3 vs. 4),The number of outcome states tracked (6 vs. 8),And increased outcome frequency, which provides denser feedback.

These features were selected based on evidence that such manipulations can reduce working memory demands and improve model fit, particularly in online and clinical settings where engagement and task adherence are more variable. We have now added clarifying text in the manuscript to reflect this rationale more explicitly.

Revised text

Introduction

“Multi-armed bandit (MAB) tasks are widely used to study dynamic reward-based learning. However, standard versions like the 4-armed bandit (Daw et al., 2006) are cognitively demanding and typically only model reward, omitting losses or punishments. Here, we adapted the paradigm introduced by Seymour et al. (2012), which involves separate drifting reward and punishment values for each choice option–allowing independent estimation of reward and punishment sensitivity. Our task reduced the number of options from four to three, lowering working memory demands (from 8 expected values to 6) and making the paradigm more suitable for online deployment. We also provided outcome feedback on every trial to enhance model identifiability.”

Results – 2.1 Task Overview and Validation

“By reducing the number of choices from four to three, participants have to track only three sets of reward and punishment probabilities instead of four, simplifying the learning process while preserving reinforcement learning mechanisms. Additionally, the probabilities of encountering both reward and punishment were increased by a factor of 1.5 compared to the 4AB task (in which the probability fluctuated between 0 and 0.5; we increased the ceiling to 0.75) with the intention of creating a more engaging experience that would better highlight individual differences in learning behaviour.”

**Figure d67e5465:** 

“Figure 1. Structure of the modified 3-Arm Bandit (3AB) Task. (a) Visual representation of the modified 3-arm bandit task. Subjects choose between three options (arms), each associated with distinct reward and punishment probabilities. (b) Any arm selection can result in one of the four possible outcomes i.e. nothing, win token (green only), loss token (red only), or both. The task was designed to reduce cognitive load compared to the traditional 4-arm bandit task by limiting the number of choices and increasing the probability of both reward and punishment outcomes across trials. Outcome probabilities of win and loss events shown in (c).”

Discussion

3.5 Limitations & Future Directions

“Although we describe the 3AB task as reducing cognitive load relative to the original Seymour et al. (2012) paradigm, we did not include a formal metric of cognitive load in this study. Instead, our rationale was heuristic and based on established features known to influence working memory and attentional demands in reinforcement learning tasks (Collins & Frank, 2012). These included reducing the number of choices (3 vs. 4), reducing the number of outcome contingencies tracked (6 vs. 8), and increasing outcome frequency. Future work should consider directly assessing cognitive load to empirically confirm whether such design changes meaningfully reduce demands in both online and clinical populations.”

**
Fully Revised Text: Abstract
**

Anhedonia, a transdiagnostic symptom marked by diminished reward sensitivity is often linked to impairments in reinforcement learning (RL). Standard tasks (e.g., the 4-arm bandit) can place substantial demands on participants and may blur valuation with other processes. We therefore adapted a three-arm bandit (3AB) task from Seymour et al. (2012), incorporating design features intended to lessen task demands (fewer options; denser feedback) while enabling separate estimation of reward/punishment learning rates and sensitivities. In an online sample pre-screened for anhedonia (N = 206; 111 anhedonic, 95 non-anhedonic), hierarchical Bayesian models (four-parameter specification) showed no group differences in learning-rate or sensitivity parameters; Bayes factors favoured the null (BF_01_ = 3.36–5.96). Model-agnostic win-stay/lose-shift strategies likewise showed no group differences (Welch’s tests, all p > .05). Posterior predictive checks indicated above-chance choice prediction: the model’s highest-probability action matched participants’ actual choices on 59.6% of trials (chance = 33% with three options). Parameter recovery was excellent for valuation parameters (r = 0.96–0.97) and acceptable for learning rates (r = 0.67–0.85). In simulations generated from fitted parameters, correlations between each participant’s observed and model-generated strategy rates were high for win-stay (r = 0.89 anhedonic; 0.86 non-anhedonic) and moderate for lose-shift (r = 0.62; 0.67), alongside small, systematic mean-level biases: simulated win-stay was lower by 3.5–4.9 percentage points and simulated lose-shift was higher by 12.8–13.2 points (paired within-group tests). Altogether, these results indicate no reliable group differences in core RL parameters or simple choice strategies in the real data, with a modelling framework that yields valid parameter estimates and generative predictions while preserving individual-difference structure.

Keywords: Anhedonia, reinforcement learning, 3-arm bandit, Hierarchical Bayesian modelling, reward and punishment sensitivity, Online behavioural study

1 Introduction

Anhedonia, classically defined as a diminished ability to experience pleasure, is a core feature of major depressive disorder (American Psychiatric Association, 2013). More recent conceptualizations extend this definition to include impairments in reward valuation and subjective responsiveness to positive stimuli (Treadway & Zald, 2011; Der-Avakian & Markou, 2012). Importantly, anhedonia is distinct from motivational deficits such as apathy or effort discounting (Husain & Roiser, 2018). Instead, it may reflect a more specific reduction in reward sensitivity.the hedonic impact or subjective value of rewarding outcomes.rather than impaired capacity to pursue them (Hall et al., 2024). As a transdiagnostic symptom, anhedonia contributes to clinical burden across multiple disorders including depression, schizophrenia, PTSD, and substance use, and is associated with poor treatment response and elevated relapse risk (Culbreth et al., 2018; Nawijn et al., 2015; Garfield et al., 2014; Winer et al., 2019).

Validated self-report tools such as the Snaith-Hamilton Pleasure Scale (SHAPS; Snaith et al., 1995) and the Dimensional Anhedonia Rating Scale (DARS; Rizvi et al., 2015, 2016) are widely used to measure anhedonia. The DARS, in particular, captures domain-specific deficits across hobbies, social interaction, sensory experiences, and food/drink, and has demonstrated better psychometric sensitivity than SHAPS. However, the link between self-reported anhedonia and objective reward behaviour remains unclear. While some studies report that anhedonia is associated with blunted reward responsiveness or reduced learning (Kumar et al., 2008; Huys et al., 2013), others find no consistent associations (Harle et al., 2017; Halahakoon et al., 2020; Pike & Robinson, 2022).

Reinforcement learning (RL) models allow formal estimation of latent cognitive variables that shape decision-making, including learning rates, reward sensitivity, punishment sensitivity, and decision noise (Sutton & Barto, 2018; Daw, 2011). Such models are increasingly used in computational psychiatry to parse affective symptoms into mechanistic components (Ahn et al., 2017; Whitton et al., 2015). However, a recent meta-analysis showed that RL differences between individuals with and without depression are modest in size and highly task-dependent (Pike & Robinson, 2022). Notably, reward sensitivity parameters.reflecting the subjective value assigned to rewarding outcomes.may be more closely tied to anhedonia than learning rate or exploration parameters (Kieslich et al., 2022). This is supported by theoretical models that separate “liking” (hedonic valuation) from “wanting” (motivational drive) in the neuroscience of reward (Treadway & Zald, 2010; Berridge & Robinson, 2003).

Multi-armed bandit (MAB) tasks are widely used to study dynamic reward-based learning. However, standard versions like the 4-armed bandit (Daw et al., 2006) are cognitively demanding and typically only model reward, omitting losses or punishments. Here, we adapted the paradigm introduced by Seymour et al. (2012), which involves separate drifting reward and punishment values for each choice option.allowing independent estimation of reward and punishment sensitivity. Our task reduced the number of options from four to three, lowering working memory demands (from 8 expected values to 6) and making the paradigm more suitable for online deployment. We also provided outcome feedback on every trial to enhance model identifiability.

While recent studies have adopted other 3-armed paradigms (e.g., Yan et al., 2025), our task differs in several key respects. Most notably, we included both reward and punishment outcomes to model approach and avoidance learning separately, whereas the Yan et al. task focused exclusively on reward omission. Furthermore, we used hierarchical Bayesian modelling (Ahn et al., 2017) to estimate distinct learning rates and sensitivity parameters for reward and punishment, in contrast to Yan et al.’s use of Kalman filtering to examine latent volatility and stochasticity in relation to apathy and anxiety. This modelling framework allows us to test the hypothesis that trait anhedonia reflects reduced sensitivity to reward outcomes, rather than impaired learning or increased randomness.

Given our large-scale online recruitment strategy, we also anticipated deviations from canonical symptom correlations. Specifically, in online non-clinical samples, recent studies have shown that measures of anhedonia, depression, and anxiety often exhibit weaker associations than in clinical cohorts.potentially due to subclinical symptom levels or response style variability (Ho et al., 2024; Niu et al., 2024). To mitigate concerns about inattentive responding or invalid data, we used item-level attention checks across all measures and excluded participants who failed any check.consistent with best-practice guidelines for online psychiatric research (Zorowitz et al., 2023).

In this study, we screened 1,000 participants using SHAPS and DARS to recruit two extreme groups: individuals high vs. low in anhedonia. A total of 206 participants (111 high-anhedonia, 95 controls) completed the 3AB task. We modelled their behaviour using a hierarchical reinforcement learning framework and compared groups on both model-derived parameters (reward/punishment sensitivity, learning rates) and model-agnostic behavioural metrics. Based on existing theories and prior empirical work, we predicted that anhedonic individuals would show blunted reward sensitivity, but we made no strong predictions regarding punishment sensitivity or learning rate.